



# Variations of dissolved greenhouse gases (CO$_2$, CH$_4$, N$_2$O) in the Congo River network overwhelmingly driven by fluvial-wetland connectivity

Alberto V. Borges[1,*], François Darchambeau[1,**], Thibault Lambert[1,***], Cédric Morana[2], George Allen[3], Ernest Tambwe[4], Alfred Toengaho Sembaito[4], Taylor Mambo[4], José Nlandu Wabakhangazi[5], Jean-Pierre Descy[1], Cristian R. Teodoru[2,****], Steven Bouillon[2]

[1] Chemical Oceanography Unit, University of Liège, Liège, Belgium
[2] Department of Earth and Environmental Sciences, KULeuven, Leuven, Belgium
[3] Department of Geography, Texas A&M University, USA
[4] Université de Kisangani, Centre de Surveillance de la Biodiversité, DRC Congo
[5] Congo Atomic Energy Commission, Kinshasa, DRC Congo

[*] alberto.borges@uliege.be
** Present address: Direction générale opérationnelle Agriculture, Ressources naturelles et Environnement, Service Publique de Wallonie, Belgium
*** Present address: University of Lausanne, Institute of Earth Surface Dynamics, Lausanne, Switzerland
**** Present address: Eidgenössische Technische Hochschule Zürich, Switzerland.





**Abstract**

We report the spatial variations of dissolved carbon dioxide ($CO_2$), methane ($CH_4$), and nitrous oxide ($N_2O$) concentrations in the lowland part of the Congo River network obtained during ten field expeditions carried out between 2010 and 2015, in the eastern part of the basin (Democratic Republic of Congo). Two transects of 1,650 km were carried out from the city of Kisangani to the city of Kinshasa, along the longest possible navigable section of the river, and corresponding to 41% of the total length of the mainstem. Additionally, three time series of $CH_4$ and $N_2O$ were obtained at fixed points in the mainstem of the middle Congo (2013-2018, biweekly sampling), in the mainstem of the lower Kasaï (2015-2017, monthly sampling), and in the mainstem of the middle Oubangui (2010-2012, biweekly sampling). The variations of dissolved $N_2O$ concentrations were modest, with values oscillating around the concentration corresponding to saturation with the atmosphere, with $N_2O$ saturation level ($\%N_2O$) ranging between 0% and 561% (average 142%). The relatively narrow range of $\%N_2O$ variations was consistent with low $NH_4^+$ ($2.3\pm1.3$ $\mu$mol $L^{-1}$) and $NO_3^-$ ($5.6\pm5.1$ $\mu$mol $L^{-1}$) levels in these near pristine rivers and streams with low agriculture pressure on the catchment (croplands correspond to 0.1% of catchment land cover of sampled rivers), dominated by forests (~70% of land cover). The co-variations of $\%N_2O$, $NH_4^+$, $NO_3^-$, and dissolved oxygen saturation level ($\%O_2$) indicate $N_2O$ removal by sedimentary denitrification in low $O_2$, high $NH_4^+$ and low $NO_3^-$ environments (typically small and organic matter rich streams) and $N_2O$ production by nitrification in high $O_2$, low $NH_4^+$ and high $NO_3^-$ (typical of larger rivers that are poor in organic matter). Surface waters were very strongly over-saturated in $CO_2$ and $CH_4$ with respect to atmospheric equilibrium, with values of the partial pressure of $CO_2$ ($pCO_2$) ranging between 1,087 and 22,899 ppm (equilibrium ~400 ppm), and dissolved $CH_4$ concentrations ranging between 22 and 71,428 nmol $L^{-1}$ (equilibrium ~2 nmol $L^{-1}$). Spatial variations were overwhelmingly more important than seasonal variations for $pCO_2$, $CH_4$ and $\%N_2O$, and than diurnal (day-night) variations for $pCO_2$. The wide range of $pCO_2$ and $CH_4$ variations was consistent with the equally wide range of $\%O_2$ (0.3-122.8%) and of dissolved organic carbon (DOC) (1.8-67.8 mg $L^{-1}$), indicative of intense processing of organic matter that generated these two greenhouse gases. However, the emission rate of $CO_2$ to the atmosphere from riverine surface waters was on average about 10 times higher than the flux of $CO_2$





produced by aquatic net heterotrophy (as evaluated from measurements of pelagic respiration and primary production). This indicates that the $CO_2$ emissions from the river network were sustained by lateral inputs of $CO_2$ (either from *terra firme* or from wetlands). The $pCO_2$ and $CH_4$ values decreased and $\%O_2$ increased with increasing

Strahler order, showing that stream size explains part of the spatial variability of these quantities. In addition, several lines of evidence indicate that lateral inputs of carbon from wetlands (flooded forest and aquatic macrophytes) were of paramount importance in sustaining high $CO_2$ and $CH_4$ concentrations in the Congo river network, as well as driving spatial variations: the rivers draining the "Cuvette Centrale

Congolaise" (CCC) (a giant wetland of flooded forest in the core of the Congo basin) were characterized by significantly higher $pCO_2$ and $CH_4$ and significantly lower $\%O_2$ and $\%N_2O$ values than those not draining the CCC; $pCO_2$ and $\%O_2$ values were correlated to the coverage of flooded forest on the catchment. The flux of GHGs between rivers and the atmosphere averaged 2,469 mmol m$^{-2}$ d$^{-1}$ for $CO_2$ (range 86

and 7,110 mmol m$^{-2}$ d$^{-1}$), 12,553 µmol m$^{-2}$ d$^{-1}$ for $CH_4$ (range 65 and 597,260 µmol m$^{-2}$ d$^{-1}$), 22 µmol m$^{-2}$ d$^{-1}$ for $N_2O$ (range -52 and 319 µmol m$^{-2}$ d$^{-1}$). The estimate of integrated $CO_2$ emission from the Congo River network (251 TgC ($10^{12}$ gC) yr$^{-1}$) corresponded to nearly half the $CO_2$ emissions from tropical oceans globally (565 TgC yr$^{-1}$) and was nearly two times the $CO_2$ emissions from the tropical Atlantic

Ocean (137 TgC yr$^{-1}$). Moreover, the integrated $CO_2$ emission from the Congo River network is more than three times higher than the estimate of terrestrial net ecosystem exchange (NEE) on the whole catchment (77 TgC yr$^{-1}$). This shows that it is unlikely that the $CO_2$ emissions from the river network were sustained by the hydrological carbon export from *terra firme* soils (typically very small compared to

terrestrial NEE), but most likely, to a large extent, they were sustained by wetlands (with a much higher hydrological connectivity with rivers and streams).

**Keywords:** Africa; Congo River, carbon dioxide, methane, nitrous oxide, tropical wetlands; tropical rivers



## 1. Introduction

Emissions to the atmosphere of greenhouse gases (GHGs) such as carbon dioxide ($CO_2$), methane ($CH_4$) and nitrous oxide ($N_2O$) from inland waters (rivers, lakes and reservoirs) might be quantitatively important for global budgets (Seitzinger and Kroeze 1998; Cole et al. 2007; Bastviken et al. 2011). Yet, there are very large uncertainties in the estimates of GHGs emission to the atmosphere from rivers, as reflected in the wide range of reported values, between 0.2 and 1.8 PgC ($10^{15}$ gC) yr$^{-1}$ for $CO_2$ (Cole et al. 2007; Raymond et al. 2013), 2 and 27 Tg$CH_4$ ($10^{12}$ g$CH_4$) yr$^{-1}$ for $CH_4$ (Bastviken et al. 2011; Stanley et al. 2016), 32 and 2100 Gg$N_2$O-N ($10^{9}$ g$N_2$O-N) yr$^{-1}$ for $N_2O$ (Kroeze et al. 2010; Hu et al. 2016). This uncertainty is mainly due to the scarcity of data in the tropics that account for the majority of riverine GHG emissions (~80% for $CO_2$ (Raymond et al. 2013; Borges et al. 2015a,b; Lauerwald et al. 2015), 79% for $N_2O$ (Hu et al. 2016), 70% for $CH_4$ (Sawakushi et al. 2014)), but also to scaling procedures of varying complexity that use different input data (Raymond et al. 2013; Borges et al. 2015b; Lauerwald et al. 2015), in addition to uncertainty on the estimate of surface area of rivers (Downing et al. 2012; Raymond et al. 2013; Allen and Pavelsky 2018), and the parameterisation of the gas transfer velocity ($k$) (Raymond et al. 2012; Huotari et al. 20013; Maurice et al. 2017; Kokic et al. 2018; McDowell and Johnson 2018). The exchange of $CO_2$ between rivers and the atmosphere is in most cases computed from the air-water gradient of the $CO_2$ concentration and $k$ parameterised as function of stream morphology (e.g. slope or depth) and water flow (or discharge). However, there can be large errors associated to the computation of the dissolved $CO_2$ concentration from pH and total alkalinity (TA) in particular for low alkalinity waters, so that high-quality direct measurements of dissolved $CO_2$ concentration are preferred (Abril et al. 2015), although scarce. In the tropics, research on GHGs in rivers has mainly focussed on South American rivers, and in particular on the Central Amazon (Richey et al. 1988; 2002; Melack et al. 2004; Abril et al. 2014; Barbosa et al. 2016; Scofield et al. 2016), while until recently African rivers were nearly uncharted with a few exceptions (Koné et al. 2009; 2010).

There is also a lack of understanding of the drivers of the fluvial concentrations of GHGs, hence, ultimately of the drivers of their exchange with the atmosphere. It is unclear to which extent $CO_2$ emissions from rivers are sustained by in-situ net heterotrophy and/or by lateral inputs of $CO_2$. The global organic carbon degradation





by net heterotrophy of rivers and streams given by Battin et al. (2008) of 0.2 PgC yr$^{-1}$ is insufficient to sustain global riverine $CO_2$ emissions given by the most recent estimates of 0.7-1.8 PgC yr$^{-1}$ (Raymond et al. 2013; Lauerwald et al. 2015), suggesting an important role of lateral $CO_2$ inputs in sustaining emissions to the

atmosphere from rivers. In a regional study in the US, Hotchkiss et al. (2015) estimated that in-stream organic matter degradation could only sustain 14% and 39% of $CO_2$ riverine emissions in small and large systems, respectively. It is also unclear to which extent $CO_2$ and $CH_4$ emissions from rivers are sustained by carbon inputs either from the terrestrial biome (*terra firme*) or from wetlands (flooded forests and

macrophytes) (Abril and Borges 2019). This difference has large implications for our fundamental understanding of carbon cycling in rivers and their connectivity with respective catchments, but also, consequently, for our capacity to predict how GHG emissions from rivers might be modified in response to climate change (warming and modification of the hydrological cycle), water diversion (damming, water abstraction)

or land use change on the catchment (e.g. Klaus et al. 2018). In upland areas, low order stream $CO_2$ emissions are undoubtedly related to soil-water and ground-water $CO_2$ inputs, although $CO_2$ degassing takes place over very short distances from point sources (≤200 m), is highly variable in space and seasonally, and mainly occurs during short-lived high-flow events that promote shallow flowpaths (e.g. Duvert et al.

2018). Low order streams (1-3) might account for about 1/3 of the global riverine $CO_2$ emissions (Marx et al. 2017). Recently acquired data-sets allowed to show that inputs from riparian wetlands seem to be of paramount importance in sustaining $CO_2$ and $CH_4$ emissions to the atmosphere from large tropical lowland rivers (Abril et al. 2014; Borges et al. 2015a,b). About half of the global surface area of wetlands is

located in the tropics and sub-tropics and about 40% is located in Northern temperate areas (Lehner and Döll 2004), whereas only about one quarter of global river surface area is located in temperate areas, while more than half of river surface area is located in the tropics and sub-tropics (Allen and Pavelsky 2018).

The Congo River (~4400 km length, freshwater discharge ~44,000 m$^3$ s$^{-1}$) has

a large drainage basin (3.7 10$^6$ km²) covered by evergreen forest (dense and mosaic) (~67%) and savannah (shrubland and grassland) (~30%), owing to the tropical climate (annual average precipitation of 1530 mm and air temperature of 23.7°C). The Congo basin accounts for 89% of African rainforests. These rainforests are spread between the Democratic Republic of Congo (54%), Gabon (11%), Cameroon



(10%) and the Republic of Congo (10%), the remaining 15% being shared by several other countries. The mean above-ground biomass of the rainforests in Central Africa (43 kg dry biomass (db) m$^{-2}$) is much higher than the mean in Amazonia (29 kg$_{db}$ m$^{-2}$) and nearly equals the mean in the notorious high biomass forests of Borneo (44 kg$_{db}$

m$^{-2}$) (Malhi et al. 2013). Current estimates of carbon transport from the Congo River close to the mouth rank it as the Earth's second largest supplier of organic carbon to the oceans (Coynel et al. 2005). Despite its overwhelming importance, our knowledge of carbon and nutrient cycling in the Congo river basin is limited to some transport flux data from the 1980's, reviewed by Laraque et al. (2009) and a number

of more recent small-scale studies (Bouillon et al. 2012; 2014; Spencer et al. 2012; Lambert et al. 2016), in sharp contrast to the extensive and sustained work that has been done on the Amazon river basin (Alsdorf et al. 2016). The Congo basin has a wide range of contrasting tributaries (differing in lithology, soil characteristics, vegetation, rainfall patterns, …), and extensive flooded forests in its central region,

the "Cuvette Centrale Congolaise" (CCC), with an estimated flooded cover of 360,000 km$^2$, for a total surface area of 1,760,000 km$^2$ (Bwangoy et al. 2010). Extensive peat deposits are present beneath the swamp forest of the CCC, that store belowground 31 PgC of organic carbon, a quantity similar to the above-ground carbon stock of the forests of the entire Congo basin (Dargie et al. 2017). The

tributaries partly drain semi-humid catchments with alternating dry and wet seasons on both sides of the Equator, resulting in a relatively constant discharge and water level for the mainstem Congo River (Runge 2008). Hence, the CCC is an extended zone of year-round inundation (Bwangoy et al. 2010), in sharp contrast with other large tropical rivers such as the Amazon where floodplain inundation shows clear

seasonality (Hamilton et al. 2002).

Data on dissolved GHGs (CH$_4$, N$_2$O and CO$_2$) have been reported in rivers in the western part of the Congo basin in four major river basins in the Republic of Congo (Alima, Lefini, Sangha, Likouala-Mossaka) (Mann et al. 2014; Upstill-Goddard et al. 2017), and we previously reported GHGs data collected in the eastern part of

the basin in the Democratic Republic of Congo in the framework of a broad synthesis of riverine GHGs at the scale of the African continent (Borges et al. 2015a) and of a general comparison of the Congo and the Amazon Rivers (Borges et al. 2015b). Here, we describe in more detail the variability of GHGs based on a data-set collected during 10 field expeditions from 2010 to 2015 (Fig. 1 and 2), in particular





with regards to spatial and seasonal patterns, as well as with regards to the origin of fluvial $CO_2$ by integrating metabolic measurements (primary production and respiration), stable isotope ratios of dissolved inorganic carbon (DIC), and characteristics of the catchments with regards to the cover of wetlands. Comparison

of data obtained in streams within and outside the giant wetland area of the CCC allows a natural large-scale test of the influence of the connectivity of wetlands on $CO_2$ and $CH_4$ dynamics in lowland tropical rivers.

### 2. Material and Methods

#### 2.1.   Field expeditions and fixed monitoring

Samples were collected during a total of ten field expeditions (Figs. 1 and 2). Three were done from a medium sized boat (22 m long) on which we deployed the
equipment for continuous measurements of the partial pressure of $CO_2$ ($pCO_2$) (total $n$=30,490) as well as the field laboratory for conditioning water samples. Sampling in the mainstem and large tributaries was made from the medium sized boat, while sampling in smaller tributaries was made with a pirogue. These large-scale field expeditions covered twice the Kisangani-Kinshasa transect (03/12/13-19/12/13;
10/06/14-30/06/14), and the Kwa river up to Ilebo (16/04/15-06/05/15). During the other cruises, the field laboratory was installed in a base camp (typically in a village along the river), and travelling and sampling was made with small pirogues on which it was not possible to deploy the apparatus for continuous measurements of $pCO_2$. Three cruises covered the section from Kisangani to the mouth of the Lomami River
(20/11/12-08/12/12; 17/09/13-26/09/13; 13/03/14-21/03/14), one cruise the section from Kisangani to the mouth of the Itimbiri River (08/05/10-06/06/10), and three cruises (previously reported by Bouillon et al. (2012; 2014)) covered the Oubangui river network around the city of Bangui (21-23/03/10; 20-23/03/11; 20-24/11/12).

Fixed sampling was carried out in the Congo mainstem at proximity of the city
of Kisangani (10/12/2012-16/04/2018), the Oubangui mainstem at proximity of the city of Bangui (20/03/2010-31/03/2012), the Kasaï mainstem at proximity of the village of Dima close to the city of Bandundu (14/04/2015-15/05/2017). Sampling was carried out at regular intervals, every 15 days in the Congo and Oubangui



mainstems, and every month in the Kasaï mainstem. Data from the Oubangui catchment were previously reported by Bouillon et al. (2012; 2014).

### 2.2.    Water sampling, direct field measurements, and incubations

Continuous measurements (1 min interval) of $pCO_2$ were made with an equilibrator designed for turbid waters (Frankignoulle et al. 2001) coupled to a non-dispersive infra-red gas analyser (IRGA) (Li-Cor 840) in parallel to water temperature, specific conductivity, pH, dissolved oxygen saturation level (%$O_2$),

turbidity with an YSI multi-parameter probe (6600), and position with a Garmin geographical position system (Map 60S) portable probe. Water was pumped to the equilibrator and the multi-parameter probe (on deck) with a 12V-powered water pump (LVM105) attached to the side of the boat at about 1 m depth. Instrumentation was powered by 12V batteries that were re-charged in the evening with power generators.

In smaller streams, sampling was done from the side of a pirogue with a 1.7L Niskin bottle (General Oceanics) for gases ($CO_2$, $CH_4$, $N_2O$) and a 5L polyethylene water container for other variables. Water temperature, specific conductivity, pH, and %$O_2$ were measured in-situ with a YSI multi-parameter probe (ProPlus). $pCO_2$ was measured with a Li-Cor Li-840 IRGA based on the headspace technique with 4

polypropylene syringes (Abril et al. 2015). During two cruises (20/11/12-08/12/12; 17/09/13-26/09/13, $n$=38), a PP Systems EGM-4 was used as an IRGA instead of the Li-Cor Li-840. During one of the cruises (13-21/03/2014, $n$=20) the equilibrated headspace was stored in pre-evacuated 12 ml Exetainer (Labco) vials and analysed in the home laboratory with a gas chromatograph (GC) (see below). Similarly, the

equilibrated headspace was stored in pre-evacuated 12 ml Exetainer (Labco) vials for $pCO_2$ analysis from the fixed sampling in Kisangani. This approach was preferred to the analysis of $pCO_2$ from the samples for $CH_4$ and $N_2O$ analysis, as the addition of $HgCl_2$ to preserve the water sample from biological alteration led to an artificial increase of $CO_2$ concentrations most probably related to the precipitation of $HgCO_3$

(Fig. S1). Both YSI multi-parameter probes were calibrated according to manufacturer's specifications, in air for %$O_2$ and with standard solutions for other variables: commercial pH buffers (4.00 and 7.00), a 1000 µS cm$^{-1}$ standard for conductivity and a 124 nephelometric turbidity unit (NTU) standard for turbidity. The turbidity data from





the YSI 6600 compared satisfactorily with discrete total suspended matter (TSM) measurements (Fig. S2), so hereafter we'll refer to TSM for both discrete samples and sensor data. The Li-Cor 840 IRGAs were calibrated before and after each cruise with ultrapure $N_2$ and a suite of gas standards (Air Liquide Belgium) with $CO_2$ mixing

ratios of 388, 813, 3788 and 8300 ppm. The overall precision of $pCO_2$ measurements was ±2.0%.

Primary production (PP) was measured during 2h incubations along a gradient of light intensity using $^{13}C\text{-}HCO_3^-$ as a tracer, as described in detail by Descy et al. (2017). Data were integrated vertically with PAR profiles made with a Li-Cor Li-193

underwater spherical sensor, and at daily scale with surface PAR data measured during the cruises with a Li-190R quantum sensor. In order to extend the number of PP data points we developed a very simple model as function of Chlorophyll-*a* concentration (Chl-*a*) and of Secchi depth ($S_d$):

PP = -4.4 + 0.166 x $S_d$ + 3.751 x Chl-*a*

where PP is in mmol $m^{-2}$ $d^{-1}$, $S_d$ in cm and Chl-*a* in µg $L^{-1}$

This approach is inspired from empirical models such as the one developed by

Cole and Cloern (1987) that accounts for phytoplankton biomass given by Chl-*a*, light extinction given by the photic depth and daily surface irradiance. We use $S_d$ as a proxy for photic depth, and in order to simplify the computations, we did not include in the model daily surface irradiance, since it is nearly constant year round in our study site close to the Equator. The model predicts satisfactorily the PP (Fig. S3) except at

very low Chl-*a* values at which the model over-estimates PP (due to the $S_d$ term). In order to overcome this, we assumed a zero PP value for Chl-*a* concentrations < 0.3 µg $L^{-1}$.

Pelagic community respiration (CR) was determined from the decrease of $O_2$ in 60 ml biological oxygen demand bottles (Weathon) over ~24 h incubation periods.

The bottles were kept in the dark and close to in-situ temperature in a cool-box filled with in-situ water. The $O_2$ decrease was determined from triplicate measurements at the start and the end of the incubation with an optical $O_2$ probe (YSI ProODO).

For methane oxidation measurements, seven 60 ml borosilicate serum bottles (Weathon) were filled sequentially from the Niskin bottle and immediately sealed with





butyl stoppers and crimped with aluminum caps. The butyl stoppers were cleaned of leachable chemicals by boiling in deionized water during 15 minutes in the home laboratory. The first and the last bottle to be filled were then poisoned with a saturated solution of $HgCl_2$ (100 µl) injected through the butyl stopper with a

polypropylene syringe and a steel needle, corresponding to the initial concentration of the incubation (T0). The other bottles were stored in a cool-box full of in-situ water (to keep samples close to in-situ temperature and in the dark), and were poisoned approximately 12h after T0 (T1), and then approximately every 24h after T0 (T2, T3, T4 and T5). The difference between the two T0 samples was close to the typical

precision of measurements showing that the water was homogeneous within the Niskin bottle with regards to $CH_4$ concentration, and no measurable loss of $CH_4$ occurred when filling the seven vials. Methane oxidation was computed from the linear decrease of $CH_4$ concentration with time.

## 2.3.    Sample conditioning and laboratory analysis

Samples for $CH_4$ and $N_2O$ were collected from the Niskin bottle with a silicone tube in 60 ml borosilicate serum bottles (Weathon), poisoned with 200 µl of a saturated solution of $HgCl_2$ and sealed with a butyl stopper and crimped with

aluminium cap. Measurements were made with the headspace technique (Weiss 1981) and a GC (SRI 8610C) with a flame ionisation detector for $CH_4$ (with a methanizer for $CO_2$) and electron capture detector for $N_2O$ calibrated with $CO_2$:$CH_4$:$N_2O$:$N_2$ gas mixtures (Air Liquide Belgium) with mixing ratios of 1, 10 and 30 ppm for $CH_4$, 404, 1018, 3961 ppm for $CO_2$, and 0.2, 2.0 and 6.0 ppm for $N_2O$.

The precision of measurement based on duplicate samples was ±3.9% for $CH_4$ and ±3.2% for $N_2O$. The $CO_2$ concentration is expressed as partial pressure in parts per million (ppm) and $CH_4$ as dissolved concentration (nmol $L^{-1}$), in accordance with convention in existing topical literature, and because both quantities were systematically and distinctly above saturation level (400 ppm and 2-3 nmol $L^{-1}$,

respectively). Variations of $N_2O$ were modest and concentrations fluctuated around atmospheric equilibrium, so data are presented as percent of saturation level (%$N_2O$), computed from the global mean $N_2O$ air mixing ratios given by the Global Monitoring Division (GMD) of the Earth System Research Laboratory (ESRL) of the National      Oceanic      and      Atmospheric      Administration      (NOAA)



(https://www.esrl.noaa.gov/gmd/hats/combined/N2O.html), and using the Henry's constant given by Weiss and Price (1982).

The flux ($F$) of $CO_2$ ($FCO_2$), $CH_4$ ($FCH_4$) and $N_2O$ ($FN_2O$) between surface waters and the atmosphere was computed according to Liss and Slater (1974):

$$F = k.\Delta G$$

where $k$ is the gas transfer velocity (cm h$^{-1}$) and $\Delta G$ is the air-water gradient of a given gas.

Atmospheric $pCO_2$ data from Mount Kenya station (NOAA ESRL GMD) and a constant atmospheric $CH_4$ partial pressure of 2 ppm were used. Atmospheric mixing ratios given in dry air were converted to water vapor saturated air, using the water vapor formulation of Weiss and Price (1982) as function of salinity and water temperature.

During the June 2014 field expedition, samples for the stable isotope composition of $CH_4$ ($\delta^{13}C$-$CH_4$) were collected and preserved similarly as described above for the $CH_4$ concentration. The $\delta^{13}C$-$CH_4$ was determined with a custom developed interface, whereby a 20 ml He headspace was first created, and $CH_4$ was flushed out through a double-hole needle, non-$CH_4$ volatile organic compounds were

trapped in liquid $N_2$, $CO_2$ was removed with a soda lime trap, $H_2O$ was removed with a magnesium perchlorate trap, and the $CH_4$ was quantitatively oxidized to $CO_2$ in an online combustion column similar to that of an elemental analyzer. The resulting $CO_2$ was subsequently pre-concentrated by immersion of a stainless steel loop in liquid $N_2$, passed through a micropacked GC column (Restek HayeSep Q 2m length,

0.75mm internal diameter), and finally measured on a Thermo DeltaV Advantage isotope ratio mass spectrometer (IRMS). Calibration was performed with $CO_2$ generated from certified reference standards (IAEA-CO-1 or NBS-19, and LSVEC) and injected in the line after the $CO_2$ trap. Reproducibility of measurement based on duplicate injections of samples was typically better than ±0.5 ‰.

The fraction of $CH_4$ removed by methane oxidation ($F$ox) was calculated with a closed-system Rayleigh fractionation model (Liptay et al. 1998) according to:

$$\ln(1 - Fox) = [\ln(\delta^{13}C\text{-}CH_{4\_initial} + 1000) – \ln(\delta^{13}C\text{-}CH_4 + 1000)] / [\alpha - 1]$$



where $\delta^{13}$C-CH$_{4\_initial}$ is the signature of dissolved CH$_4$ as produced by methanogenesis in sediments, $\delta^{13}$C-CH$_4$ is the signature of dissolved CH$_4$ in-situ, and α is the fractionation factor.

We used a value of 1.02 for α based on field measurements in a tropical lake

(Morana et al. 2015). For $\delta^{13}$C-CH$_{4\_initial}$, we used a value of -60.2‰ which we measured in bubbles from the sediment trapped with a funnel in one occasion in a river dominated by *Vossia cuspidata* wetlands (16/04/15-06/05/15).

Samples for the stable isotope composition of DIC ($\delta^{13}$C-DIC) were collected from the Niskin bottle with a silicone tube in 12 ml Exetainer vials (Labco) and

poisoned with 50 μL of a saturated solution of HgCl$_2$. Prior to the analysis of $\delta^{13}$C-DIC, a 2 ml helium headspace was created and 100 μL of phosphoric acid (H$_3$PO$_4$, 99%) was added in the vial in order to convert CO$_3^{2-}$ and HCO$_3^-$ to CO$_2$. After overnight equilibration, up to 1 mL of the headspace was injected with a gastight syringe into a coupled elemental analyser - IRMS (EA-IRMS, Thermo FlashHT or

Carlo Erba EA1110 with DeltaV Advantage). The obtained data were corrected for isotopic equilibration between dissolved and gaseous CO$_2$ as described by Gillikin and Bouillon (2007). Calibration was performed with certified standards (NBS-19 or IAEA-CO-1, and LSVEC). Reproducibility of measurement based on duplicate injections of samples was typically better than ±0.2 ‰.

Water was filtered on Whatman glass fiber filters (GF/F grade, 0.7 μm porosity) for TSM (47 mm diameter), particulate organic carbon (POC) and particulate nitrogen (PN) (25 mm diameter) (precombusted at 450°C for 5h) and Chl-*a* (47 mm diameter). Filters for TSM and POC were stored dry and filters for Chl-*a* were stored frozen at -20°C. Filters for POC analysis were decarbonated with HCl

fumes for 4h and dried before encapsulation into silver cups; POC and PN concentration and carbon stable isotope composition ($\delta^{13}$C-POC) were analysed on an EA-IRMS (Thermo FlashHT with DeltaV Advantage), with a reproducibility better than ±0.2 ‰ for stable isotopic composition and better than ±5% for bulk concentration of POC and PN. Data were calibrated with certified (IAEA-600:

caffeine) and in-house standards (leucine and muscle tissue of Pacific tuna) that were previously calibrated versus certified standards. The Chl-*a* samples were analysed by high performance liquid chromatography according to Descy et al. (2005), with a reproducibility of ±0.5% and a detection limit of 0.01 μg L$^{-1}$. Part of the Chl-*a* data were previously reported by Descy et al. (2017).





The water filtered through GF/F Whatman glass fibber filters was collected and further filtered through polyethersulfone syringe encapsulated filters (0.2 μm porosity) for stable isotope composition of O of $H_2O$ ($\delta^{18}O$-$H_2O$), TA, dissolved organic carbon (DOC), major elements ($Na^+$, $Mg^{2+}$, $Ca^{2+}$, $K^+$ and dissolved silicate (Si)), nitrate ($NO_3^-$

), nitrite ($NO_2^-$) and ammonium ($NH_4^+$). Samples for $\delta^{18}O$-$H_2O$ were stored at ambient temperature in polypropylene 8 ml vials and measurements were carried out at the International Atomic Energy Agency (IAEA, Vienna), where water samples were pipetted into 2 ml vials, and measured twice on different laser water isotope analyzers (Los Gatos Research or Picarro). Isotopic values were determined by

averaging isotopic values from the last four out of nine injections, along with memory and drift corrections, with final normalization to the VSMOW/SLAP scales by using 2-point lab standard calibrations, as fully described in Wassenaar et al. (2014) and Coplen and Wassenaar (2015). The long-term uncertainty for standard $\delta^{18}O$ values was ±0.1‰. Samples for TA were stored at ambient temperature in polyethylene 55

ml vials and measurements were carried out by open-cell titration with HCl 0.1 mol L$^{-1}$ according to Gran (1952), and data quality checked with certified reference material obtained from Andrew Dickson (Scripps Institution of Oceanography, University of California, San Diego, USA), with a typical reproducibility better than ±3 μmol kg$^{-1}$. DIC was computed from TA and $pCO_2$ measurements using the carbonic acid

dissociation constants for freshwater of Millero (1979) using the CO2sys package. Samples to determine DOC were stored at ambient temperature and in the dark in 40 ml brown borosilicate vials with polytetrafluoroethylene (PTFE) coated septa and poisoned with 50 μL of $H_3PO_4$ (85%), and DOC concentration was determined with a wet oxidation total organic carbon analyzer (IO Analytical Aurora 1030W), with a

typical reproducibility better than ±5%. Part of the DOC data were previously reported by Lambert et al. (2016). Samples for major elements were stored in 20 ml scintillation vials and preserved with 50 μl of $HNO_3$ (65%). Major elements were measured with inductively coupled plasma MS (ICP-MS; Agilent 7700x) calibrated with the following standards: SRM1640a from National Institute of Standards and

Technology, TM-27.3 (lot 0412) and TMRain-04 (lot 0913) from Environment Canada, and SPS-SW2 Batch 130 from Spectrapure Standard. Limit of quantification was 0.5 μmol L$^{-1}$ for $Na^+$, $Mg^{2+}$ and $Ca^{2+}$, 1.0 μmol L$^{-1}$ for $K^+$ and 8 μmol L$^{-1}$ for Si. Samples were collected during four cruises (08/05/10-06/06/10; 20/11/12-08/12/12; 03/12/13-19/12/13; 16/04/15-06/05/15) for $NO_3^-$, $NO_2^-$ and $NH_4^+$ and were stored





frozen (-20°C) in 50 ml polypropylene vials. $NO_3^-$ and $NO_2^-$ were determined with the sulfanilamide colorimetric with the vanadium reduction method (APHA, 1998), and $NH_4^+$ with the dichloroisocyanurate-salicylate-nitroprussiate colorimetric method (SCA, 1981). Detection limits were 0.3, 0.01, and 0.15 µmol $L^{-1}$ for $NH_4^+$, $NO_2^-$ and

$NO_3^-$, respectively. Precisions were ±0.02 µmol $L^{-1}$, ±0.02 µmol $L^{-1}$, and ±0.1 µmol $L^{-1}$ for $NH_4^+$, $NO_2^-$ and $NO_3^-$, respectively.

### 2.4.  Geographical Information System (GIS)

The limits of the river catchments and Strahler order of rivers and streams were    determined    from    the    geospatial    HydroSHEDS    data-set (https://hydrosheds.cr.usgs.gov/) using ArcGIS® (10.3.1). Land cover data were extracted    from    the    global    land    cover    (GLC)    2009    data-set (http://due.esrin.esa.int/page_globcover.php) from the    European Space Agency

GlobCover    2009    project    for    the    following    classes:    Croplands,    Mosaic Cropland/vegetation, Dense forest, Flooded dense forest, Open forest/woodland, Shrublands, Mosaic forest or shrubland/grassland, Grasslands, Flooded grassland, Water bodies. Shrubland and grassland classes were aggregated for the estimate of savannah. Theoretical contribution of $C_4$ vegetation were extracted based on the

vegetation $\delta^{13}C$ isoscape for Africa from Still and Powel (2010) but corrected for agro-ecosystems according to the method of Powell et al. (2012).

The geospatial and statistical methods to compute river width, length, Strahler stream order, surface area, slope, flow velocity, and discharge throughout the Congo River network are described in detail in the Supplemental Information file. The $k$

normalized to a Schmidt number of 600 ($k_{600}$) were derived from the parameterisation as a function slope and stream water velocity given by equation 5 of Raymond et al. (2012).

### 3.  Results and discussion

### 3.1.  Spatial variations

Figures 3 and 4 show the spatial distribution of variables in surface waters of the mainstem Congo River and confluence with tributaries along the Kisangani-



Kinshasa transect during high water (HW, December 2013) and falling water (FW, June 2014) periods (Fig. 1 and 2). Numerous variables show a regular pattern in the mainstem (increase or decrease) due to the gradual inputs from tributaries with a different (higher or lower) value than the mainstem. Specific conductivity, TA, $\delta^{13}$C-

DIC, pH, %$O_2$, %$N_2O$, TSM, pH, and $\delta^{18}$O-$H_2O$ decreased, while $pCO_2$ and DOC increased in the mainstem along the Kisangani-Kinshasa transect. Numerous tributaries had black-water characteristics (low conductivity, TA, pH, %$O_2$, TSM and high DOC) while the mainstem had generally white-water characteristics. The black-water tributaries were mainly found in the region of the CCC, while tributaries

upstream or downstream of the CCC had in general more white-water characteristics. The differences between black-waters and white-waters were apparent in the patterns of continuous measurements of $pCO_2$ showing a negative relationships with %$O_2$, TSM, pH and specific conductivity (Fig. S4).

          Specific conductivity in the mainstem Congo River decreased from 48.3 (HW) /

78.9 (FW) µS cm$^{-1}$ in Kisangani to 26.5 (HW) / 32.7 (FW) µS cm$^{-1}$ in Kinshasa (Figs. 3 and 4). This decreasing pattern was related to a gradual dilution with tributary water with lower conductivities, on average 27.6±9.9 (HW) and 31.9±17.8 (FW) µS cm$^{-1}$. The lowest specific conductivity was measured in the Lefini River that is part of the Téké plateau, where rain water infiltrates into deep aquifers across thick sandy

horizons leading to water with a low mineralisation (Laraque et al. 1998). TA in the mainstem decreased from 344 (HW) / 697 (FW) µmol kg$^{-1}$ in Kisangani to 195 (HW) / 269 (FW) µmol kg$^{-1}$ in Kinshasa, with an average in tributaries of 141±119 (HW) and 136±141 (FW) µmol kg$^{-1}$. $\delta^{13}$C-DIC roughly followed the patterns of TA, decreasing in the mainstem from -7.9 (HW) / -13.8 (FW) ‰ in Kisangani to -11.8 (HW) / -17.8 (FW)

‰ in Kinshasa, with an average in tributaries of -21.7±3.2 (HW) and -20.9±4.3 (FW) ‰. TSM in the mainstem decreased from 92.9 (HW) / 23.2 (FW) mg L$^{-1}$ in Kisangani to 45.4 (HW) / 18.2 (FW) mg L$^{-1}$ in Kinshasa, with an average in tributaries of 9.3±13.4 (HW) / 8.5±9.3 (FW) mg L$^{-1}$. The highest TSM values were recorded in the Kwa (44.4 (HW) and 15.8 (FW) mg L$^{-1}$) and in the Nsele, a small stream draining

savannah and flowing into pool Malebo (71.4 (HW) and 34.8 (FW) mg L$^{-1}$). pH in the mainstem decreased from 6.73 (HW) / 7.38 (FW) in Kisangani to 6.11 (HW) / 6.29 (FW) in Kinshasa, with an average in tributaries of 5.44±0.88 (HW) / 4.91±1.05 (FW). %$O_2$ values in the mainstem decreased from 89.2 (HW) / 92.6 (FW) % in Kisangani to 57.8 (HW) / 79.7 (FW) % in Kinshasa, with an average in tributaries of 36.7±33.3





(HW) / 43.7±35.8 (LW) %. DOC increased from 5.9 (HW) / 5.1 (FW) mg $L^{-1}$ in Kisangani to 11.9 (HW) / 9.4 (FW) mg $L^{-1}$ in Kinshasa, with an average in tributaries of 16.1±10.6 (HW) / 17.9±15.2 (FW) mg $L^{-1}$. Extremely low pH values were recorded in rivers draining the CCC, with values as low as 3.6, coinciding with nearly anoxic

conditions (%$O_2$ down to 0.3 %) in surface waters and very high DOC content (up to 67.8 mg $L^{-1}$). $\delta^{18}O\text{-}H_2O$ in the mainstem decreased from -2.2 (HW) / -1.6 (FW) ‰ in Kisangani to -2.9 (HW) / -2.2 (FW) ‰ in Kinshasa, with an average in the tributaries of -2.7±0.9 (HW) / -2.2±0.8 ‰. Temperature increased in the mainstem from 26.0 (HW) / 27.1 (FW) °C in Kisangani to 27.8 (HW) / 27.3 (FW) °C in Kinshasa, due to

exposure in the uncovered mainstem to solar radiation, as temperature was lower in tributaries with an average of 26.0±1.1 (HW) / 26.1±1.7 (FW) °C due to more shaded conditions (forest cover).

The $pCO_2$ values in the mainstem increased from 2424 (HW) / 1670 (FW) ppm in Kisangani to 5343 (HW) / 2896 (FW) ppm in Kinshasa, with an average in

tributaries of 8306±4089 (HW) / 8039±5311 (FW) ppm. $pCO_2$ in tributaries was in general higher than in the mainstem with a few exceptions, namely in rivers close to Kinshasa (1582 to 1903 (HW) and 1087 to 2483 (FW) ppm), due to degassing at water falls upstream of the sampling stations, as the terrain is more mountainous in this area than in more gently sloping catchments upstream, and also due to a larger

contribution of savannah to the catchment cover. The highest $pCO_2$ values (up to 16,942 ppm) were observed in streams draining the CCC. $CH_4$ in the mainstem decreased from 85 (HW) / 63 (FW) nmol $L^{-1}$ in Kisangani to 24 (HW) / 22 (FW) nmol $L^{-1}$ just before Kinshasa, and then increased again in the Malebo pool (82 (HW) / 78 (FW) nmol $L^{-1}$) related to the shallowness (~3 m in Malebo pool versus ~30 m depth

at station just upstream). The general decreasing pattern of $CH_4$ in the mainstem resulted from $CH_4$ oxidation, as indicated by $^{13}C$ enriched values in the mainstem (-38.1 to -49.4 ‰) with regards to sediment $CH_4$ (-60.2 ‰), and the increasing $^{13}C$ enrichment along the transect (Fig. 5). The calculated fraction of oxidized $CH_4$ ranged between 0.43 and 0.68 in the mainstem and also increased downstream along the

transect (Fig. S5). $CH_4$ in the tributaries showed a very large range of $CH_4$ concentration (68 to 51,839 nmol $L^{-1}$ (HW) and 102 to 56,236 nmol $L^{-1}$ (FW)). $CH_4$ in the tributaries showed a variable degree of $^{13}C$ enrichment compared to sediment $CH_4$ ($\delta^{13}C\text{-}CH_4$ between -19.3 and -56.3 ‰, Fig. 5), and the calculated fraction of oxidized $CH_4$ ranged between 0.18 and 0.88 (Fig. S5). Unlike the large rivers of the



Amazon where a loose negative relation has been reported (Sawakuchi et al. 2016), the $\delta^{13}$C-CH$_4$ values in the Congo River were unrelated to dissolved CH$_4$ concentrations, and a relatively high $^{13}$C enrichment ($\delta^{13}$C-CH$_4$ up to -19.3 ‰) was observed even at high CH$_4$ concentrations (correspondingly 3,118 nmol L$^{-1}$) (Fig. 5).

This lack of correlation between concentration and isotope composition of CH$_4$ was probably related to spatial heterogeneity of sedimentary (and corresponding water column) CH$_4$ content over a very large and heterogeneous sampling area. The highest CH$_4$ concentrations were observed at the mouths of small rivers in the CCC. At the confluence with the Congo mainstem, the flow is slowed down leading to the

development of shallow delta-type systems with very dense coverage of aquatic macrophytes (in majority *Vossia cuspidata* with a variable contribution of *Eichhornia crassipes,* but also other species, Fig. S6). Such sites are favourable for intense sediment methanogenesis, but due to the stable environment related to near stagnant waters, and also very favourable for the establishment of a stable

methanotrophic bacterial community in the water column and associated to the root system of floating macrophytes that sustain intense methane oxidation (Yoshida et al. 2014; Kosten et al. 2016). Indeed, we found a very strong relation between CH$_4$ oxidation and CH$_4$ concentration on a limited number of incubations carried out in the Kwa river network in April 2015 (Fig. S7). Such conditions can explain the apparently

paradoxical combination of high CH$_4$ concentrations in some cases associated to a high degree of CH$_4$ oxidation. Samples with extremely $^{13}$C depleted POC ($\delta^{13}$C-POC down to -39.0 ‰) were observed in low %O$_2$ and high CH$_4$ environments (Fig. 6) located in small streams of the CCC, that were also characterised by low POC and TSM values (not shown), and high POC:Chl-*a* ratios (excluding the possibility that *in-*

*situ* PP would be at the basis of the $^{13}$C-depletion). This suggests that in these environments poor in particles (typical of black-water streams) but with high CH$_4$ concentrations, methanotrophic bacteria (that are typically strongly depleted in $^{13}$C) contribute substantially to POC. While such patterns have been reported at the oxic-anoxic transition zone of lakes with high hypolimnic CH$_4$ concentrations such as Lake

Kivu (Morana et al. 2015), this has never been reported in surface waters of rivers.

%N$_2$O decreased from 198 (HW) / 139 (FW) % in Kisangani to 168 (HW) / 153 (FW) % in Kinshasa, and in most cases %N$_2$O was lower in tributaries, on average 114±73 (HW) / 120±69 (FW) %. The under-saturation in N$_2$O was observed in streams with high DOC and low %O$_2$, and most probably related to sedimentary



denitrification of $N_2O$ as also reported in the Amazon river (Richey et al. 1988), and in temperate rivers (Baulch et al. 2011). Indeed, there was a general negative relationship between $\%N_2O$ and $\%O_2$ (Fig. 7), with an average $\%N_2O$ of 78.5±59.3% for $\%O_2 < 25\%$ and an average $\%N_2O$ of 155.7±57.7% for $\%O_2 > 25\%$ (Mann-

Whitney $p$<0.0001). The decreasing pattern of $NH_4^+$:DIN and increasing pattern of $NO_3^-$:DIN with $\%O_2$ indicated the occurrence of nitrification in oxygenated (typical of high Strahler stream order) rivers and prevalence of $NH_4^+$ in the more reducing and lower oxygenated (typical of low Strahler order) streams in particular draining the CCC, where $NO_3^-$ was probably also removed from the water by sedimentary

denitrification (Fig. 7). We hypothesize that $N_2O$ was removed by sedimentary denitrification in low oxygenated streams while in more oxygenated streams and rivers, $N_2O$ was produced by nitrification. This is consistent with the positive relation between $\%N_2O$ and $NO_3^-$:DIN and negative relation between $\%N_2O$ and $NH_4^+$:DIN (Fig. 8) In the mainstem of the Congo, there might be in addition a loop of nitrogen

recycling (ammonification-nitrification) sustained by phytoplankton growth and decay, that contributed to maintain over-saturation of $N_2O$ with respect to atmospheric equilibrium, as phytoplankton growth was only observed in the mainstem (Descy et al. 2017). The generally low $\%N_2O$ values in the Congo River network are due to the near pristine nature of these systems with low $NH_4^+$ (2.3±1.3 µmol $L^{-1}$) and $NO_3^-$

(5.6±5.1 µmol $L^{-1}$) levels, typical of rivers and streams draining a large fraction (~70%) of forests. Croplands only represented at most 17% on average of the land cover of the studied river catchments, where traditional agriculture is practised with little use of artificial fertilizers. This corresponds to an upper bound of cropland surface area since it was estimated aggregating the "cropland" and "mosaic

cropland/vegetation" GLC 2009 categories, the latter corresponding to mixed surfaces with <50% of cropland. The "cropland" GLC 2009 category only accounts for 0.1% on average of the land cover of the studied river catchments. The largest cities along the Congo River mainstem are of modest size such as Kisangani (1,600,000 habitants) and Mbandaka (350,000 habitants), especially considering the large

dilution due to the massive discharge of the mainstem (sampling was done upstream of the influence of the megacity of Kinshasa with 11,900,000 habitants).

       The input of the Kwa led to distinct changes of TA and $\delta^{18}O$-$H_2O$ (decrease) and TSM (increase) of mainstem values (comparing values upstream and downstream of the Kwa mouth) (Figs 3-4). In the mainstem, continuous



measurements of $pCO_2$, pH, %$O_2$, and conductivity showed more variability compared to discrete samples acquired in the middle of channel, in particular in the region of CCC (Figs 3-4). These patterns were related to gradients across the section of channel, as the boat sailed either along the mid-channel or closer to shore. The

water from the tributaries flowed along the river banks and did not mix with mainstem middle channel waters, as visible in natural colour remote sensing images (Fig. S8), leading to strong gradients across the section of mainstem channel. During the June 2014 field expedition, this was investigated in more detail by a series of six transects perpendicular to the river mainstem channel (Fig. 9). In the upper part (1590 km from

Kinshasa), the variables showed little cross-section gradients except for a decrease of conductivity towards the right bank due to inputs from the Lindi River that had distinctly lower specific conductivities (27.2 (HW) / 35.1 (FW) µS cm$^{-1}$) than the main stem (48.3 (HW) / 78.9 (FW) µS cm$^{-1}$). At 795 km from Kinshasa, marked gradients appeared in all variables with the presence of black-water characteristics close to the

right bank (higher $pCO_2$, and lower %$O_2$, pH, specific conductivity, temperature and TSM values). This feature was related to inputs from large right bank tributaries such as the Aruwimi and the Itimbiri (Table S1). Upstream of this section the only major left-bank tributary is the Lomami that had white-water characteristics, relatively similar to those of the mainstem (Figs. 3-4). The presence of black-water

characteristics became apparent also on left-bank in cross-sections at 307 and 254 km upstream of Kinshasa, where the river is particularly wide (> 6 km wide), and received the inputs from the Ruki, the second largest left-bank tributary (Table S1) with black-water characteristics. The cross section gradients became less marked at 203 km upstream from Kinshasa, as in this region the river becomes more narrow (2

km wide) leading to increased currents and more lateral mixing. The cross-section gradients nearly disappeared at 158 km upstream of Kinshasa corresponding to 30 km downstream of the Kwa mouth due to homogenisation by the large Kwa inputs (nearly 20% of total freshwater discharge from Congo River, Table S1) into a relatively narrow river section (~ 1 km).

Spatial features of biogeochemical variables along the transect in the Kwa River network (Fig. 10) showed some similarities with two Kisangani-Kinshasa transects along the Congo mainstem (Figs 3-4). The mainstem Kwa had a higher specific conductivity than tributaries (25.1±4.2 versus 21.2±11.5 µS cm$^{-1}$), higher TA (281±64 vs. 119±118 µmol kg$^{-1}$), higher temperatures (27.7±0.7 versus 26.3±2.2 °C),



higher TSM (40.1±8.9 versus 15.2±35.1 mg L$^{-1}$), higher pH (6.1±0.2 versus 4.5±0.7), higher %O$_2$ (67.0±7.0 versus 37.9±26.8 %), higher δ$^{13}$C-DIC (-16.5±1.2 versus -22.6±3.4 ‰), and lower DOC (5.2±1.4 versus 13.0±8.5 mg L$^{-1}$). Unlike the Kisangani-Kinshasa transects along the Congo mainstem, the δ$^{18}$O-H$_2$O was lower in the

mainstem Kwa (-4.3±0.3 ‰) than in the tributaries (-3.3±0.6 ‰). Note that both the main upstream branches of the Kwa (Kasai and Sankuru) had low δ$^{18}$O-H$_2$O values. The δ$^{18}$O-H$_2$O values of the upper Kasai and Sankuru (-4.4 and -4.7 ‰, respectively) were lower than those of the Lualaba (Congo at Kisangani) (-2.2 (HW) / -1.6 (FW) ‰). This was probably related to the lower evapotranspiration over the catchments of

the upper Kasai and Sankuru than those of Lualaba (Bultot 1972), leading to more $^{18}$O-depleted water (e.g. Simpson and Herczeg 1991). Additionally, the catchment of Kwa has a high fraction of unconsolidated sedimentary (41.4%) and siliciclastic sedimentary (44.3%) rocks than the catchment of the Lualaba that is dominated by metamorphic rocks (68.4%) Unconsolidated sedimentary and siliciclastic sedimentary

rocks are more favourable to the infiltration of water and development of aquifers that will minimize evaporation and $^{18}$O enrichment, unlike catchments dominated by metamorphic rocks. Note that the tributaries with the lowest δ$^{18}$O-H$_2$O values were situated downstream of the Kwa and upstream of Kinshasa (Figs. 3-4) and are part of the Téké plateau. These rivers are fed by deep aquifers derived from infiltration of

rain through sandy soils (Laraque et al. 1998).

Another difference with the Kisangani-Kinshasa transects along the Congo mainstem, relates to N$_2$O values that were closer to saturation in the Kwa mainstem (110.1±8.8 %), while surface waters oscillated from under-saturation to oversaturation in the tributaries (122.4±59.5 %). This might have been related to the

strong flow in the Kwa that led to high gas transfer velocities and strong degassing of N$_2$O to the atmosphere. In the Kwa mainstem, pCO$_2$ (3473±974 ppm) and CH$_4$ (255±150 nmol L$^{-1}$) values were lower than in tributaries (8804±5108 ppm, 6,783±16,479 nmol L$^{-1}$). The highest pCO$_2$ and CH$_4$ values of the entire data-set in the Congo River network were observed in a tributary of the Fimi (22,899 ppm and

71,428 nmol L$^{-1}$) that is bordered by very extensive meadows of the aquatic macrophyte *Vossia cuspidate*, and unrelated to inputs from Lake Mai Ndombé that showed lower pCO$_2$ and CH$_4$ values (3,143 ppm and 250 nmol L$^{-1}$, respectively).

The large differences in pCO$_2$, CH$_4$, %O$_2$ and %N$_2$O (Figs 3,4,10) among the various sampled tributaries of the Congo River can be analysed in terms of size





classes as given by Strahler order (Fig. 11). There were distinct patterns in $CH_4$ and $pCO_2$ versus Strahler order, with a decrease of the central value (median and average) for both quantities as a function of Strahler order in streams draining and not draining the CCC (Fig. 11). For nearly all the stream orders, the streams draining

the CCC had significantly higher $pCO_2$ and $CH_4$ values than streams not draining the CCC. The $\%O_2$ values per Strahler order did not show any distinct pattern (increase or decrease) in the streams not draining the CCC, but the streams draining the CCC showed an increasing pattern as a function of Strahler order. The $\%O_2$ values were significantly lower in the streams draining the CCC than those not draining the CCC

(Fig. 11). For $\%N_2O$, the tendency of the central value (median and average) as function of Strahler order did not show a clear pattern for streams not draining the CCC, however, there was a clear increasing pattern with Strahler order for streams draining the CCC. In addition, the $\%N_2O$ values were significantly lower for half of the cases, in streams draining the CCC compared to those not draining it. In US rivers, a

decreasing pattern as function of Strahler order has previously been reported for $pCO_2$ (Butman and Raymond 2001; Liu and Raymond 2018), and has been interpreted as reflecting inputs of soil-water enriched in terrestrial respired $CO_2$ that have a stronger impact in smaller and lower Strahler order systems, in particular headwater streams, followed by degassing of $CO_2$ in higher Strahler order rivers

(Hotchkiss et al. 2015), although soil-water $CO_2$ inputs in headwater streams are seasonally variable and spatially heterogeneous (Duvert et al. 2018). Nevertheless, all of the low Strahler order streams we sampled were in lowlands, so, the decreasing pattern of $pCO_2$ as function of Strahler order could alternatively reflect the stronger influence of riparian wetlands on smaller streams, rather than larger

systems. The mechanism remains the same, a high ratio of lateral inputs to water volume in small streams, that is related to soil-water in temperate streams such as in the US but related in addition to riparian wetlands in tropical systems such as those sampled in the Congo River network.

        The influence of riparian wetlands on stream $pCO_2$, $CH_4$ and $\%N_2O$ can be

also highlighted when data were separated into rivers draining or not draining the CCC but aggregating into systems smaller or larger than Strahler order 5 to account simultaneously for the effect of stream size (Fig.12). The $pCO_2$ values were statistically higher in rivers draining the CCC than those not draining it, with median values more than 2 fold higher in both small and large rivers. Conversely, $\%O_2$ levels



were statistically lower in rivers draining the CCC than those not draining it, with median values 11 and 2 fold lower in small and large rivers, respectively. Additional evidence on the influence of the connectivity of wetlands with rivers in sustaining high $pCO_2$ and low $\%O_2$ values was provided by the positive relationship between $pCO_2$

and flooded dense forest cover and the converse negative relationship between $\%O_2$ and flooded dense forest cover (Fig. 13). These patterns were also consistent with the positive relation between DOC concentration and flooded dense forest reported by Lambert et al. (2016). Note that aquatic macrophytes (*Vossia cuspidata*) most probably also strongly contributed in addition to flooded forest to high $pCO_2$ and low

$\%O_2$ levels, based on *de-visu* observations of dense coverage (Fig S9), although macrophytes have not been systematically mapped and GIS data are unavailable (as for flooded dense forest).

        Wetlands coverage had also a major importance on $CH_4$ distribution, as the $CH_4$ values were statistically higher in rivers draining the CCC than those not draining

it (Fig. 12), with median values 10 and 2 fold higher in small and large rivers, respectively. $\%N_2O$ were also statistically lower in rivers draining the CCC than those not draining it, with median values 2.7 fold lower in small rivers but 1.4 fold higher in large rivers. The pattern of $\%N_2O$ followed the one of $\%O_2$, and the very low to nil $N_2O$ values were observed in the systems draining the CCC where the lowest $\%O_2$

values (close to 0) were also observed due to sedimentary organic matter degradation leading to a decrease of $O_2$ and $N_2O$ in surface waters (removal of $N_2O$ by denitrification).

        The $CH_4:CO_2$ molar ratio ranged between 0.0001 and 0.1215, with a mean of 0.0097±0.018. Such ratios were distinctly higher than those typically observed in

marine waters (0.0005) and in the atmosphere (0.005). The $CH_4:CO_2$ molar ratio strongly increased with the decrease of $\%O_2$ and was significantly higher in small rivers draining the CCC (Fig. 13). These patterns were probably related to inputs of organic matter from wetlands and in particular aquatic macrophytes that lead to important organic matter transfer to sediments and high sedimentary degradation of

organic matter. This lead to $\%O_2$ decrease in surface waters and a large fraction of organic matter degradation by anaerobic processes compared to aerobic degradation, leading to an increase of the $CH_4:CO_2$ ratio. The decrease of $O_2$ and increase of $CO_2$ in the water in presence of floating macrophytes was probably in part





also related to autotrophic root respiration and not fully related to microbial heterotrophic respiration.

### 3.2. Drivers of $CO_2$ dynamics

We compared the balance of depth-integrated planktonic PP to water column CR and compared it to $FCO_2$ to test if *in-situ* net heterotrophy was sufficient to sustain the emissions of $CO_2$ to the atmosphere (Fig. 15). A detailed description of spatial and seasonal variations of PP as well as main phytoplankton communities is

given by Descy et al. (2017). In brief, phytoplankton biomass was mainly confined to the mainstem and was low in most tributaries. The PP values in the Congo mainstem were higher than previously reported in tropical river channels, whereas in other tropical rivers phytoplankton production mainly occurs in the floodplain lakes. This is due to generally lower TSM values in the Congo and to its relative shallowness that

allows net phytoplankton growth in the mainstream unlike other deeper and more turbid tropical rivers such as the Amazon. Measured PP was 3 out 49 times lower than CR, and on average the PP:CR ratio was 0.28. The same applied when using modelled PP, to extend the number of data points, with PP 2 out 169 times lower than CR, and on average a PP:CR ratio of 0.15. This indicates that a generalised and

strongly net heterotrophic metabolism was encountered in the sampled sites. Yet, in 174 out 187 cases, $FCO_2$ was higher than CR and in 162 out 169 cases, $FCO_2$ was higher than net community production (NCP). CR averaged 81 mmol m$^{-2}$ d$^{-1}$, and NCP averaged -75 mmol m$^{-2}$ d$^{-1}$ and the corresponding average $FCO_2$ was 740 mmol m$^{-2}$ d$^{-1}$. CR was estimated from measurements of $O_2$ concentration decrease in

bottles than were not rotated, and this has been shown to lead to an under-estimation of CR up to a factor of two in incubations lasting several days (Richardson et al. 2013; Ward et al. 2018). The under-estimation of our CR measurements due to the absence of rotation is most likely not as severe as in the Richardson et al. (2013) and Ward et al. (2018) studies, as our incubations were shorter (24h) and the organic

matter in our samples was mostly in dissolved form (median DOC of 8.6 mg L$^{-1}$), with a low particulate load (median TSM of 14 mg L$^{-1}$ and POC of 1.3 mg L$^{-1}$). Nevertheless, it seems unrealistic to envisage an under-estimation of CR by an order of magnitude that would allow reconciling the CR (and NCP) estimates with those of $FCO_2$. Although we did not measure sediment respiration, the average value reported





by Cardoso et al. (2014) of 21 mmol $m^{-2}$ $d^{-1}$ for tropical rivers and streams does not allow accounting for the imbalance between $F$CO$_2$ and NCP. This then suggests that the emission of CO$_2$ from the Congo lowland river network is to a large extent sustained by lateral inputs rather than by in-stream production of CO$_2$ by net

heterotrophy. It remains to be determined to which extent this lateral input of CO$_2$ is sustained by riparian wetlands or soil-groundwater from *terra firme*.

The low PP:CR ratio of 0.15 to 0.28 on average, and generally low PP values (on average 12 mmol $m^{-2}$ $d^{-1}$) were also reflected in the low diurnal variations of pCO$_2$. We did not carry out dedicated 24 h cycles to look at the day-night variability of

pCO$_2$, due to lack of opportunity given the important navigation time to cover large distances, but we compared the data acquired at the anchoring site on shore (typically around 17h00 universal time (UT), just before dusk) with the data on the same spot the next day (typically around 04h30 UT, just after dawn) (Fig. S10). Unsurprisingly, water temperature measured just before dusk was significantly higher

than just after dawn (on average 0.5°C higher), while specific conductivity was not significantly different, indicating that the same water mass was sampled at dusk and dawn. The pCO$_2$ and O$_2$ concentration measured just before dusk were not significantly different than just after dawn, showing that daily variability in these variables was low. The difference between pCO$_2$ at dusk and dawn ranged between -

2307 and 1186 ppm, and averaged 39 ppm (*n*=39). The wide range of values of the difference of pCO$_2$ at dusk and at dawn might reflect occasional small scale variability of pCO$_2$, as the boat anchored for the night close to shore, and frequently at proximity of riparian vegetation. Nevertheless, the average difference is not of the expected sign (in the case of a strong diurnal change of pCO$_2$ due to PP and CR,

pCO$_2$ should have been lower at dusk than dawn, so difference should have been negative). This difference was also very small compared to the overall range of spatial variations of pCO$_2$ (1,087 to 22,899 ppm). Day-night variations of pCO$_2$ have been reported in temperate headwater and low order streams and in one lowland river with an amplitude from ~50 to ~700 ppm (Lynch et al. 2010; Dinsmore et al.

2013; Peter et al. 2014; Crawford et al. 2017; Reiman and Xu 2019), although daily signals of pCO$_2$ were not systematically observed, and were absent for instance in streams covered by forest canopy (Crawford et al. 2017). In a low turbidity and very shallow low order stream of the Tana River network, Tamooh et al. (2013) reported on one occasion day-night variation of pCO$_2$ with an amplitude of ~400 ppm, and in



the Zambezi river, during the dry season corresponding to very low TSM values (<10 mg $L^{-1}$), Teodoru et al. (2015) reported day-night $pCO_2$ variation in the range of 475 ppm. In both cases, the day-night variations were small compared to spatial variations of 300 to 5,204 ppm in the Tana and 300 to 14,004 ppm in the Zambezi. In floodplain lakes of the Amazon, daily variations of $pCO_2$ can be intense (with an amplitude up to ~2,000 ppm) during cyanobacterial blooms (Abril et al. 2013; Amaral et al. 2018), but have not been documented in the river channels of the Amazon. Our data show that in nutrient poor and light limited lowland tropical rivers such as the Congo River, where pelagic PP is low, day-night variations of $pCO_2$ were negligible compared to spatial variations of $pCO_2$. We conclude that accounting for night-day variations of $pCO_2$ should not lead to a dramatic revision of global $CO_2$ emissions, unlike the recently claim based on data from a lowland temperate river by Reiman and Xu (2019), given that tropical rivers account for 80% of $CO_2$ emissions (Raymond et al. 2013; Borges et al. 2015a,b; Lauerwald et al. 2015).

The stable isotope composition of DIC can provide information on the origin of $CO_2$, although the signal depends on the combination of the biological processes that remove or add $CO_2$ to the water column (CR and PP), rock weathering that adds $HCO_3^-$, and outgassing which removes $CO_2$ that is $^{13}C$-depleted relative to $HCO_3^-$. The variable contribution to DIC in time and in space of $CO_2$ relative to $HCO_3^-$ complicates the interpretation of $\delta^{13}C$-DIC data. The stable isotopic composition of DIC due to rock weathering will depend on the type of rock (silicate or carbonate) and on the origin of the $CO_2$ involved in rock dissolution (either $CO_2$ from the atmosphere or from respiration of soil organic matter). The stable isotopic composition of DIC due to the degradation of organic matter will depend in part on whether the organic matter is derived from terrestrial vegetation following the $C_3$ photosynthetic pathway (woody plants and trees, temperate grasses; $\delta^{13}C$ ~ -27‰) compared to the less fractionating $C_4$ photosynthetic pathway (largely tropical and subtropical grasses; $\delta^{13}C$ ~ -13‰) (Hedges et al. 1986; Bird et al. 1994). Spatial and temporal changes of $\delta^{13}C$ are related to the change of the relative abundance of $HCO_3^-$ over $CO_2$. The degassing of $CO_2$ to the atmosphere and the addition of $HCO_3^-$ from rock weathering lead to $\delta^{13}C$-DIC values becoming dominated by those related to $HCO_3^-$. Since $CO_2$ is isotopically depleted relative to $HCO_3^-$, $CO_2$ degassing leads to a gradual enrichment of the remaining DIC pool (e.g. Doctor et al. 2008; Deirmendjian and Abril 2018). The combination of these processes can lead to spatial changes of $\delta^{13}C$-DIC that co-vary





with $CO_2$ and $HCO_3^-$ concentrations. For instance, in the Tana River network, an altitudinal gradient of $\delta^{13}C$-DIC was attributed to a downstream accumulation of $HCO_3^-$ due to rock weathering combined to $CO_2$ degassing (Bouillon et al. 2009; Tamooh et al. 2003). Figure 16 shows $\delta^{13}C$-DIC as a function of the TA:DIC ratio in

the rivers and streams of the Congo, with a general increase of $\delta^{13}C$-DIC from TA:DIC=0 (all of the DIC is in the form of $CO_2$) towards TA:DIC=1 (nearly all of the DIC is in the form of $HCO_3^-$). This pattern reflects the mixing of two distinct types of rivers and streams: the lowland systems draining the CCC with low rock weathering due to dominance of deep organic soils (low $HCO_3^-$) and high $CO_2$ from respiration

leading to TA:DIC close to 0 with low $\delta^{13}C$-DIC, and the systems draining highland regions (Lualaba and Kasaï) with high rock weathering and lower generation of $CO_2$ from respiration and/or higher $CO_2$ degassing leading to TA:DIC close to 1 with high $\delta^{13}C$-DIC. The plots of TA:$Na^+$ and $Mg^{2+}$:$Na^+$ versus $Ca^{2+}$:$Na^+$ (Fig. 17) showed aggregation of data close to what would be expected for silicate rock weathering

based on the average values proposed by Gaillardet et al. (1999). This is in agreement with the dominance of silicate rocks over carbonate rocks in the Congo basin (Nkounkou and Probst 1987). Note that TA values from the Congo are generally very low compared to other large rivers globally (Meybeck 1987), due to the large proportion of relatively insoluble rocks on the catchment (70% of metamorphic

rocks) and a small proportion of low soluble rocks such as siliciclastic sedimentary rocks (10% mainly as sand stone), unconsolidated sediments (17% as sand and clays), and a very small proportion of high soluble volcanic rocks (1%). The high TA values in the Lualaba are due to a larger proportion of volcanic rocks in high altitude areas, such as the Virunga region that is rich in volcanic rocks (including basalts) and

has been shown to be a hotspot of chemical weathering (Balagizi et al. 2015).

At TA:DIC=0, the $\delta^{13}C$-DIC values are exclusively related to those of $CO_2$ and might be indicative of the source of mineralized organic matter. The $\delta^{13}C$-DIC values ranged between -28.2 and -15.9 ‰, and averaged -24.1 ‰ (Fig. 16) indicating that $CO_2$ was produced from the degradation of mixture of organic matter from $C_3$ and $C_4$

origin. Furthermore, $\delta^{13}C$-DIC was positively related to $pCO_2$ (Fig. 16), indicating that in the streams with high $pCO_2$ values, the $CO_2$ was generated from the degradation organic matter with a higher contribution from $C_4$ plants. The $\delta^{13}C$-DIC values were unrelated to the contribution of $C_4$ vegetation on the catchment (*terra firme*), as modelled by Still and Powell (2010), and the cover by savannah on the catchment



given by GLC2009 (Fig. S11). Further, the most enriched $\delta^{13}$C-DIC values (>-21 ‰), corresponded on average to a low contribution of $C_4$ vegetation on the catchment (7.4%), and low contribution of savannah cover on the catchment (0.8%). These patterns are inconsistent with an origin of the $C_4$ material that led to high $pCO_2$ in

streams (Fig. 16) from *terra firme* origin, but is rather more consistent with a larger contribution of degradation of $C_4$ aquatic macrophyte material in high $CO_2$ streams.

### 3.3.    Seasonal variations

The difference between the HW and FW in the mainstem along the Kisangani-Kinshasa transects (Fig. 3 and 4) were relatively modest with higher specific conductivity, TA, $\delta^{13}$C-DIC and TSM (at Kisangani), the biogeochemical signatures of higher surface run-off during the HW sampling (December 2013) and water flows from deeper soil horizons (or ground-water) during the FW sampling (June 2014).

The comparison of tributaries that were sampled during both HW and FW periods along the Kisangani-Kinshasa transects shows that $pCO_2$ was significantly higher (Wilcoxon match-pairs signed rank test p=0.078) during HW for large systems (freshwater discharge ≥ 300 $m^3$ $s^{-1}$, Table S1) but not significantly different for small systems (freshwater discharge < 300 $m^3$ $s^{-1}$), irrespective whether left or right bank

(Fig. 18). However, no significant differences among the HW and FW periods occurred in $\%O_2$, $CH_4$ and $\%N_2O$ for either small of large systems, irrespective whether left or right bank. This indicates that across the basin, spatial differences among tributaries are more important than seasonal variations within a given tributary. The average difference for large rivers between HW and FW was on

average only 1,745 ppm for $pCO_2$ when the respective range of variation for the whole data-set was from 1,087 to 22,899 ppm.

Yearly cycles of $CH_4$ and $N_2O$ were established on the Lualaba (6 yrs), the Oubangui (2yrs) and the Kasaï (2 yrs) (Figs. 19 and 20), while $pCO_2$ is only available during 1 yr in the Lualaba. In the Oubangui and the Kasaï, $CH_4$ concentration peaked

with the onset of rising water and decreased as water level continued to increase. The decrease of $CH_4$ as discharge peaked in the Oubangui and the Kasaï is most probably related to dilution by surface runoff as also testified by the decrease of specific conductivity and TA (not shown). The increase at the onset of rising water could be related to initial flushing of soil atmosphere enriched in $CH_4$ as rain



penetrates superficial layers of soils. In the Lualaba the $CH_4$ seasonal variations seem to follow more closely those of freshwater discharge (for instance in 2014). Unlike the Kasaï and the Oubangui, the Lualaba has very extensive permanent marshes and swamps such as the Upemba wetland system (inundated area of

18,000 $km^2$) and the Luama swamps (inundated area of 6,000 $km^2$) (Hughes and Hughes 1992). Yet, it remains to be determined if the seasonality of $CH_4$ observed in Lualaba can be attributed to higher inputs during high water from these major wetlands located respectively, 1000 and 600 km upstream of Kisangani. Even so, there are numerous more modest marshes and swamps that border upstream

tributaries of the Lualaba closer to Kisangani (Hughes and Hughes 1992). The seasonal amplitude of dissolved $CH_4$ in the Oubangui was about 10 times higher than in the Kasaï and the Oubangui, and the seasonal amplitude of $CH_4$ in these three rivers seemed to be related to the relative seasonal amplitude of freshwater discharge as indicated by the positive relation with the ratio of maximum and

minimum of freshwater discharge (Fig. 21).

The seasonal cycles of $\%N_2O$ show patterns that were consistent with those $CH_4$, with a loose parallelism of $\%N_2O$ and discharge in the Lualaba, an inverse relation in the Kasaï, and peak of $\%N_2O$ during rising waters (although delayed with respect to the $CH_4$ peak). In addition to the seasonal cycle there seemed to be a

longer term decrease of the annual average of $\%N_2O$ in the Lualaba (Fig. S12). There were no significant changes of the annual average of other variables such a POC, PN (not shown). So the observed decrease of annual $\%N_2O$ is probably unrelated to changes in terrestrial productivity (Zhou et al. 2014) or nitrogen content of terrestrial vegetation (Craine et al. 2018) that act at longer time scales (several

decades) and seem to be related to long term changes in climate (precipitation). Freshwater discharge and water temperature show an increasing pattern during the time period (Fig. S12). An increasing discharge could lead to increased gas transfer velocities and a loss of $N_2O$ to the atmosphere. Water temperature in the Congo River is already close to the optimum of denitrification (~27°C, Canion et al. 2014), so

an enhancement of denitrification with increase of temperature is unlikely. Freshwater discharge decreased in 2017 when $\%N_2O$ also seemed to stabilise compared to 2016. It is likely that the decreasing trend in $\%N_2O$ during the 2013-2017 period is a transient feature in response to short term fluctuations in hydrology, as indicated by freshwater discharge.



The $pCO_2$ time-series in the Lualaba at Kisangani is shorter than for $CH_4$, but a general positive relationship between $pCO_2$ and discharge was observed (Fig. S13). A similar positive relationship between $pCO_2$ and discharge was also observed in the Oubangui (Bouillon et al. 2012) and the Madeira River (Almeida et al. 2017), as

well is several rivers in the in the Amazon (Richey et al. 2002) based on $pCO_2$ calculated from pH and TA. Such $pCO_2$-discharge patterns were interpreted as resulting from higher connectivity during high-water between river mainstem and the floodplains and wetlands. In large temperate rivers, there seems to be a general negative relationship between $pCO_2$ and discharge as observed in the Meuse

(Borges et al. 2018) and in more than half of the US rivers analyzed by Liu and Raymond (2018). The difference in the relationship of $pCO_2$ versus discharge between tropical and temperate large rivers might be related to lower interactions between river and floodplains in temperate rivers in particular in highly human-impacted and channelized rivers such as the Meuse. Also, temperature co-varies

strongly with discharge in temperate rivers, so that the warmer months that promote biological production of $CO_2$ are also characterized by lower discharge (Borges et al. 2018).

The seasonal amplitude of $CH_4$ (~50 nmol $L^{-1}$ in the Kasaï and the Congo, versus 200-400 nmol $L^{-1}$ in the Oubangui) and $\%N_2O$ (20-90% in the 3 rivers) was

overall much lower than the spatial gradients across the basin of 22 and 71,428 nmol $L^{-1}$ for $CH_4$ and 0 and 561% for $N_2O$.

### 3.4.  Significance of integrated GHG fluxes at basin and global scales

Since $pCO_2$ and $CH_4$ and $N_2O$ concentrations followed distinct patterns as a function of Strahler stream order (Fig. 11) we used $k_{600}$ and stream surface area as a function of Strahler stream order (Fig. S14) to compute the air-water GHG fluxes and to integrate them at basin scale. This was done separating data for streams draining and not draining the CCC since $pCO_2$ and $CH_4$ and $N_2O$ patterns were very different

(Figs. 11 and 12). The stream surface area decreased regularly with increasing Strahler stream order but showed a large increase for Strahler stream order 9 (Fig. S14). The latter mainly corresponds to the Congo mainstem that downstream of Kisangani is characterized by anastomosing river channels with extended sand bars and numerous islands (Runge 2008; O'Loughlin et al. 2013). In particular along the





section of about 500 km long between Mbandaka and Kwa mouth, the mainstem river channel undergoes a general expansion, and width increases from ~4 km to ~10 km. This corresponds to the area of CCC depression with a corresponding decrease of the slope from 6 to 3 cm km$^{-1}$ (O'Loughlin et al. 2013). The calculated $k_{600}$ decreased regularly with increasing Strahler stream order, as previously reported (Butman and Raymond 2011; Raymond et al. 2012; Deirmendjian and Abril 2018; Liu and Raymond 2018) due to higher turbulence in low order streams associated with higher stream flow due to steeper slopes. HydroSHEDS stream order classification is missing at least 1 stream order, because small streams are not correctly represented (Benstead and Leigh 2012; Raymond et al. 2013). Hence, to correct this bias, we added 1 to stream orders determined by HydroSHEDS, meaning that the lowest stream order to which GHG were attributed was 2. We then extrapolated $pCO_2$ and $CH_4$ and $N_2O$ concentrations to stream order 1, separating streams draining and not draining the CCC, using a linear regression with higher orders or by using the same value as for order 2 (Fig. S15).

The calculated $FCO_2$ ranged between 86 and 7,110 mmol m$^{-2}$ d$^{-1}$, averaging 2,469 mmol m$^{-2}$ d$^{-1}$ (weighted by surface area of Strahler stream order), encompassing the range $FCO_2$ reported by Mann et al. (2014) (312 to 1,429 mmol m$^{-2}$ d$^{-1}$) in 25 sites during a single period (November 2010) from four major river basins in the Republic of Congo (Alima, Lefini, Sangha, Likouala-Mossaka). The $pCO_2$ values ranged between 1,087 and 22,899 ppm, also encompassing the values reported by Mann et al. (2014) (2,600 to 15,802 ppm) that were not measured directly but computed from pH and DIC measurements, although pH measurements in black-water rivers can be biased by the presence of humic dissolved organic matter (Abril et al. 2015), and the addition of $HgCl_2$ seems to alter the $CO_2$ content of samples (Fig. S1). The calculated $FCH_4$ ranged between 65 and 597,260 µmol m$^{-2}$ d$^{-1}$, averaging 12,553 µmol m$^{-2}$ d$^{-1}$ (weighted by surface area of Strahler order), encompassing the range $FCH_4$ reported by Upstill-Goddard et al. (2017) (33 to 48,705 µmol m$^{-2}$ d$^{-1}$) in 41 sites draining the Congo basin in the Republic of Congo (November 2010 and August 2011) in the same four river basins sampled by Mann et al. (2014). The $CH_4$ values ranged between 22 and 71,428 nmol L$^{-1}$, also encompassing the values reported by Upstill-Goddard et al. (2017) (11 to 9,553 nmol L$^{-1}$). The calculated $FN_2O$ ranged between -52 and 319 µmol m$^{-2}$ d$^{-1}$, averaging 22 µmol m$^{-2}$ d$^{-1}$ (weighted by surface area of Strahler order), encompassing the range





$F$N$_2$O reported by Upstill-Goddard et al. (2017) (-19 to 67 µmol m$^{-2}$ d$^{-1}$) in 41 sites draining the Congo basin in the Republic of Congo. The %N$_2$O values ranged between 0 and 561 %, also encompassing the values reported by Upstill-Goddard et al. (2017) (6 to 266 %). The wider ranges of GHGs and respective fluxes we report

compared to those of Mann et al. (2014) and Upstill-Goddard et al. (2017) reflect the larger number of river systems sampled over a wider geographical area (*n*=25 *vs n*=278 for pCO$_2$; *n*=41 *vs n*=367 for CH$_4$/N$_2$O), hence, representing a wider range of river types, morphologies, catchment characteristics, and wetland density. Information on the seasonal variability of concurrent $F$CO$_2$, $F$CH$_4$ and $F$N$_2$O values

was only available in the Lualaba (at Kisangani) where the three GHGs were measured simultaneously (Fig. S16). $F$CO$_2$, $F$CH$_4$ and $F$N$_2$O were loosely positively correlated with freshwater discharge, as the seasonal variations of $k_{600}$ were small (ranging between 23.4 and 30.3 cm h$^{-1}$), and although pCO$_2$ was correlated to freshwater discharge (Fig. S13) this was not the case for CH$_4$ and N$_2$O

concentrations (Figs. 19 and 20). The range of seasonal variations at Kisangani of $F$CO$_2$ (234 and 948 mmol m$^{-2}$ d$^{-1}$), $F$CH$_4$ (116 and 876 µmol m$^{-2}$ d$^{-1}$) and $F$N$_2$O (-2 and 40 µmol m$^{-2}$ d$^{-1}$) was small compared to the range of spatial variations of $F$CO$_2$ (86 and 7,110 mmol m$^{-2}$ d$^{-1}$), $F$CH$_4$ (65 and 597,260 µmol m$^{-2}$ d$^{-1}$), and $F$N$_2$O (-52 and 319 µmol m$^{-2}$ d$^{-1}$ µmol m$^{-2}$ d$^{-1}$).

The $F$CO$_2$ and $F$CH$_4$ decreased with increasing Strahler order, as given by per surface area, and also when integrated by surface area of the streams (Table 1). Strahler orders 1-2 accounted for nearly 80% of the integrated $F$CO$_2$ (79.6%) and $F$CH$_4$ (77.0%), while Strahler orders 1-4 accounted for > 90% of the integrated $F$CO$_2$ (90.7%) and $F$CH$_4$ (91.9%). Strahler orders 5-10 only accounted for 9.3% of

integrated $F$CO$_2$ and 8.1% of integrated $F$CH$_4$. The rivers draining the CCC contributed to 6% of the basin wide emissions for CO$_2$ and 22% for CH$_4$, although the contribution in stream surface area was only 11%. The low contribution of $F$CO$_2$ from the CCC to the basin wide emissions was due to the lower $k_{600}$ values, although pCO$_2$ values were higher than the rest of the basin (Table 1). In the case of $F$CH$_4$,

the much higher CH$_4$ concentrations in the CCC overcome the lower $k_{600}$ values. $F$N$_2$O per surface area in rivers and streams outside the CCC were relatively similar for Strahler orders 10 to 3, and increased for Strahler orders 1 and 2 (Table 1). $F$N$_2$O per surface area in rivers and streams draining the CCC steadily decreased from Strahler order 8 to 1, with rivers of orders 5, 3, 2 and 1 acting as sinks for N$_2$O.





Consequently, the relative contribution per Strahler order of integrated $F$N$_2$O was less skewed than for integrated $F$CO$_2$ and $F$CH$_4$. Strahler orders 1-4 contributed 69.9% of integrated $F$N$_2$O compared to >90% for FCO$_2$ and FCH$_4$.

The rivers and streams draining the CCC were a very small sink of atmospheric N$_2$O (-0.01 GgN$_2$O-N yr$^{-1}$) while the rivers and streams outside the CCC were a source of N$_2$O (5.1 GgN$_2$O-N yr$^{-1}$). The integrated $F$N$_2$O for the Congo River network was 5.1 GgN$_2$O-N yr$^{-1}$ corresponded to 14-17% of total riverine emissions of N$_2$O reported by Hu et al. (2016). Note that the N$_2$O riverine emissions computed by Hu et al. (2016) were indirectly computed from data on global nitrogen deposition on catchments and on emission factors rather than derived from direct measurements of dissolved N$_2$O concentrations. The estimates given by Hu et al. (2016) are more conservative than older estimates (e.g. Kroeze et al. 2010) because they are based on revised emission factors, and converge with a similar more recent study by Maavara et al. (2018). Our estimate of the integrated $F$N$_2$O is consistent with the range of N$_2$O emissions of 3.8 to 4.3 GgN$_2$O-N yr$^{-1}$ given by Maavara et al. (2018) for the Congo river network, also based on an indirect calculation based on nitrogen deposition and emission factors.

The integrated $F$CH$_4$ for the Congo River network of 1.7 TgCH$_4$ yr$^{-1}$ is nearly two times higher than the estimate given by Bastviken et al. (2011) for tropical rivers, and corresponds to 6% of the global emission of CH$_4$ from rivers given by Stanley et al. (2016), while the surface area of Congo River network corresponds to a lower proportion (3 %) of the global riverine surface area (773,000 km$^2$, Allen and Pavelsky). Note that the meta-analysis of Stanley et al. (2016) includes part of our data-set from Congo River, as published by Borges et al. (2015a). The integrated $F$CH$_4$ we report for the Congo River network is also more than three times higher than the estimate of CH$_4$ emissions for Amazonian large rivers reported by Sawakuchi et al. (2014) (0.49 TgCH$_4$ yr$^{-1}$). The integrated $F$CH$_4$ we report for the Congo River network corresponds to 7% of the emission from the Congo wetlands inferred from remote sensed atmospheric CH$_4$ data (Bloom et al. 2010) (25.7 TgCH$_4$ yr$^{-1}$). Note that the estimate given by Bloom et al. (2011) includes the CH$_4$ emission from all ecosystems over the Congo basin, and should also include the fluvial emissions. Note that the CH$_4$ emissions we report for the Congo River basin only include the diffusive flux component, when the ebullitive CH$_4$ emission component represents the majority of CH$_4$ emissions from inland waters (Bastviken et al. 2011),



which would be consistent with the gap between our emission estimates and those from atmospheric $CH_4$ inventories. Finally, note that the $CH_4$ emissions were only calculated and integrated for the rivers and streams draining the CCC but not for the actual wetland flooded area of the CCC. The emission of $CH_4$ from actual wetland

flooded area of the CCC can be estimated to a massive 51 $TgCH_4$ $yr^{-1}$ by extrapolating the area averaged $FCH_4$ from the streams (24,468 µmol $m^{-2}$ $d^{-1}$, Table 1) to the flooded extent (360,000 $km^2$, Bwangoy et al. 2010). This corresponds to about 29% of the $CH_4$ emissions from natural wetlands (~180 $TgCH_4$ $yr^{-1}$, Saunois et al. 2010), and is higher than the estimate of $CH_4$ emissions from the Congo wetlands

inferred from remote sensed atmospheric $CH_4$ data (Bloom et al. 2010).

The integrated $FCO_2$ for the Congo River network is 251 $TgC$ $yr^{-1}$ and is equivalent to the $CO_2$ emission value for rivers globally given by Cole et al. (2007), and to 14% and 39% of the $CO_2$ emission value for rivers globally given by Raymond et al. (2013) (1,8000 $TgC$ $yr^{-1}$) and Lauerwald et al. (2015) (650 $TgC$ $yr^{-1}$),

respectively. The integrated $CO_2$ emission from the Congo River network corresponds to 44% of the $CO_2$ emissions from tropical (24°N-24°S) oceans globally (565 $TgC$ $yr^{-1}$) and 183% of $CO_2$ emissions from the tropical Atlantic Ocean (137 $TgC$ $yr^{-1}$), based on the Takahashi et al. (2002) $FCO_2$ climatology. The terrestrial net ecosystem exchange (NEE) of the watershed of the Congo River can be estimated

based on the NEE estimate of 23 $gC$ $m^{-2}$ $d^{-1}$ for savannahs given by Ciais et al. (2011) and of 20 $gC$ $m^{-2}$ $d^{-1}$ for forests given by Fisher et al. (2013). Based on the respective land cover from GLC2009 (30% savannah and 70% forest) the terrestrial NEE is 77 $TgC$ $yr^{-1}$, more than three times lower than the riverine $CO_2$ emission from the Congo River. This is extremely surprising since the hydrological export from *terra*

*firme* forests of DOC and DIC, that are assumed to sustain fluvial emissions are typically 2-3% compared to terrestrial NEE (Kindler et al. 2011; Deirmendjian et al. 2018). Hydrological carbon export is higher compared to NEE in European grasslands (on average 22%) (Kindler et al. 2011). We ignore if this is transposable to tropical grasslands, such as savannahs, although they only occupy 30% of the

Congo catchment surface. Accordingly, the $CO_2$ emission from the Congo River network should have been an order of magnitude lower than the estimates of terrestrial NEE from *terra firme* biomes rather than more than three times higher. However, in wetlands such as peatlands in Europe, the hydrological export of DOC and DIC represent 109% of NEE and is enough to sustain the riverine $CO_2$ emission





that represents 17% of NEE (Billet et al. 2004). This would then strongly suggest that the $CO_2$ emission from the lowland Congo River network is to a large extent sustained by another source of carbon than from the terrestrial *terra firme* biome. The most likely alternate source would be wetlands (flooded forest and aquatic

macrophytes), in agreement with the analysis in the Central Amazon River by Abril et al. (2013).

## 4. Conclusions

Net heterotrophy in rivers and lakes sustained by inputs of organic matter from the terrestrial vegetation on the catchments is the prevailing paradigm to explain over-saturation of $CO_2$ in inland surface waters and corresponding emissions to the atmosphere, based on process studies, the earliest in the Amazon (Wissmar et al. 1981) and boreal systems (Del Giorgio et al. 1999; Prairie et al. 2002), and then

generalised at global scales for lakes (Cole et al. 1994) and rivers (Cole and Caraco 2001). Yet, the comparison of 169 measurements of aquatic NEP and $F$CO$_2$ estimates in the central Congo River network covering a wide range of size and type of rivers and streams shows that the aquatic NEP cannot account for fluvial $F$CO$_2$. This implies that lateral inputs of $CO_2$ sustain a large part of the $CO_2$ emissions from

rivers and streams in the Congo River network.

     The comparison of the integrated $CO_2$ emission from the Congo River network with terrestrial *terra firme* NEE shows that it unlikely that fluvial $CO_2$ emissions are sustained by lateral hydrological transfer of carbon from *terra firme*. Indeed, integrated $F$CO$_2$ from the river network was more than three times higher than

terrestrial NEE, when forests typically only export a very small fraction of NEE as carbon to rivers that can sustain fluvial $CO_2$ emissions. It is then likely that the fluvial $CO_2$ emissions from the central Congo River are sustained by organic and inorganic $CO_2$ inputs from extensive riparian wetlands (flooded forest and aquatic floating macrophytes). This is consistent with the stable isotopic signature of DIC, the

differences in the spatial distribution of dissolved $CO_2$, %$O_2$, and $CH_4$ between rivers and streams draining or not draining the CCC, a large wetland region in the core of the basin, and based on the correlation between pCO$_2$ levels and the cover of flooded forest in the catchment.





The fact that fluvial $CO_2$ emissions in lowland rivers are to a large extent sustained by carbon inputs from wetlands in addition to those from *terra firme* has consequences for the conceptualisation of statistical and mechanistic models of carbon cycling in river networks. While progress has been made in integrating

wetland connectivity in mechanistic regional models (Lauerwald et al. 2017), this has not been the case so far for statistical global models that rely on terrestrial (*terra firme*) productivity (Lauerwald et al. 2015). The comparison of the output of such a statistical model for the Congo River with observational data (Fig. S17) shows that the model fails to represent spatial gradients, and in particular the higher $pCO_2$

values of streams and rivers draining the CCC. This illustrates how ignoring the river-wetland connectivity can lead to the misrepresentation of $pCO_2$ dynamics in river networks, in particular tropical ones that account for the vast majority (80%) of global riverine $CO_2$ emissions.

**Data availability.** *Full data-set will be made publically available at www.zenodo.org should the manuscript be accepted in Biogeosciences.*

**Author contributions.** AVB and SB conceived the study; AVB, FD, TL, ET, ATS, TB, JB, CRT, SB collected field samples; AVB, FD, TL, CM, CRT, J-PD, SB made
laboratory analysis; GA and TL carried out GIS analysis; AVB drafted the manuscript with substantial inputs from SB; all authors contributed to manuscript.

**Acknowledgements.** This work was funded by the European Research Council (ERC-StG 240002 AFRIVAL), the Fonds National de la Recherche Scientifique
(FNRS, TransCongo, 14711103), the Belgian Federal Science Policy (BELSPO, COBAFISH SD/AR/05A), the Research Foundation Flanders (FWO-Vlaanderen), the Research Council of the KU Leuven, and the Fonds Leopold-III pour l'Exploration et Conservation de la Nature. The Boyekoli-Ebale-Congo 2010 Expedition was funded by the Belgian Development Cooperation, BELSPO, and Belgian National Lottery.
We thank the Isotope Hydrology Laboratory of IAEA for analyses of water stable isotope ratios which contribute to the Coordinated Research Project CRPF33021 (Application and development of isotope techniques to evaluate human impacts on water balance and nutrient dynamics of large river basins) implemented by the IAEA, Jonathan Richir (University of Liège) for advise with the statistical analysis, Sandro

Petrovic and Marc-Vincent Commarieu (University of Liège), Bruno Leporcq (UNamur), and Zita Kelemen (KU Leuven) for analytical assistance, the Régie des Voies Fluviales (RFV, Kinshasa) for providing water height data. AVB is a senior research associate at the FNRS.

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





Table 1: Partial pressure of $CO_2$ (p$CO_2$ in ppm), dissolved $CH_4$ concentration (nmol $L^{-1}$), dissolved $N_2O$ saturation level (%$N_2O$ in %), gas transfer velocity ($k_{600}$ in cm $h^{-1}$), air-water fluxes of $CO_2$ ($FCO_2$ in mmol $m^{-2}$ $d^{-1}$ and in TgC $yr^{-1}$), of $CH_4$ ($FCH_4$ in µmol $m^{-2}$ $d^{-1}$ and in TgC$H_4$ $yr^{-1}$), of $N_2O$ ($FN_2O$ in µmol $m^{-2}$ $d^{-1}$ and in GgN$_2$O-N $yr^{-1}$), stream surface area (S.A. in $km^2$), as a function of Strahler stream order (S.O.) in the Congo River network for rivers and streams draining and not draining the Cuvette Centrale Congolaise.

| S.O. | p$CO_2$ ppm | $CH_4$ nmol $L^{-1}$ | $N_2O$ nmol $L^{-1}$ | %$N_2O$ % | $k_{600}$ cm $h^{-1}$ | Temp. °C | $FCO_2$ mmol $m^2$ $d^{-1}$ | $FCH_4$ µmol $m^2$ $d^{-1}$ | $FN_2O$ mmol $m^2$ $d^{-1}$ | S.A. $km^2$ | $FCO_2$ TgC $yr^{-1}$ | $FCH_4$ TgC$H_4$ $yr^{-1}$ | $FN_2O$ GgN$_2$O-N $yr^{-1}$ |
|---|---|---|---|---|---|---|---|---|---|---|---|---|---|
| **Draining the Cuvette Centrale Congolaise** | | | | | | | | | | | | | |
| 1 | 12,304 | 8,349 | 0.5 | 7.3 | 26.4 | 27.0 | 2,892 | 61,732 | -43.1 | 350 | 4.4 | 0.126 | -0.154 |
| 2 | 12,304 | 8,349 | 2.7 | 44.1 | 16.1 | 27.0 | 1,766 | 37,747 | -15.9 | 335 | 2.6 | 0.074 | -0.054 |
| 3 | 11,923 | 7,411 | 4.9 | 77.5 | 13.0 | 26.1 | 1,376 | 25,765 | -5.2 | 324 | 2.0 | 0.049 | -0.017 |
| 4 | 12,065 | 9,409 | 6.5 | 104.5 | 12.3 | 26.7 | 1,323 | 33,431 | 0.9 | 291 | 1.7 | 0.057 | 0.003 |
| 5 | 13,705 | 9,533 | 2.0 | 32.2 | 12.5 | 27.0 | 1,525 | 33,255 | -14.8 | 278 | 1.9 | 0.054 | -0.042 |
| 6 | 7,830 | 532 | 13.2 | 214.7 | 12.2 | 27.5 | 834 | 1,855 | 24.7 | 480 | 1.8 | 0.005 | 0.121 |
| 7 | 6,643 | 503 | 11.3 | 188.8 | 11.7 | 27.5 | 670 | 1,582 | 18.2 | 163 | 0.5 | 0.002 | 0.030 |
| 8 | 6,977 | 401 | 12.9 | 219.8 | 13.6 | 28.7 | 824 | 1,581 | 28.6 | 366 | 1.3 | 0.003 | 0.107 |
| **Not draining the Cuvette Centrale Congolaise** | | | | | | | | | | | | | |
| 1 | 10,719 | 1,584 | 9.4 | 136.6 | 161.6 | 24.1 | 15,417 | 66,695 | 107.2 | 2235 | 150.9 | 0.871 | 2.449 |
| 2 | 8,921 | 1,275 | 9.4 | 135.8 | 56.5 | 24.1 | 4,450 | 19,171 | 36.4 | 2143 | 41.8 | 0.240 | 0.798 |
| 3 | 7,225 | 1,185 | 9.3 | 133.4 | 31.1 | 24.0 | 1,972 | 9,554 | 18.9 | 1883 | 16.3 | 0.105 | 0.364 |
| 4 | 5,766 | 726 | 8.5 | 126.3 | 21.5 | 24.3 | 1,065 | 4,098 | 10.1 | 1752 | 8.2 | 0.042 | 0.181 |
| 5 | 4,110 | 538 | 8.6 | 131.4 | 17.2 | 25.5 | 589 | 2,452 | 9.6 | 1688 | 4.4 | 0.024 | 0.165 |
| 6 | 1,929 | 299 | 9.3 | 143.4 | 16.8 | 25.6 | 239 | 1,328 | 12.9 | 1772 | 1.9 | 0.014 | 0.233 |
| 7 | 2,667 | 220 | 9.3 | 153.6 | 16.3 | 27.5 | 343 | 1,030 | 15.4 | 2168 | 3.3 | 0.013 | 0.340 |
| 8 | 3,010 | 226 | 6.7 | 109.2 | 12.9 | 27.7 | 310 | 837 | 2.1 | 1696 | 2.3 | 0.008 | 0.036 |
| 9 | 2,521 | 170 | 8.7 | 145.7 | 10.9 | 27.7 | 217 | 533 | 8.8 | 4639 | 4.4 | 0.014 | 0.419 |
| 10 | 3,445 | 33 | 9.4 | 153.1 | 20.4 | 27.7 | 574 | 178 | 19.2 | 646 | 1.6 | 0.001 | 0.127 |
| **Total** | | | | | | | | | | | 251 | 1.7 | 5.1 |



Figure 1: Freshwater discharge (m³ s⁻¹) of the Congo River at Kinshasa from 2010 to 2015, with an indication of field expedition duration (thick lines).

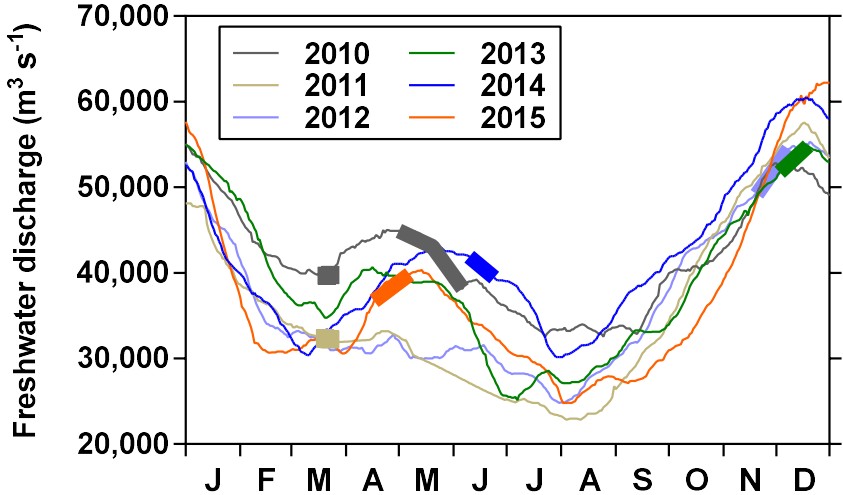





Figure 2: Map showing the sampling stations of the ten field expeditions in the Congo River network.



Figure 3: Variation in surface waters of the specific conductivity (µS cm$^{-1}$), water temperature (°C), oxygen stable isotope composition of H$_2$O ($\delta^{18}$O-H$_2$O in ‰), total suspended matter (TSM in mg L$^{-1}$), total alkalinity (TA in µmol kg$^{-1}$), dissolved organic carbon (DOC in mg L$^{-1}$), carbon stable isotope composition of dissolved inorganic carbon ($\delta^{13}$C-DIC in ‰), pH, partial pressure of CO$_2$ (pCO$_2$ in ppm), dissolved O$_2$ saturation level (%O$_2$ in %), dissolved CH$_4$ concentration (nmol L$^{-1}$), dissolved N$_2$O saturation level (%N$_2$O in %) as a function of the distance upstream of Kinshasa along a transect along the Congo River from Kisangani (03/12/2013-19/12/2013, $n$=10,505). Grey and black symbols indicate samples from the mainstem and green samples from tributaries.

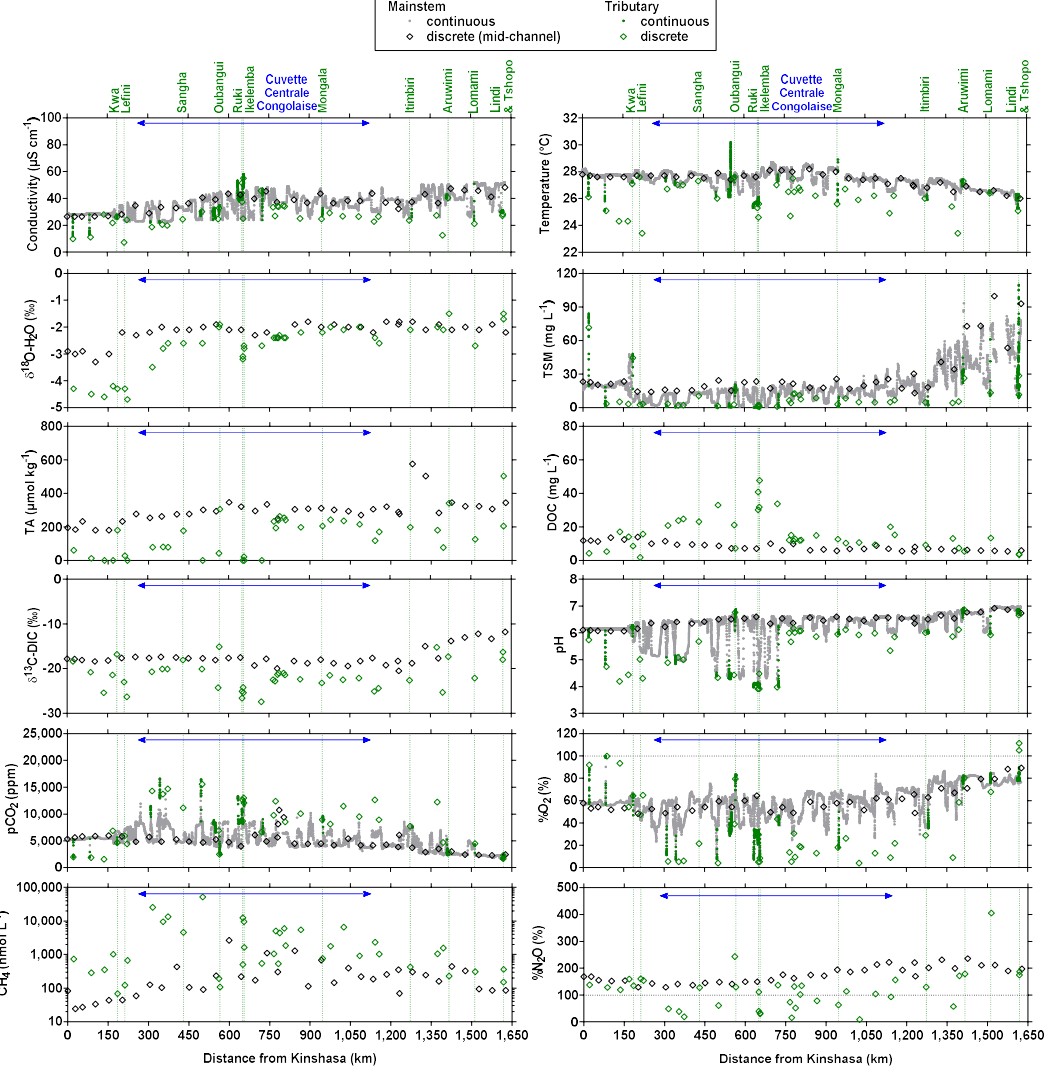



Figure 4: Variation in surface waters of the specific conductivity (µS cm$^{-1}$), water temperature (°C), oxygen stable isotope composition of $H_2O$ ($\delta^{18}O$-$H_2O$ in ‰), total suspended matter (TSM in mg L$^{-1}$), total alkalinity (TA in µmol kg$^{-1}$), dissolved organic carbon (DOC in mg L$^{-1}$), carbon stable isotope composition of dissolved inorganic carbon ($\delta^{13}C$-DIC in ‰), pH, partial pressure of $CO_2$ ($pCO_2$ in ppm), dissolved $O_2$ saturation level (%$O_2$ in %), dissolved $CH_4$ concentration (nmol L$^{-1}$), dissolved $N_2O$ saturation level (%$N_2O$ in %) as a function of the distance upstream of Kinshasa along a transect along the Congo River from Kisangani (10/06/14-30/06/14, *n*=12,968). Grey and black symbols indicate samples from the mainstem and green samples from tributaries.

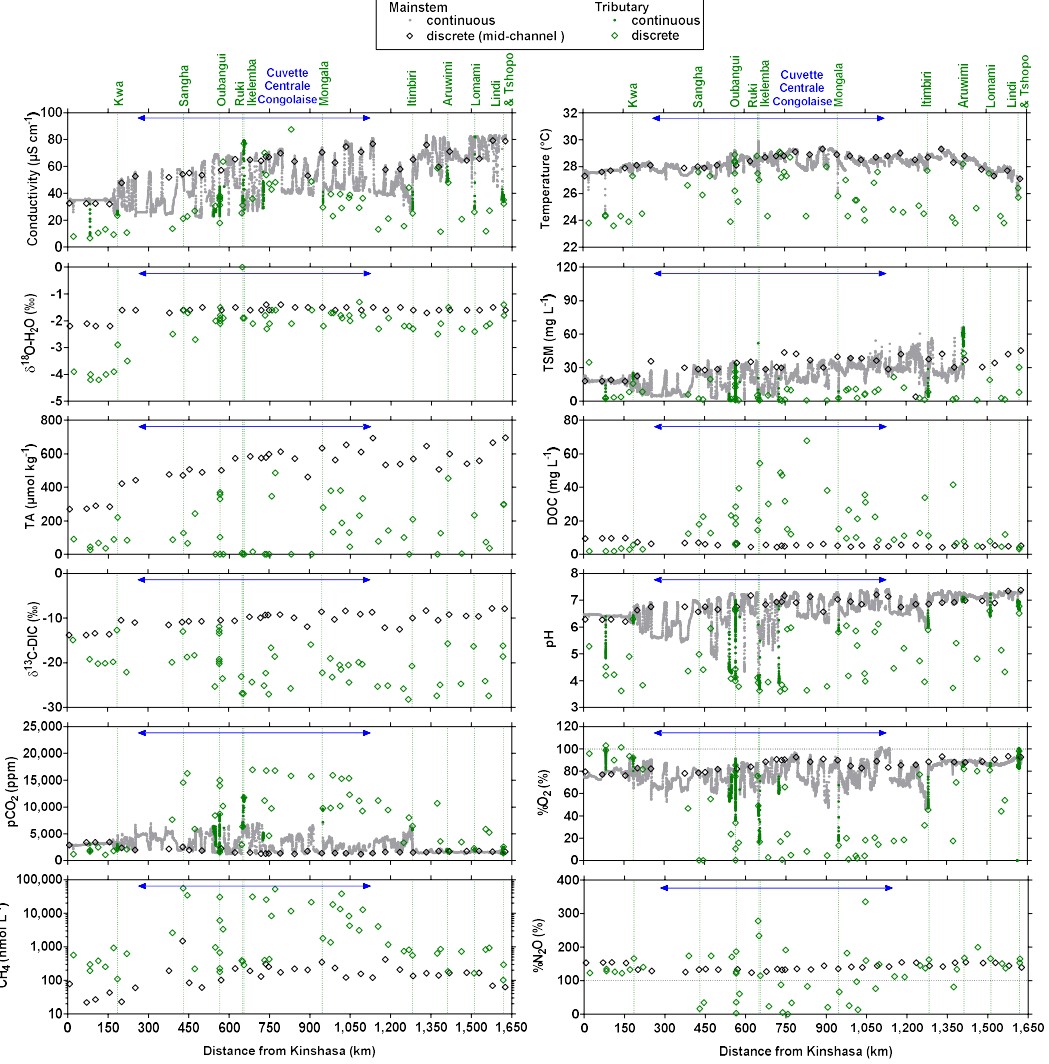



Figure 5: Carbon stable isotope composition of $CH_4$ ($\delta^{13}C$-$CH_4$ in ‰) in surface waters of the Congo River mainsteam (black symbols) and tributaries (green symbols) as a function of dissolved $CH_4$ concentration (nmol $L^{-1}$) and as function of the distance upstream of Kinshasa, obtained along a longitudinal transect along the Congo River from Kisangani (10/06/14-30/06/14). Dotted line indicates linear regression.

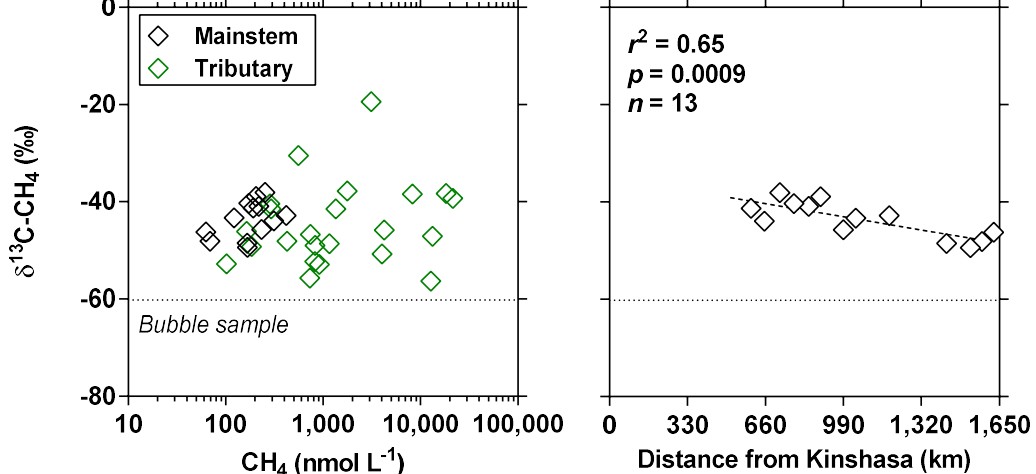



Figure 6: Carbon stable isotope composition of particulate organic carbon ($\delta^{13}$C-POC in ‰) in surface waters of the Congo River network as a function of dissolved $O_2$ saturation level (%$O_2$ in %) and dissolved $CH_4$ concentration (nmol $L^{-1}$). Grey lines indicate average (full line) and standard deviation (dotted lines) of average soil organic carbon stable isotope composition with a dominance of $C_4$ (-15.5±0.8 ‰) and $C_3$ plants (-28.4±0.7 ‰) (Bird and Pousai 1997). Data with $\delta^{13}$C-POC < -30‰ were associated to significantly lower %$O_2$ and higher $CH_4$ than data with $\delta^{13}$C-POC > -30‰ (Mann-Whitney $p$<0.0001 for both tests).

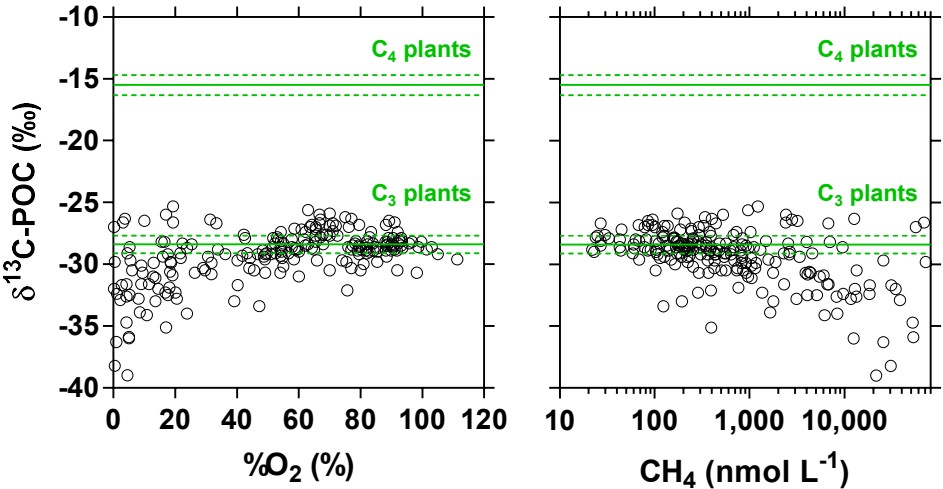





Figure 7: Dissolved $N_2O$ saturation level (%$N_2O$ in %), ratio of $NH_4^+$ to dissolved inorganic nitrogen (DIN=$NH_4^+$+$NO_2^-$+$NO_3^-$) (µmol:µmol), and ratio of $NO_3^-$ to DIN (µmol:µmol) in surface waters of the Congo River network as a function of dissolved $O_2$ saturation level (%$O_2$ in %). Outliers (red dots) were identified with a Cook's Distance procedure and removed prior linear regression analysis (solid line).

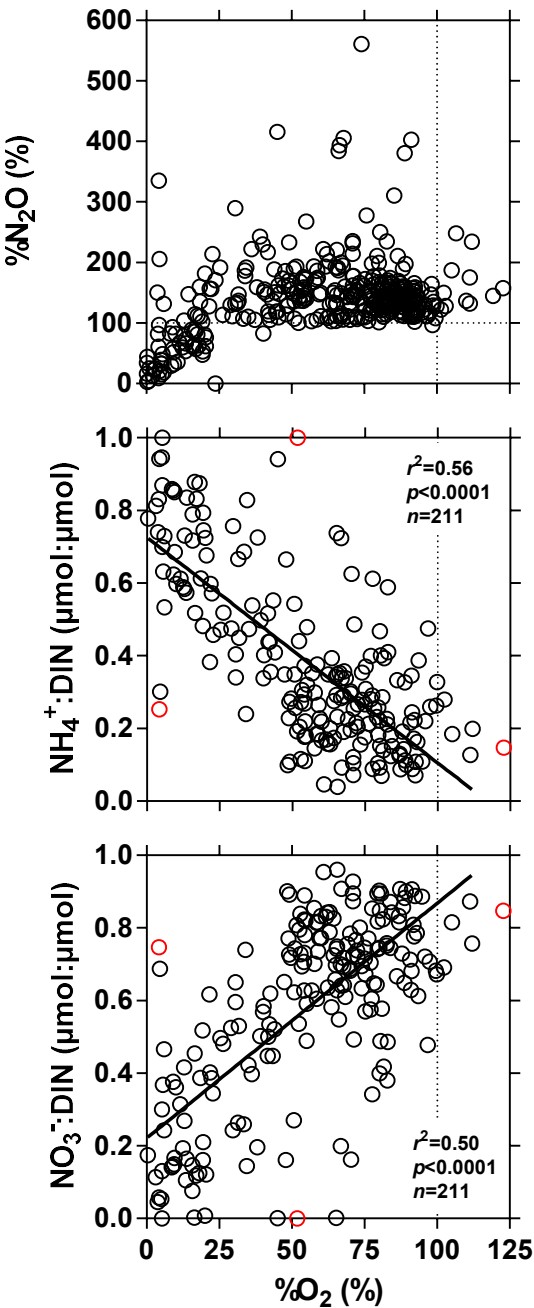



Figure 8: Dissolved $N_2O$ saturation level (%$N_2O$ in %), as a function of the ratio of $NH_4^+$ to dissolved inorganic nitrogen (DIN=$NH_4^+$+$NO_2^-$+$NO_3^-$) (µmol:µmol), and of the ratio of $NO_3^-$ to DIN (µmol:µmol) in surface waters of the Congo River network. Outliers (red dots) were identified with a Cook's Distance procedure and removed prior linear regression analysis (solid line).

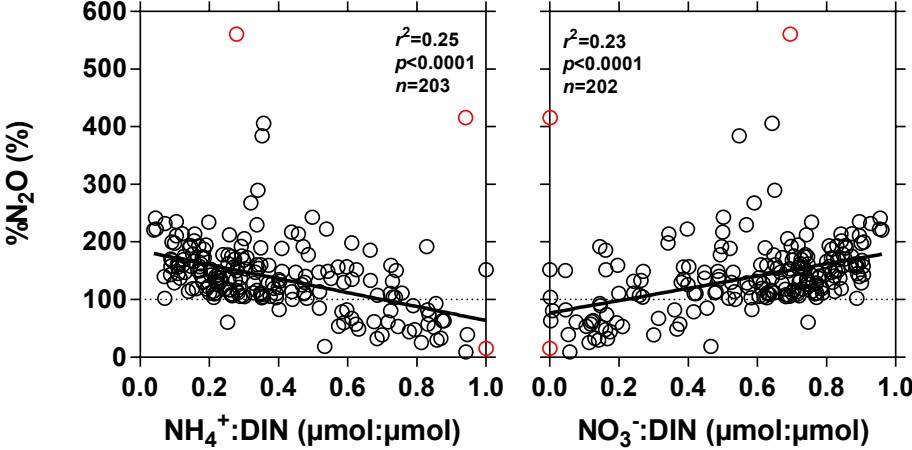



Figure 9: Variations in surface water of the partial pressure of $CO_2$ ($pCO_2$ in ppm), dissolved oxygen saturation level ($\%O_2$ in %), pH, specific conductivity ($\mu S\ cm^{-1}$), water temperature (°C) and total suspended matter (TSM in $mg\ L^{-1}$) along perpendicular transects to the mainstem Congo River as function of distance from the left bank (m), and at a variable distance from Kinshasa (10/06/14-30/06/14).

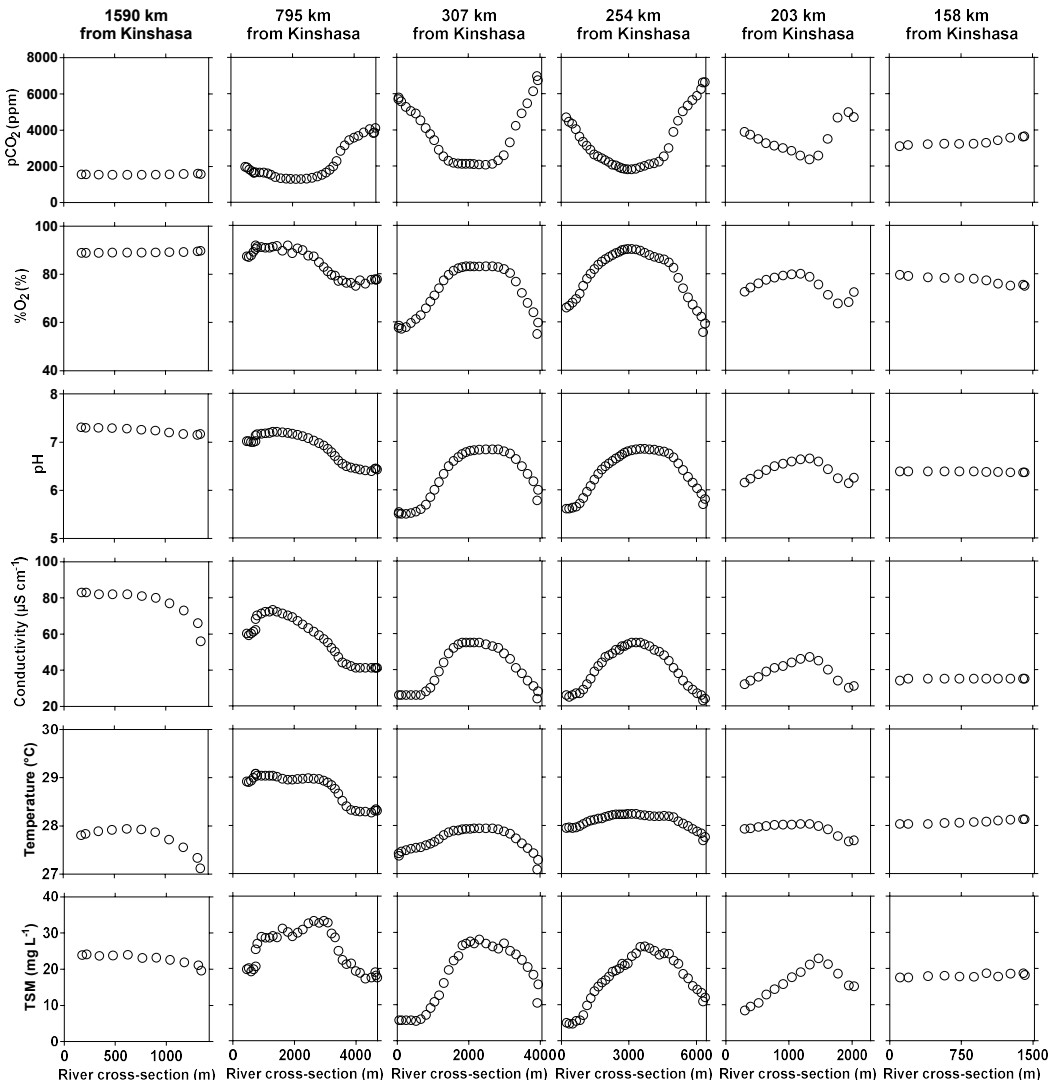



Figure 10: Variation in surface waters of the specific conductivity (µS cm$^{-1}$), water temperature (°C), oxygen stable isotope composition of H$_2$O ($\delta^{18}$O-H$_2$O in ‰), total suspended matter (TSM in mg L$^{-1}$), total alkalinity (TA in µmol kg$^{-1}$), dissolved organic carbon (DOC in mg L$^{-1}$), carbon stable isotope composition of dissolved inorganic carbon ($\delta^{13}$C-DIC in ‰), pH, partial pressure of CO$_2$ (pCO$_2$ in ppm), dissolved O$_2$ saturation level (%O$_2$ in %), dissolved CH$_4$ concentration (nmol L$^{-1}$), dissolved N$_2$O saturation level (%N$_2$O in %) as a function of the distance upstream of the Kwa mouth along a transect along the Kwa River (16/04/15-06/05/15, $n$=7,017). Grey and black symbols indicate samples from the mainstem and green samples from tributaries.

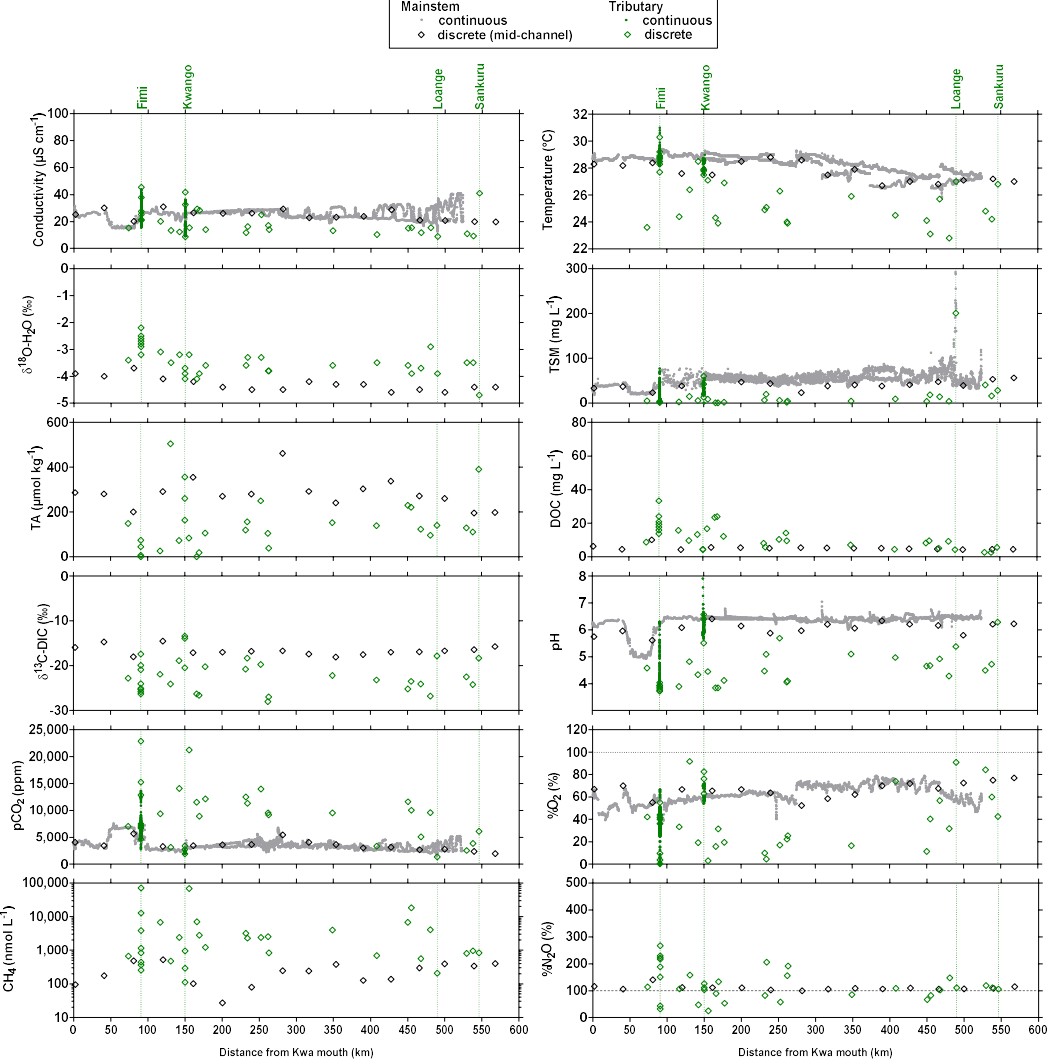



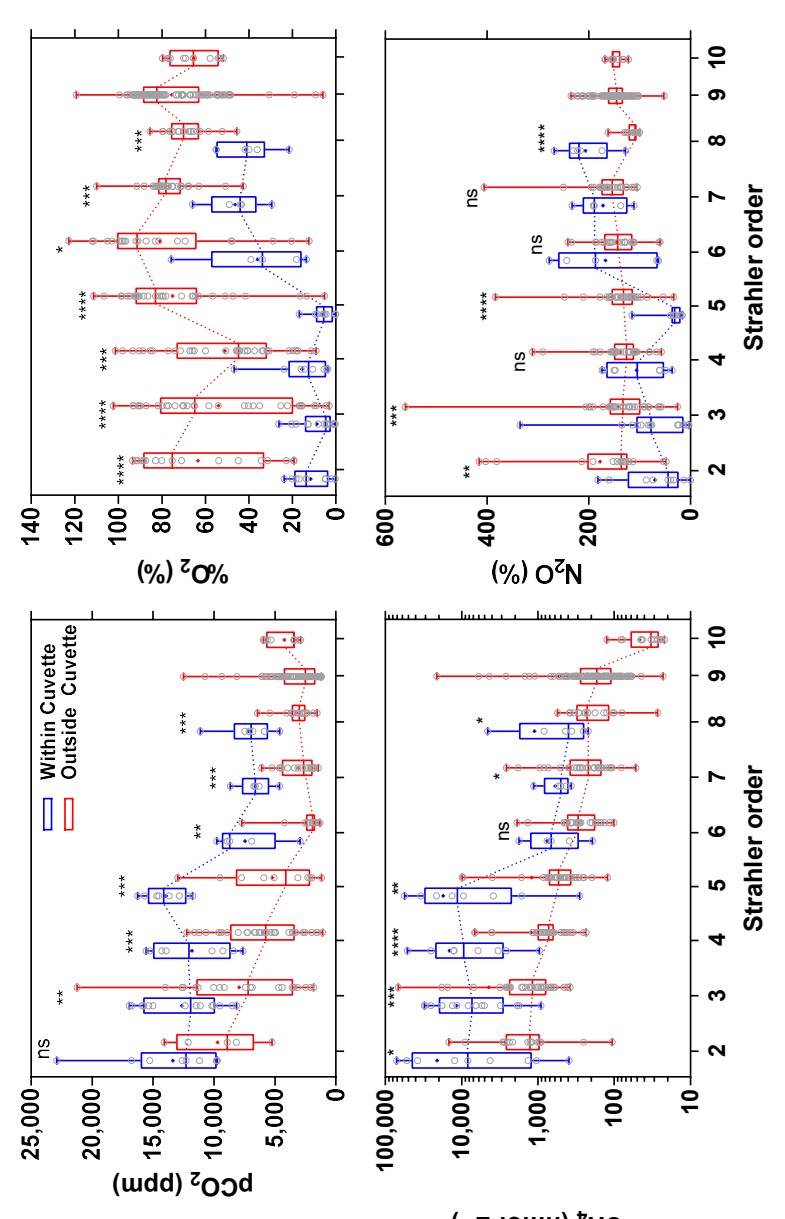

Figure 11: Box plot as function of Strahler stream order of the partial pressure of $CO_2$ (p$CO_2$ in ppm), dissolved $CH_4$ concentration (nmol $L^{-1}$), dissolved $N_2O$ saturation level (%$N_2O$ in %), and dissolved $O_2$ saturation level (%$O_2$ in %) for rivers and streams of the Congo River network draining and not draining the Cuvette Centrale Congolaise. The box represents the first and third quartile, horizontal line corresponds to the median, the cross to the average, error bars correspond to the maximum and minimum, symbols show all data points. A Mann-Whitney test was used to test statistical differences: ns = not significant, **** = p<0.0001; *** = p<0.001; ** =p<0.01; * =p<0.05.





Figure 12: Box plot of the partial pressure of CO$_2$ (pCO$_2$ in ppm), dissolved O$_2$ saturation level (%O$_2$ in %), dissolved CH$_4$ concentration (nmol L$^{-1}$), and dissolved N$_2$O saturation level (%N$_2$O in %) in surface waters of rivers and streams of the Congo river network draining and not draining the Cuvette Centrale Congolaise for small and large systems (Strahler stream order 5≤ and >5, respectively) (03/12/2013-19/12/2013; 10/06/14-30/06/14). The box represents the first and third quartile, horizontal line corresponds to the median, the cross to the average, error bars correspond to the maximum and minimum, symbols show all data points. A Mann-Whitney test was used to test statistical differences: **** = p<0.0001; *** = p<0.001; **=p<0.01.

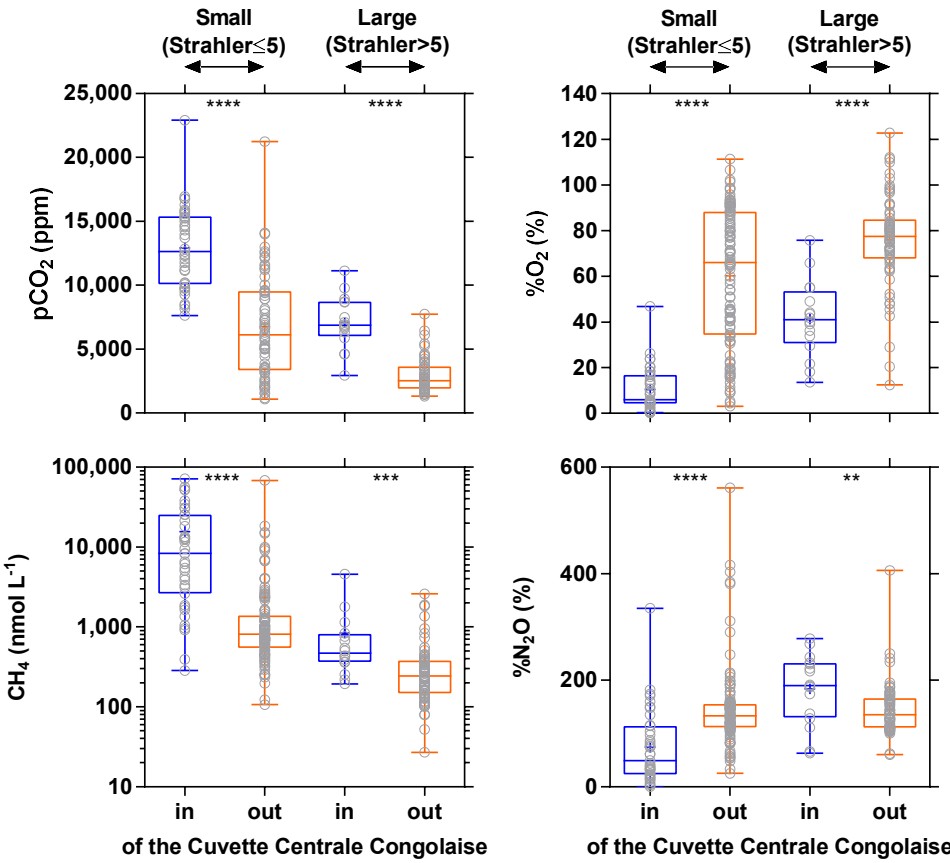





Figure 13: Partial pressure of $CO_2$ (p$CO_2$ in ppm) and dissolved $O_2$ saturation level (%$O_2$ in %) in surface waters of rivers and streams of the Congo River network as a function of the flooded dense forest over the respective catchment (Global Land Cover 2009). Grey open dots are individual data points, and black full dots are binned averages (± standard deviation) by intervals of 20%.

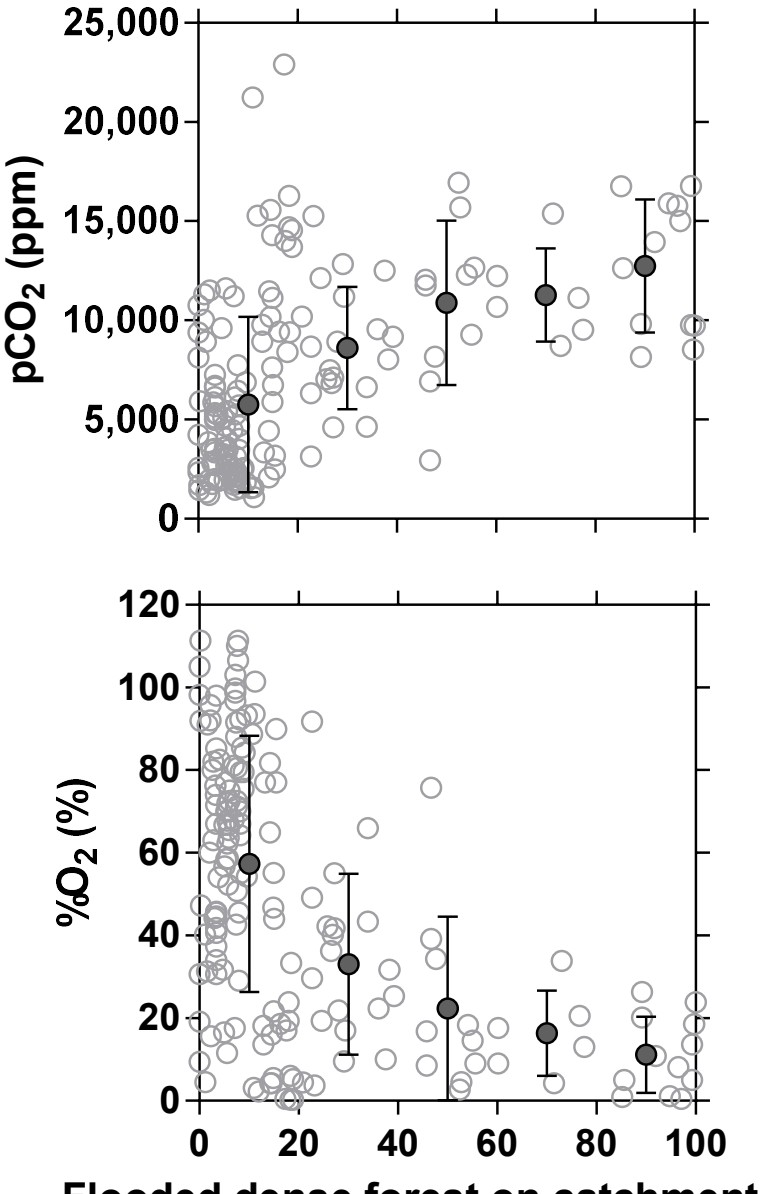





Figure 14: Ratio of dissolved $CH_4$ and $CO_2$ concentration (µmol:µmol) plotted as a function of $O_2$ saturation level (%$O_2$ in %) and plotted in box plots draining and not draining the Cuvette Centrale Congolaise for small and large systems (Strahler stream order 5≤ and >5, respectively) (03/12/2013-19/12/2013; 10/06/14-30/06/14). The box represents the first and third quartile, horizontal line corresponds to the median, the cross to the average, error bars correspond to the maximum and minimum, symbols show all data points. A Mann-Whitney test was used to test statistical differences: ns=not significant; **** = p<0.0001.

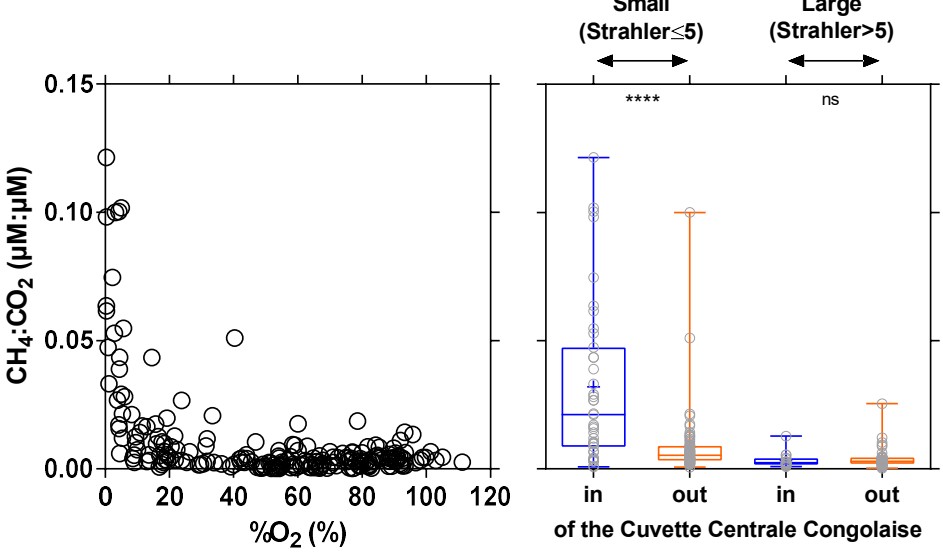





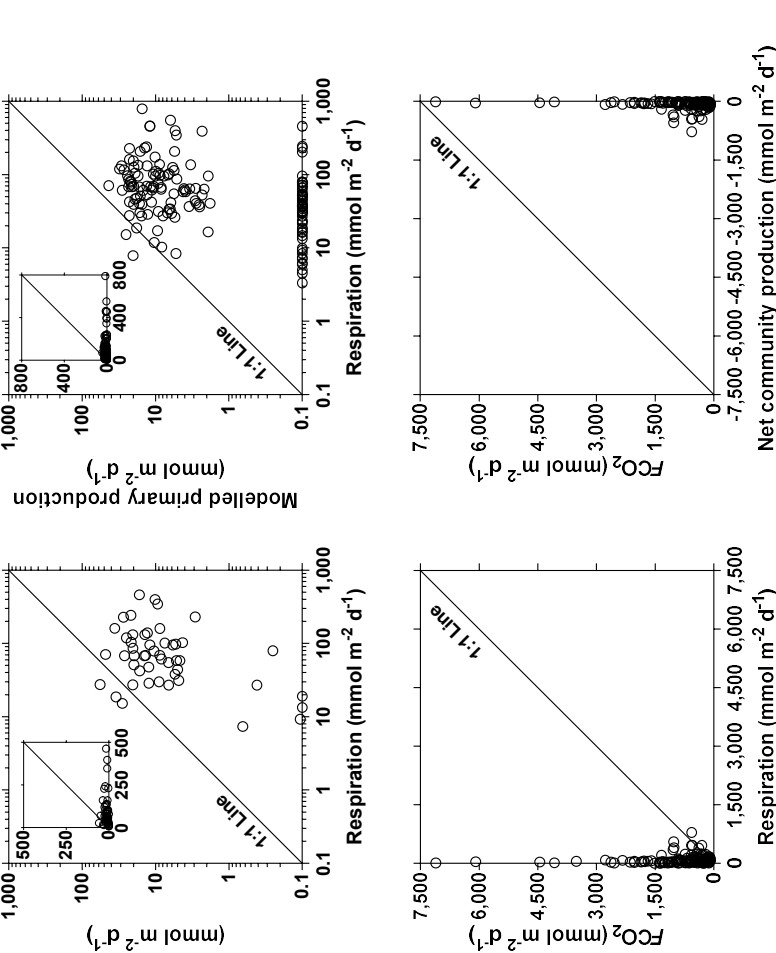

Figure 15: Primary production (measured and modelled) (mmol m$^{-2}$ d$^{-1}$) as a function of community respiration (mmol m$^{-2}$ d$^{-1}$), and air-water $CO_2$ fluxes ($FCO_2$ in mmol m$^{-2}$ d$^{-1}$) as a function of community respiration (mmol m$^{-2}$ d$^{-1}$) and net community production (mmol m$^{-2}$ d$^{-1}$) in surface waters of rivers and streams of the Congo River network. Insets show the data in a linear scale (instead of a log-log scale).



Figure 16: Carbon stable isotope composition of dissolved inorganic carbon (DIC) ($\delta^{13}$C-DIC in ‰) as a function of the total alkalinity (TA) to DIC ratio (µmol:µmol) and as function of the partial pressure of $CO_2$ (pCO$_2$ in ppm) for a TA:DIC ratio equal to zero in surface waters of of rivers and streams of the Congo River network. Open dots indicate individual data points, full dots indicate binned averages (± stand deviation) (bins <5,000, 5,000-10,000 and >10,000 ppm). Horizontal dotted lines indicate the $\delta^{13}$C values of atmospheric $CO_2$ and of average soil organic carbon stable isotope composition with a dominance of $C_4$ (-15.5±0.8 ‰) and $C_3$ plants (-28.4±0.7 ‰) (Bird and Pousai 1997). Red dotted line provides polynomial fit ($\delta^{13}$C-DIC = - 22.5 + 21.37 x TA:DIC - 6.97 x TA:DIC, $r^2$=0.76)

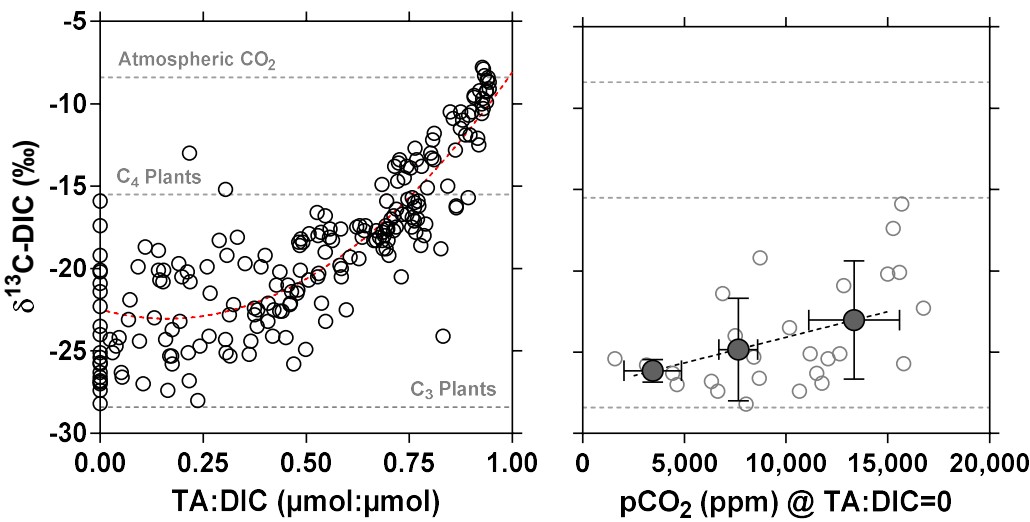



Figure 17: Total alkalinity (TA) and Mg$^{2+}$ as function of Ca$^{2+}$ of the surface waters of rivers and streams of the Congo River network, in Na$^+$ normalized plots (μmol:μmol) showing the composition fields for rivers draining different lithologies from a global compilation of the 60 largest rivers in the World (Gaillardet et al., 1999).

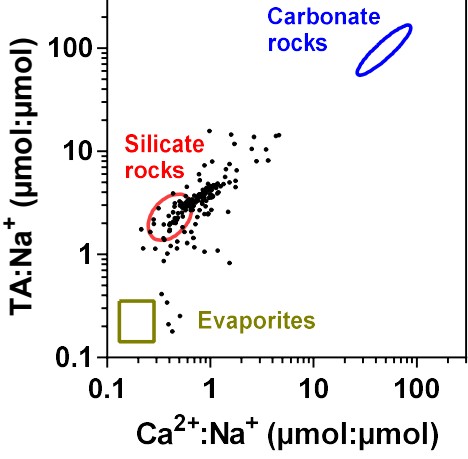 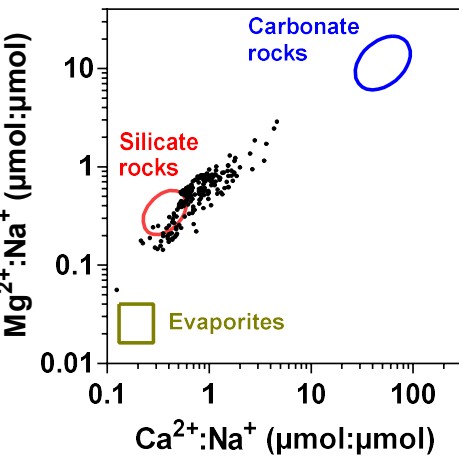



Figure 18: Comparison of the partial pressure of $CO_2$ (pCO$_2$ in ppm), dissolved $CH_4$ concentration (nmol L$^{-1}$), dissolved $O_2$ saturation level (%O$_2$ in %), dissolved $N_2O$ saturation level (%N$_2O$ in %) in surface waters of the Congo River tributaries sampled during both high water (03/12/2013-19/12/2013) and falling water periods (10/06/14-30/06/14). Tributaries were separated into left and right bank, as well as into large and small systems, with a freshwater discharge (Q) < and ≥ 300 m$^3$ s$^{-1}$, respectively.

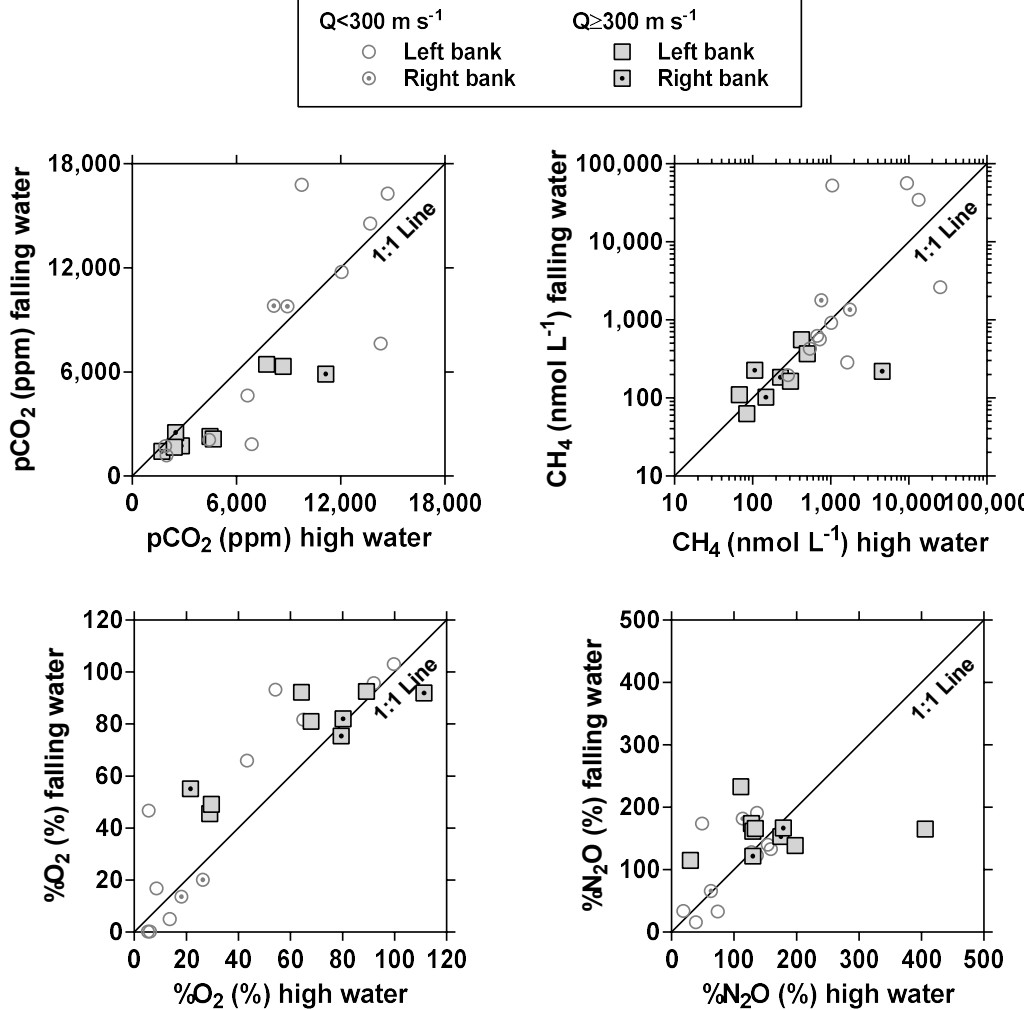





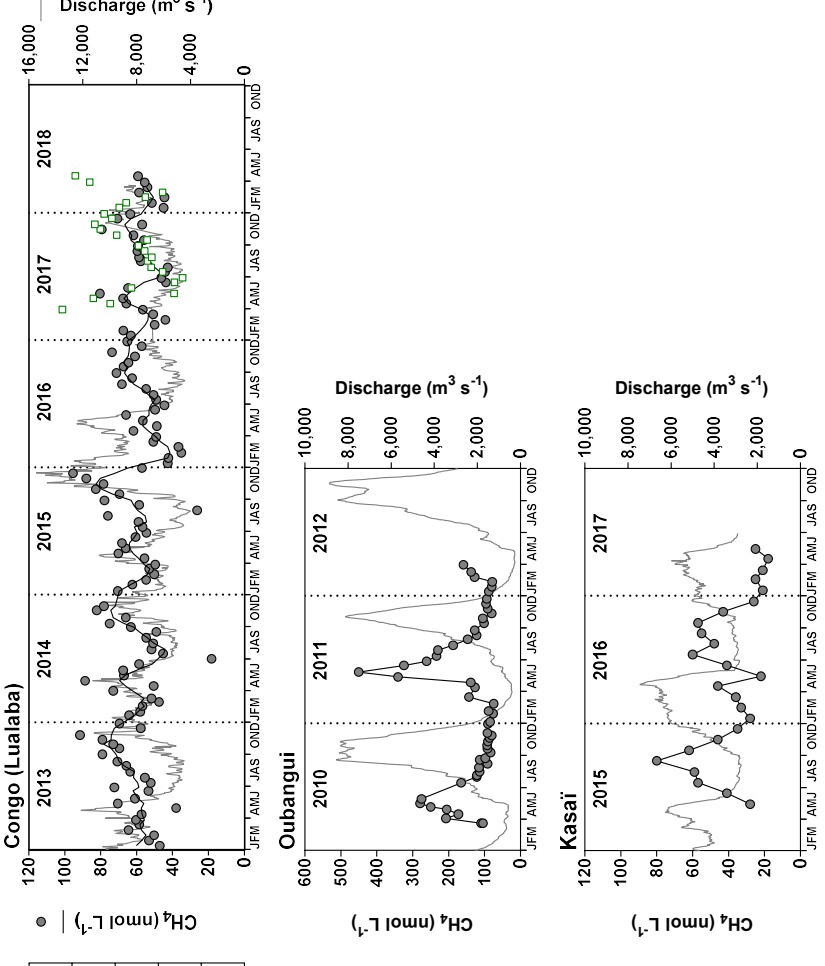

Figure 19: Time series of dissolved $CH_4$ concentration (nmol $L^{-1}$) in surface waters and freshwater discharge (grey line) in the Congo River (at Kisangani, 2013-2018), the Oubangui (at Bangui, 2010-2012), and the Kasaï (at Dima, 2015-2017) Rivers. Black line shows a 5 sample running average. Time series of the partial pressure of $CO_2$ (p$CO_2$ in ppm) in surface waters was also obtained in the Congo River (at Kisangani, 2017-2018). Data in the Oubangui were previously reported by Bouillon et al. (2012; 2013).





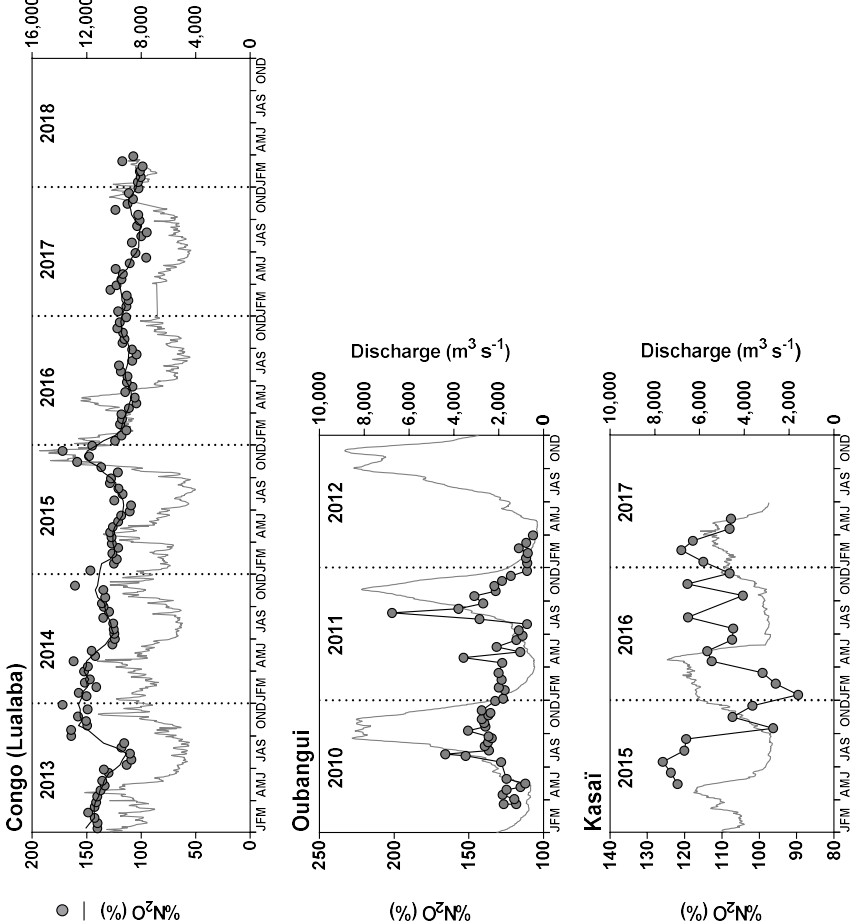

Figure 20: Time series of dissolved N$_2$O saturation level (%N$_2$O in %) in surface waters and freshwater discharge (grey line) in the Congo River (at Kisangani, 2013-2018), the Oubangui (at Bangui, 2010-2012), and the Kasaï (at Dima, 2015-2017) Rivers. Black line shows a 5 sample running average. Data in the Oubangui were previously reported by Bouillon et al. (2012; 2013).




Figure 21: Seasonal amplitude of dissolved CH$_4$ concentration ($\Delta$CH$_4$ in nmol L$^{-1}$) in surface waters as a function of the ratio of seasonal maximum and minimum of freshwater discharge (Qmax:Qmin) in the Congo River (at Kisangani, 2013-2018), the Oubangui (at Bangui, 2010-2012), and the Kasaï (at Dima, 2015-2017) Rivers (Fig. 19).

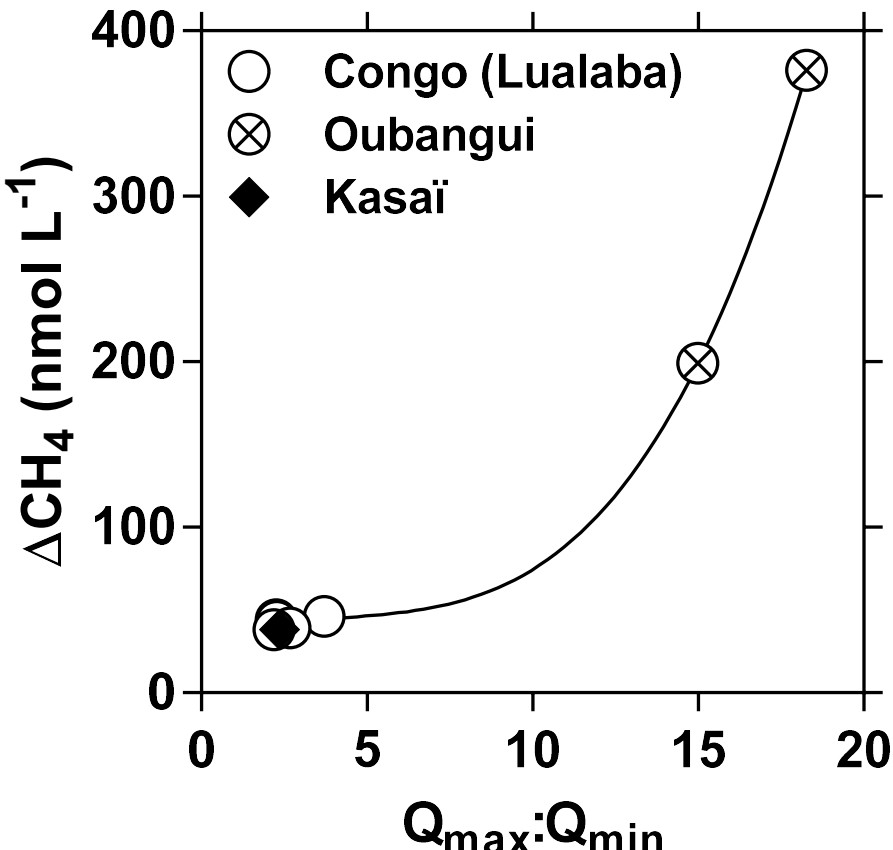