# Peer review of "Spatial and temporal variations of dissolved greenhouse gases (CO2, CH4, N2O) in the Congo River network"

_Biogeosciences, 2019_

## Referee Comment (RC1) · Anonymous Referee #1 · 8 Apr 2019

This manuscript is based on data from10 field expeditions carried along different parts of the lowland Congo river. Measurements of water concentrations of CO2, CH4, and N2O is in focus, but in addition many more variables including temperature, conductivity, pH, turbidity, suspended solids, DOC, POC, cations, dissolved silicate, nitrate, nitrite, ammonia, chlorophyll-a, primary productivity, pelagic respiration, and dissolved O2, methane oxidation, stable isotopic compositions – 13C in CH4, DIC, POC and 18O in water. In addition to the 10 specific expeditions, there were continuous or regular multi-year monitoring of selected variables on a few sites. Therefore, the data reported

is exceptionally rich for any river and even more impressive given that they regard the Congo river. This in itself bring a very high value to the scientific community and we should be grateful for the author's persistence and long-term dedication for studying this very important river basin.

The patterns emerging from this data is very interesting and the authors bring forward a several aspects under the umbrella of a greenhouse gas study. In my view this study contains much more than just greenhouse gases and a general comment would be to broaden the umbrella and title to better reflect all topics actually covered – also those that are not strictly greenhouse gas-related.

In general I find the text well-written and interesting, but I have some questions and comments outlined below that I think need consideration.

General comments:

- The manuscript is long and in some parts a bit hard to follow when it combines many aspects and variables in the same paragraphs. I think all information given is valuable so I am not asking for removing any of them, but if possible to provide a more clear structure and to perhaps shorten the text a bit that would make reading and understanding easier.

- In some cases when trying to explain observed patterns/correlations, it seems single cause-effect patterns are suggested, while other alternatives may also be possible but not mentioned as the main explanation. . . .and sometimes these other alternatives are brought up later in the text, separated from the first explanation. This regards e.g. to what extent patterns observed in the river depend on in-river processes or on what is brought to the river from the soils. Clearly the authors have expert awareness about all possible explanations, so this is a request to clarify that several alternative explanations are possible when relevant.

- I miss method evaluations, sensitivity analyses, reliability checks, and attempts to

validate estimates regarding the greenhouse gas emissions. This is very important because the suggested extrapolated emissions of CO2 and CH4 are exceptionally large – much larger than reported from any other river worldwide - including the Amazon. There are many potentially very large sources of uncertainty in the flux estimation and its upscaling that is not mentioned nor analyzed, and single approaches from the literature are simply accepted without critical discussions and testing other alternatives.

More details on the two last general comments are provided in the detailed comments below.

Detailed comments:

P2 L15: Please clarify how % values link to atmospheric concentrations, eg is 0% or 100% in equilibrium with the atmosphere?

P2 L19-24. Are these explanations for the observed patterns the only options? Could patterns not also simply correspond to the relative input from anaerobic soil water (higher such input would correspond to lower O2 and No3- but higher NH4+). Is the abstract the best place for explanations of correlations if there are several alternatives? Is it not better to simply report the correlations in the abstract and, unless reasons are very clear, leave the discussions on explanations to the discussion section?

P2 L32-33: As in the above comment, I wonder if the proposed direct link between pCO2, CH4 and DOC variability via organic matter processing is the only option. Can transport patterns including the balance between input and output be excluded? It seem reasonable that soil water bringing lots of DOC from surrounding soils could also bring lots of CO2 and CH4, and that this could explain correlations even if carbon processing in the river itself was not the reason, which is also supported by the following text in the manuscript. Therefore, I find this sentence emphasizing a link between pCO2 and CH4 and in-river DOC confusing and also potentially misleading and I suggest to remove it.

P5 L24-26: This sentence claim half of the global wetlands in the tropics and 40% in the northern temperate zone, leaving only 10% for the peatlands in the boreal and subarctic zones. This does not fit with e.g. N. C. Davidson A B D , E. Fluet-Chouinard C and C. M. Finlayson A. 2018. Global extent and distribution of wetlands: trends and issues. Marine and Freshwater Research 69(4) 620-627 https://doi.org/10.1071/MF17019., or with several other wetland extent studies. Please check and revise sentence.

P8 L6-14: Please give more details on the equilibrator system to prevent dependence on access to key method papers (not easy to access papers for all readers worldwide). What was the water residence time relative to the gas exchange equilibration time in the equilibrator?

P10 L12-13: Was linear models best for estimating methane oxidation? Sometimes exponential decay model can work better.

P11 L31: The used model was developed for closed systems. Why was not the open-steady state system model tried (e.g. Happell, J. D.; Chanton, J. P.; Showers, W. S. Geochim. Cosmochim. Acta 1994, 58, 4377-4388.)?

P14 L26. Please provide the equations used. Readers from all countries may not easily get access to other papers and can then not adopt or evaluate this study as should be possible.

P18 L2-18 and elsewhere: Some of the discussions on reasons for observed patterns seem to focus entirely on potential explanatory processes in the river channels, while alternative but not always mentioned explanations could relate more to the balance between different compounds entering the channel from the surrounding soils. Please check for possible alternative explanations to the observed patterns and report all relevant alternatives if not clear that a single explanation is most likely.

P29-34 Section 3.4. The mean concentrations found are very high – at least for CH4. Mean values reported here are more than 2-fold higher than other recent estimates

**BGD**

from the Congo basin by Upstill-Goddard et al 2017 (doi.org/10.5194/bg-14-2267-2017). Further, the extrapolated CO2 emission is 251 TgC yr-1, i.e. 0.25 PgC per year which is more than 10% of the total global estimate of inland water C emissions (2.1 PgC yr-1; Raymond et al 2013 Nature). This value is also 3.3-fold higher than the total terrestrial NEE of the Congo basin (77 Tg C yr-1 as reported by the authors). Hence, given that not all this terrestrial NEE will be exported laterally to the river, degraded there and be emitted as CO2, it means that wetlands in the basin not being a part of the terrestrial or the river network carbon balances, presumably would need to have an NEE of > 200-300 TgC or even more, and it is unclear how reasonable this. It would be good to discuss this and make a reliability check from a catchment carbon mass balance perspective.

Another concern with the extrapolated values are that a rigorous self-evaluation of the methods used regarding flux estimates seems to be missing. What are the main uncertainties? How large are the uncertainties? How robust are the area determinations? What possible area range is there? How about the k estimation? How reliable is that? There are different models to determine k in rivers. Was several of them tried? Do single k models really work across all stream/river orders? Were there any tidal influence that could have affected k in the most down-stream river sections? What ranges were derived? Were there any real in-situ flux measurement that can be used to validate the k estimates? If not, can k or flux estimates be validated in other ways? How about the observed concentrations? In rivers/streams it is often found that water concentrations are regulated by the emission rates so that the highest water concentrations are found were k values and fluxes are lowest, while concentrations are lower where fluxes are high which keep concentrations at low levels. How was this considered in the upscaling? Please add a careful method evaluation and examine and discuss the flux extrapolation critically.

Figure 2. Detailed maps with green borders to the right seem not correctly positioned on the central big map.

Figure 7. The outliers indicated as red dots do not seem to deviate so much from the other nearby dots. How much difference did removal of them make and is such a removal needed for some good reasons?

Tab S1-S3 in Spatial Analysis supplement: Is it realistic that the velocity was lowest in the highest slope low order streams? I realize this is the consequence of Equation S2 and that the river cross section dimensions change much more than the slope and drives this pattern. . . but is it realistic and was this validated? One could imagine a positive correlation between velocity and slope and that velocity-slope relationship eventually break down or change character for higher order rivers.

---

## Referee Comment (RC2) · Anonymous Referee #2 · 9 Apr 2019

General Comments:

This study examines geochemical dynamics, with a focus on greenhouse gases, in the Congo River. The manuscript presents an impressive amount of data and has unprecedented spatiotemporal coverage in a globally important, yet understudied river network. However, the amount of data presented makes the manuscript very hard to read. Further, the study's main conclusion, and title of the paper, "Variations of dissolved greenhouse gases (CO2, CH4, N2O) in the Congo River network overwhelmingly driven by fluvial-wetland connectivity" is based on some major assumptions that

are not adequately addressed in the manuscript or data analyses. The data and discussion driving this conclusion is presented in essentially one paragraph buried in a 34 page manuscript. The manuscript has 21 Figures, many of which are difficult to read because of numerous panels and large ranges in axis scales, so it is difficult to evaluate the data.

I suggest that the authors significantly refocus the manuscript to tell a more concise story and perhaps consider separating this large dataset into several manuscripts. With the current data that is presented, a more appropriate title would be something like: "Geochemical dynamics in the Congo River." A conclusive statement should not be made in the title, as there is little quantitative evidence for the conclusion that has been made other than some simple mass balances with considerable assumptions.

Specific Comments:

P5, L31: Much of the information in this paragraph is perhaps more suitable as a "site description" at the beginning of the methods section

P8, L13: Did the depth of the pump adequately prevent aeration while the ship was traveling at high speeds? Also, please describe how the underway data was post-processed to exclude erroneous data related to factors such as aeration, etc. It is also very difficult to evaluate the quality of data in the figures because of the amount of information displayed in each.

P11, L5: This methods section is poorly organized. Consider breaking into additional subsections. For example, flux calculations are mentioned here, then how k was calculated is not mentioned until the next section, which refers readers to the supplement for the actual details.

Methods: What statistical tests were used to evaluate the data? For example, throughout the results, the word "significantly" is used, and P20, L30 says "The pCO2 values were statistically higher..."

Results and discussion: Perhaps consider a separate results and discussion section. Considering the large amount of data presented, the discussion points get buried in the weeds.

P15: While this information provides interesting information about the Congo River, the volume of information distracts from the overall story about GHG cycling and makes the manuscript difficult to read.

P22, L30: Does the presence of high CH4:CO2 ratios in the wetlands fit with the hypothesis that wetlands drive CO2 emissions in the basin?

P23, L20: Was depth-integrated community respiration calculated? This is not mentioned in the methods. It would also be useful to indicate how much lower CR was from FCO2 rather than simply saying it was lower. Further, only an average value was reported for CR. Please indicate the range of observed values and where these values were observed. This data is perhaps the most important contributor to the main conclusion of the manuscript, but is only described in a few sentences.

P23, L30: It is unclear why a shorter incubation time would alleviate the need to disturb the sample in any way. In biological sciences it is well-known that agitation significantly influences biological oxygen demand compared to stagnant conditions, and that microbial reaction kinetics occur on the time scale of seconds to minutes. for example, see the following studies:

Al-Homoud, A., M. Hondzo, and T. LaPara. 2007. Fluid dynamics impact on bacterial physiology: biochemical oxygen demand. J. Environ. Eng. 133: 226–236.

Coleman, M. E., M. L. Tamplin, J. G. Phillips, and B. S. Marmer. 2003. Influence of agitation, inoculum density, pH, and strain on the growth parameters of Escherichia coli O157: H7—relevance to risk assessment. Int. J. Food Microbiol. 83: 147–160.

Perhaps consider describing this methodological constraint as one factor leading to uncertainty in your conclusions rather than making an excuse and writing off the results of the Richardson and Ward studies. The Ward et al., 2018 study showed that bottle effects are also a factor leading to underestimates, and that rotation velocity also influenced respiration in clearwater rivers with little suspended sediment load. This statement "Nevertheless, it seems unrealistic to envisage an under-estimation of CR by an order of magnitude that would allow reconciling the CR (and NCP) estimates with those of FCO2" does nothing to contribute to the advancement of aquatic sciences by ignoring efforts to improve mechanistic understanding and methodological biases.

P24, L5: The primary conclusion that the authors make, and the title of this paper are based on these several sentences rather than robust quantitative conclusions, and is buried in a 35 page manuscript. The title is not appropriate for this reason.

P34, L10: See previous comment. If this is the main conclusion that is made, the authors should focus much more detailed evaluation to this conclusion rather than diluting the manuscript with massive amounts of unrelated data.

P34, L25: What is meant by "...organic and inorganic CO2 inputs?" Do you mean to say organic matter and CO2 inputs from riparian wetlands." By invoking inputs of wetland OM into rivers as an important pathway for CO2 emissions, then you must also consider that river respiration of this organic matter drives a large fraction of river CO2 degassing. When evaluating how much total CO2 efflux from river channels is from inputs of CO2 from wetlands to the river, the residence time of CO2 must also be considered, i.e. how much respiration is needed to sustain high levels of CO2 downstream of floodplain inputs?

---

## Author Response (AR1)

**Chemical Oceanography Unit - MARE**
University of Liège
Institut de Physique (B5)
B-4000 Liège Belgium
**Alberto Borges**
**Senior research Associate FNRS**

Liège, 30 August 2019

Ji-Hyung Park
Biogeosciences

Dear Dr. Park,

Please find enclosed the revised version of ms bg-2019-68.

First of all, I would like to apologize for the delay in re-submitting the revised version of the ms. Our revised ms was ready for re-submission at the moment we responded on-line to the reviews (BGD forum), but we needed some extra time to account for your additional comments given in the decision letter (below), and there was a collusion with an extensive field expedition.

We have modified the text to accommodate the reviewers comments and suggestions, and we provided a point-by-point reply hereafter.

Reviewer#1 main concerns were to improve the readability of the text, to perform an error analysis, and to check independently if the emissions of gaseous C to the atmosphere could be sustained by external carbon inputs. We addressed these main concerns by adding more sub-section headings and providing a brief "road-map" at the start of each section in the results-discussion; these changes should improve the readability of the text. We added an error analysis on the flux computations, and we estimated the inputs of C from flooded forests to rivers. Other more specific/formal comments from the reviewer were also addressed (description of the equilibrator system, explicit formulation of gas transfer velocity equation, …)

Reviewer # 2 made numerous rather subjective comments/suggestions on the title, structure (splitting discussion-results), and length of paper. Regarding the length there are no size restrictions in BG, and we assume that the length of text and number of figures were validated during the initial evaluation of the ms before publication in the Discussion forum. Our dataset is indeed extensive and hence the manuscript is longer than average, but we feel it is important to keep this dataset together. We did not restructure the text into a separate results and discussion sections as this would have increased the length and would have made the text difficult to read (as it stands there's a clear "story-line"). We have modified the title since both reviewers requested a change of title. Since the reviewer stated that our respiration measurements were the most important aspect of the data-set, we elaborated the analysis of the data, by looking into correlations of respiration and other variables, and looking into variations of the ratio of $CO_2$ emission and respiration as function of stream size. We also address other specific requests of clarification such as air in equilibration system and data post-processing.

Yours sincerely,
Alberto Borges
Senior Research Associate FRS-FNRS

Tél. : +32-4-3663187 Fax : +32-4-3669729
E-mail : alberto.borges@ulg.ac.be
www.co2.ulg.ac.be

**Editor:** Thank you for submitting your manuscript to Biogeosciences. Your manuscript has been reviewed by two reviewers with expertise on GHG dynamics in inland waters. While the reviewers agreed that your manuscript presents comprehensive, invaluable data sets collected in a "globally important, yet understudied" river system for multiple years, their evaluations of the paper quality were starkly contrasting. Irrespective of the overall evaluation, both reviewers raised a number of concerns about not only the structure but also interpretations of some critical data. Therefore, I had to reread the manuscript carefully and double checked on which factual grounds the two reviewers had based their evaluations. Your responses to reviewer comments were thoughtful and detailed, but some important points remain inadequately addressed, as described below.

Given the many raised issues and several points inadequately addressed, I might need to reconsider your manuscript for publication in Biogeosciences after major revisions. I would like to ask you to make all the changes easily identifiable in a marked-up manuscript and a point-by-point reply to reviewer comments. I would also suggest you to specify the line numbers of the revised parts when you respond to the reviewer comments and my own suggestions as follows:

Reply: We thank the editor and both reviewers for the detailed evaluation of the previous version of our work, and we hope that in light of the actual modified version of the ms as well as some additional replies to those already posted in the BDG forum, our work will be considered for publication. Modifications are given in a marked version of the ms (with track changes). Page/line numbers given below refer to this marked version of the ms (not the "clean" version).

**Editor:** 1. Manuscript structure and focusing

In my view, the way you structured data presentations and interpretation lies at the heart of many concerns raised by both reviewers and the second reviewer's negative evaluation. First, I agree with the first reviewer saying that "we should be grateful for the authors' persistence and long-term dedication for studying this very important river basin". I also agree to your statement that "long papers presenting large and broad data-sets…are not automatically of poor quality". However, both reviewers and myself, who all appear keenly interested in the topics addressed in this manuscript, found it difficult to follow many parts due to long and often redundant descriptions. As suggested by the first reviewer, you don't need to remove any of all the important results, but some efficient editing throughout the manuscript would make it a lot easier for readers to follow up the key points you want to highlight. I don't think that your plan (a paragraph at the start of the results-discussion section that explains the "road-map" of the paper) would address the concerns raised by the reviewers and myself.

Reply: We hope that by reading the amended version of the ms, the editor and reviewers will be convinced that the changes we made allow a better flow for reading the ms, and address satisfactorily the concerns on the structure. We think that our original structure was logical and elegant, but this was probably obscured by the lack of headings, and the absence of a clear "road-map". In addition to the already mentioned modifications (P17L21-30, P23L11, P24L19, P29L29, P36L24), we have added some sentences that provide the logical link between some sections (P20L23, P22L5, P24L7). These logical links were clear when we wrote the paper, but we admit they might seem less obvious for readers, and might have given the impression we were switching to an unrelated topic. Finally, please take into account that we spend considerable amount of time thinking about the best way to articulate the presentation and discussion of the data. We admit the

text is long given the very extensive data-set presented. We would like also to point out that we all have different and individual styles to present our work, and that rather long and dense papers as ours might be less common than a few years ago, but we still think this remains valid and relevant, although less fashionable.

**Editor:** Please first consider how to streamline the logical structure of your hypothesis (presumably small carbon export from terra firme – contrasting spatial patterns of CO2/CH4 vs N2O primarily determined by wetlands in CCC – relatively small in-stream respiration compared to riverine CO2 and CH4 emissions). The first reading of the abstract and three R&D sections did not allow me to understand your major point.

Reply: We have re-arranged the first sentences of abstract to clarify the objective (and major point) of the paper.

**Editor:** Furthermore, the Conclusions section is solely focused on your hypothesis regarding the dominant role of terra firme, even though you wanted to present the full spectrum of large data sets as the rational for a long paper. As suggested by the second reviewer, if you want to weave multiple pieces of information into a single coherent storyline around wetland contributions, you might need to consider refocusing and even splitting R&D. Otherwise, please clarify your goals (and hypotheses) in the abstract and introduction and accordingly provide summary paragraphs in each relevant R&D sections that could be understood as stepping stones leading to conclusions. In the current version, key messages are hidden, scattered in separate R&D sections and conclusions focused on single hypothesis spring up suddenly.

Reply: In the conclusions section, we put forward what we think is very most important conclusion of our work. We believe that the conclusions section should not duplicate the abstract and provide a mini summary of the whole paper. Also, reviewer#2 was concerned that the most important findings of our work were buried in the text, the fact that they are highlighted in the Conclusions should allow to address this.

**Editor:** Given the vast volume of data, the long abstract itself is not a problem at all. However, it could be more precise by rearranging some long and redundant sentences. For example, the lines starting from L 12 to 34 on P 3 can be shortened by focusing on key spatial patterns (please refer to the first reviewer's comment on long "explanations of correlations"). In addition, the next page repeats the same spatial patterns, but this time reiterates spatial variations in fluxes. The key message on the role of CCC appears late after just describing variations in concentration in the previous page. Again, streamlining the story based on your key questions would also reduce redundancy here in the abstract.

Reply: We do not feel there is a redundancy in the abstract, but we have re-arranged the first sentences of abstract to clarify the objective of the paper.

**Editor:** 2. Alternative explanations
The two reviewers raised several issues about balancing alternative explanations for the observed patterns. In responding to the first reviewer's question about "what extent patterns observed in the river depend on in-river processes or on what is brought to the river from the soils", you said that "We have modified the text accordingly", without detailing how you would change in which parts.

Reply: It seemed inappropriate to provide line numbers in the on-line BGD replies to the reviewers, since the revision version was not yet available to the reviewers.
The reviewer seemed particularly concerned by N2O dynamics, and text was modified in several portions of the text: P2L28, P21L6,P21L16, P21L22P22L5,P24L7,P26L10. For $CO_2$, a large fraction of the introduction is devoted to provide the different alternative explanations of $CO_2$ origin in rivers. Also, we had already envisaged different explanations for the origin of $CO_2$: P25L5-18, P28L17. We have nevertheless additionally modified text at P3L5, P26L32. In other parts of the text, we have toned down the statements (P19L21, P19L26).

**Editor:** The second reviewer also raised a critical question about the contribution of wetland-derived OM and CO2 to downstream CO2 emissions (river respiration and retention time).

Reply: It seems unreasonable and not very constructive to ask an analysis that is not possible with the available data, since the experiment was not designed to make that particular analysis. We simply cannot address that request in absence of data.

**Editor:** I agree with the reviewers in that your approach is too straightforward focusing only on your hypothesis, not balancing alternative explanations. Please clarify the following uncertainties in relevant Discussion parts and draw your conclusion more carefully considering all these uncertainties (in Conclusions):
- Estimates of NEE and the proportion of the hydrologic C export (including DOC, POC, and DIC) in NEE (you just assumed that C export would comprise a small portion, without referring to any measurements or estimates)

Reply: We did not make those measurements or estimates because the experiments were not designed for that purpose, so we relied on relevant literature. We do address the export of C from terrestrial NEE from literature data.

**Editor:** - The relative contributions of OM and CO2 derived from wetlands in CCC (no data available on measurements in source sediments and waters?)

Reply: This is addressed in P39L9-13.

**Editor:** - Estimates of PP and CR: I am afraid that many reader would not be convinced by your explanation ("it seems unrealistic to envisage an under-estimation of CR by an order of magnitude that would allow reconciling the CR (and NCP) estimates with those of FCO2").

Reply: We agree that this sentence alone is not convincing because out of context. However, our CR values are 10 times too low to account for FCO2, while the experiments of Ward and Richardson show that bottle effects and lack of rotation led at most to an under-estimation of a factor of 2 instead of a factor of 10. This seems to us a convincing argument. This is addressed P28L7-12.

**Editor:** 3. Methodological details
Some of your responses to technical questions are inadequate in terms of providing details. For instance, in responding to the first reviewer's questions about k estimation, you only mentioned your plan for error analysis.

Reply: There are indeed several parameterizations to compute k, and there's even the possibility to actually measure the fluxes directly, but we think that is not very constructive to request measurements that were not made and for which data are not available (direct flux measurements). We chose the parameterization of Raymond et al. (2012) because we think this is best option available in literature, based on the most extensive and comprehensive compilation of *k* measurements in rivers and streams. Also, this is the same parameterization used in the global estimates given by Raymond et al. (2013), and it seems important to keep the same approach for comparative purposes. We also used state-of-the-art spatial data-sets based on the most accurate available estimates of river surface area. We used advanced GIS tools to derive slope and flow. Yet, we do not have the impression that the reviewer was particularly concerned by the way the fluxes were computed but rather by absence of an error analysis that is given P35L6-17.

**Editor:** In another example, your reply to the second reviewer's questions about pump depth and boat speed could not address the technical issue about how to avoid aeration problems arising from the equilibrator employment at high speed. In addition to citing your previous papers, you could briefly describe technical details in the relevant Methods section.

Reply: We really do not know how to better address this comment. We simply never observed bubbles in the water circuit. We have nevertheless specified this as well as that the boat speed was low (P9L21). Further, bubbles can be entrained in presence of waves, however, this never occurred because when the weather became stormy, the crew prudently took the boat to shore and we stopped measurements. We did not add this anecdotic information in the material and method.

**Editor:** The same goes to your statistical analyses.

Reply: Information on statistical analysis (tests and coefficients) are given in detail in the legends of the figures and tables, it would be redundant to repeat this information in the main text.

**Reply to Reviewer#1:**

**Reviewer:** This manuscript is based on data from 10 field expeditions carried along different parts of the lowland Congo river. Measurements of water concentrations of CO2, CH4, and N2O is in focus, but in addition many more variables including temperature, conductivity, pH, turbidity, suspended solids, DOC, POC, cations, dissolved silicate, nitrate, nitrite, ammonia, chlorophyll-a, primary productivity, pelagic respiration, and dissolved O2, methane oxidation, stable isotopic compositions – 13C in CH4, DIC, POC and 18O in water. In addition to the 10 specific expeditions, there were continuous or regular multi-year monitoring of selected variables on a few sites. Therefore, the data reported is exceptionally rich for any river and even more impressive given that they regard the Congo river. This in itself bring a very high value to the scientific community and we should be grateful for the author's persistence and long-term dedication for studying this very important river basin.

The patterns emerging from this data is very interesting and the authors bring forward a several aspects under the umbrella of a greenhouse gas study. In my view this study contains much more than just greenhouse gases and a general comment would be to broaden the umbrella and title to better reflect all topics actually covered – also those that are not strictly greenhouse gas-related.

In general I find the text well-written and interesting, but I have some questions and comments outlined below that I think need consideration.

Reply : We thank the reviewer for the positive evaluation of our work and for the detailed and useful comments for improvement. Both reviewers suggested changes of the title of paper to reflect that the paper reports data other than GHGs. We decided to keep the title paper focused on GHGs that are the central topic of the paper, while other data gravitate around this central topic. A general title such as "biogeochemistry of the Congo" would in our opinion lead to a loss of visibility and attractiveness of the paper, as we expect that the main readership of the paper will be from the GHG community. We have nevertheless conceded to remove "overwhelmingly driven by fluvial-wetland connectivity" to accommodate reviewer#2's comment on this part of the title.

In green, additional information/replies to those already published in the BDG forum (in blue).

**Reviewer:** General comments:
- The manuscript is long and in some parts a bit hard to follow when it combines many aspects and variables in the same paragraphs. I think all information given is valuable so I am not asking for removing any of them, but if possible to provide a more clear structure and to perhaps shorten the text a bit that would make reading and understanding easier.

Reply: We have added new sub-section titles and we have added a few sentences at the start of the results-discussion section that give the "storyline" of the paper to explain to the reader the structure and rational of the paper. This should help the reading and understanding of the paper.

This appears at P17L21-30, P23L11, P24L19, P29L29, P36L24. We have added some sentences that provide the logical link between some sections (P20L23, P22L5, P24L7). These logical links were clear when we wrote the paper, but we admit they might seem less obvious for readers.

**Reviewer:** - In some cases when trying to explain observed patterns/correlations, it seems

single cause-effect patterns are suggested, while other alternatives may also be possible but not mentioned as the main explanation. : : :and sometimes these other alternatives are brought up later in the text, separated from the first explanation. This regards e.g. to what extent patterns observed in the river depend on in-river processes or on what is brought to the river from the soils. Clearly the authors have expert awareness about all possible explanations, so this is a request to clarify that several alternative explanations are possible when relevant.

Reply: We have modified the text accordingly.
For $N_2O$, the text was modified in several portions of the text: P2L28, P21L6,P21L16, P21L22P22L5,P24L7,P26L10. For $CO_2$, a large fraction of the introduction was already devoted to provide the different alternative explanations of $CO_2$ origin, and we had already envisaged different explanations for the origin of $CO_2$ in the discussion at P25L5-18, P28L17. We have nevertheless modified text at P3L5, P26L32. In other parts of the text, we have toned down the statements (P19L21,P19L26).

**Reviewer:** - I miss method evaluations, sensitivity analyses, reliability checks, and attempts to validate estimates regarding the greenhouse gas emissions. This is very important because the suggested extrapolated emissions of CO2 and CH4 are exceptionally large – much larger than reported from any other river worldwide - including the Amazon.
There are many potentially very large sources of uncertainty in the flux estimation and its upscaling that is not mentioned nor analyzed, and single approaches from the literature are simply accepted without critical discussions and testing other alternatives.
More details on the two last general comments are provided in the detailed comments below.

Reply: We have added an error analysis (P35L6), and we estimated the potential export of C from flooded forest to check if it is compatible with the C emissions (P39L9). Refer to detailed replies below.

**Reviewer:** Detailed comments:
**Reviewer:** P2 L15: Please clarify how % values link to atmospheric concentrations, eg is 0% or 100% in equilibrium with the atmosphere?

Reply: This has been clarified in text, 100% corresponds to atmospheric equilibrium (P2L21).

**Reviewer:** P2 L19-24. Are these explanations for the observed patterns the only options? Could patterns not also simply correspond to the relative input from anaerobic soil water (higher such input would correspond to lower O2 and No3- but higher NH4+).

Reply: we agree that denitrification can also occur in soils, and we have modified text to mention "soil or sedimentary denitrification". P2L28

**Reviewer:** Is the abstract the best place for explanations of correlations if there are several alternatives? Is it not better to simply report the correlations in the abstract and, unless reasons are very clear, leave the discussions on explanations to the discussion section?

*Reply: we find the explanations given in the abstract clear and concise. We have the impression the reader might want some words of explanation of the variations above and below atmospheric equilibrium.*

**Reviewer:** P2 L32-33: As in the above comment, I wonder if the proposed direct link between pCO2, CH4 and DOC variability via organic matter processing is the only option. Can transport patterns including the balance between input and output be excluded? It seem reasonable that soil water bringing lots of DOC from surrounding soils could also bring lots of CO2 and CH4, and that this could explain correlations even if carbon processing in the river itself was not the reason, which is also supported by the following text in the manuscript. Therefore, I find this sentence emphasizing a link between pCO2 and CH4 and in-river DOC confusing and also potentially misleading and I suggest to remove it.

*Reply: text mentions "intense processing of organic matter" but does not mention that it is in-stream. We agree that this "processing of organic matter" can occur in soils, and in fact we interpret our patterns of CO2/CH4 as resulting from "intense processing of organic matter" in flooded forests rather than in-stream. For clarity, we have modified the sentence that now reads : "The wide range of $pCO_2$ and $CH_4$ variations was consistent with the equally wide range of $\%O_2$ (0.3-122.8\%) and of dissolved organic carbon (DOC) (1.8-67.8 mg $L^{-1}$), indicative of generation of these two greenhouse gases from intense processing of organic matter either in terra firme soils, wetlands or in-stream." P3L5.*

**Reviewer:** P5 L24-26: This sentence claim half of the global wetlands in the tropics and 40% in the northern temperate zone, leaving only 10% for the peatlands in the boreal and subarctic zones. This does not fit with e.g. N. C. Davidson A B D , E. Fluet-Chouinard C and C. M. Finlayson A. 2018. Global extent and distribution of wetlands: trends and issues. Marine and Freshwater Research 69(4) 620-627 https://doi.org/10.1071/MF17019., or with several other wetland extent studies. Please check and revise sentence.

*Reply: The initial statement was correct in the sense that about 50% of wetlands are located between 33°N and 33°S (sub-tropics and tropics); what we called "northern temperate" probably corresponds to what the reviewer refers to "boreal-subarctic". In order to avoid confusion related to the boundaries between temperate and boreal regions we have rephrased the sentence that now reads: "About half of the global surface area of wetlands is located in the tropics and sub-tropics (33°N-33°S) and the rest in the Northern Hemisphere (Fluet-Chouinard et al. 2015), and more than half of river surface area is located in the tropics and sub-tropics (Allen and Pavelsky 2018)". We have also updated the reference by Fluet-Chouinard et al. 2015 who proposed two additional products to the GLWD that we initially cited (Lenher & Döll 2004) although the relative distribution per latitude is very similar. The Davidson et al. (2018) cited by the reviewer uses the data originally given by Fluet-Chouinard et al. (2015), so we preferred to cite the original paper. P6L26*

**Reviewer:** P8 L6-14: Please give more details on the equilibrator system to prevent dependence on access to key method papers (not easy to access papers for all readers worldwide). What was the water residence time relative to the gas exchange equilibration time in the equilibrator?

Reply: Text now reads: "The equilibrator consisted of a Plexiglas cylinder (height of 70 cm and internal diameter of 7 cm) filled with glass marbles; pumped water flowed from the top to the bottom of the equilibrator at a rate of about 3 L min$^{-1}$; water residence time within the equilibrator was about 10 secs, while 99% of equilibration was achieved in less than 120 secs (Frankignoulle et al. 2001). This type of equilibrator system was shown to be the fastest among commonly used equilibration designs (Santos et al. 2012)." P9L10

**Reviewer:** P10 L12-13: Was linear models best for estimating methane oxidation? Sometimes exponential decay model can work better.

Reply: We agree that both linear and exponential decay models can be used to compute MOX, but we found a linear regression to give a good fit of the data.

**Reviewer:** P11 L31: The used model was developed for closed systems. Why was not the opensteady state system model tried (e.g. Happell, J. D.; Chanton, J. P.; Showers, W. S. Geochim. Cosmochim. Acta 1994, 58, 4377-4388.)?

Reply: We added the following text to justify the choice of the closed system: "The model we used to compute $F$ox applies for closed systems, implying that $CH_4$ is assumed not to be exchanged with surroundings in contrast to open system models. Running river water corresponds to a system that is intermediate between closed and open chemical systems, since it is open to the atmosphere and the sediments, but on the other hand the water parcel can be partly viewed as a closed system being transported downstream with the flow. As such, the water parcel receives a certain amount of $CH_4$ from sediments and then is transported downstream away from the initial input of $CH_4$. We also applied two common open-system models to estimate $F$ox given by Happel et al. (1994) and by Tyler et al. (1997) that have also been applied in lake systems (Bastviken et al. 2002). However, both open-system models gave Fox values > 1 in many cases (data not shown) since the difference between $\delta^{13}C$ of the $CH_4$ source and measured $\delta^{13}C$ in dissolved $CH_4$ was often much higher than expected from the assumed isotopic fractionation (1.02). The same observation ($F$ox > 1) was also reported with open-system models in the lakes studied by Bastviken et al. (2002). Since $F$ox values > 1 are not conceptually possible, we preferred to use instead the results from the closed-system model, although we acknowledge that flowing waters are in fact intermediary systems between closed and open, and that consequently the computed $F$ox values are under-estimated." P14L7-26.

**Reviewer:** P14 L26. Please provide the equations used. Readers from all countries may not easily get access to other papers and can then not adopt or evaluate this study as should be possible.

Reply: Equations were added P13L5

**Reviewer:** P18 L2-18 and elsewhere: Some of the discussions on reasons for observed patterns seem to focus entirely on potential explanatory processes in the river channels, while alternative but not always mentioned explanations could relate more to the balance between different compounds entering the channel from the surrounding soils. Please check for possible alternative explanations to the observed patterns and report all relevant alternatives if not clear that a single explanation is most likely.

Reply: Text was modified accordingly.

For N$_2$O, the text was modified in several portions of the text: P2L28, P21L6,P21L16, P21L22P22L5,P24L7,P26L10. For CO$_2$, a large fraction of the introduction was already devoted to provide the different alternative explanations of CO$_2$ origin, and we had already envisaged different explanations for the origin of CO$_2$ in the discussion at P25L5-18, P28L17. We have nevertheless modified text at P3L5, P26L32. In other parts of the text, we have toned down the statements (P19L21,P19L26).

**Reviewer:** P29-34 Section 3.4. The mean concentrations found are very high – at least for CH4. Mean values reported here are more than 2-fold higher than other recent estimates from the Congo basin by Upstill-Goddard et al 2017 (doi.org/10.5194/bg-14-2267-2017).

Reply: Upstill-Goddard et al (2017) sampled a smaller fraction of the Congo basin (5 major river basins) and in the Western side, while we sampled a much larger number of river sites and on the Eastern side, covering river draining the core of the CCC. This explains why we obtain larger values than the estimates of Upstill-Goddard et al (2017). This is unrelated to differences in methodology, as we inter-calibrated our method with the one of used by Prof. Robert Upstill-Goddard, as reported in Wilson et al. (2018, doi: 10.5194/bg-15-5891-2018), although we admit this was for lower ranges of CH4.

**Reviewer:** Further, the extrapolated CO2 emission is 251 TgC yr-1, i.e. 0.25 PgC per year which is more than 10% of the total global estimate of inland water C emissions (2.1 PgC yr-1; Raymond et al 2013 Nature).

Reply: The Congo river accounts for 3% of the total surface of rivers globally, so it's unsurprising that it should also account to a large amount of the total riverine emissions. The fact that our estimates are high comparted to those reported by Raymond et al. (2013) is not surprising since the single "average" pCO2 value used by Raymond et al (2013) for the Congo (derived from a global pH and alkalinity data-base) is too low compared to our field data (refer to Fig. S18).

**Reviewer:** This value is also 3.3-fold higher than the total terrestrial NEE of the Congo basin (77 Tg C yr-1 as reported by the authors).
Hence, given that not all this terrestrial NEE will be exported laterally to the river, degraded there and be emitted as CO2, it means that wetlands in the basin not being a part of the terrestrial or the river network carbon balances, presumably would need to have an NEE of > 200-300 TgC or even more, and it is unclear how reasonable this.

Reply: The point is that the export of DOC and CO2 from wetlands is much larger than from dry forests so that it is not necessary that wetlands have a NEE above 200-300 TgC as suggested by the reviewer. Please also refer to reply below.

**Reviewer:** It would be good to discuss this and make a reliability check from a catchment carbon mass balance perspective.

Reply: We have addressed this comment and we have added to the discussion the following text: "(…) the carbon export from flooded forests to riverine waters of the Congo basin can be roughly estimated to 396 TgC yr$^{-1}$ and is in excess of the integrated $F$CO$_2$ (based on the export per surface area of flooded forest of 1,100 gC m$^{-2}$ yr$^{-1}$ reported by Abril et al. (2014) for the Central Amazon and on surface area of flooded forest of the CCC (360,000 km$^{-2}$, Bwangoy et al. 2010)" P39L9

**Reviewer:** Another concern with the extrapolated values are that a rigorous self-evaluation of the methods used regarding flux estimates seems to be missing. What are the main incertainties? How large are the uncertainties? How robust are the area determinations?
What possible area range is there? How about the k estimation? How reliable is that? There are different models to determine k in rivers. Was several of them tried? Do single k models really work across all stream/river orders? Were there any tidal influence that could have affected k in the most down-stream river sections? What ranges were derived? Were there any real in-situ flux measurement that can be used to validate the k estimates? If not, can k or flux estimates be validated in other ways? How about the observed concentrations? In rivers/streams it is often found that water concentrations are regulated by the emission rates so that the highest water concentrations are found were k values and fluxes are lowest, while concentrations are lower where fluxes are high which keep concentrations at low levels. How was this considered in the upscaling? Please add a careful method evaluation and examine and discuss the flux extrapolation critically.

Reply: we have now added a detailed error analysis of the fluxes (P35L6) that answers most of the above questions. To answer the other questions of reviewer: The Congo river is different from (for instance) the Amazon as it is isolated from the ocean by a series of water falls leading to an abrupt altitude change of nearly 280m over a few km (downstream of Malebo pool). So there is no tidal influence on the river Congo "proper", but only in the estuary "at sea" that we did not sample. We did carry out a limited number of flux measurements with floating chambers that allow to compute $k_{600}$ values. These numbers are briefly discussed and presented by Borges et al. (2015, doi: 10.1038/NGEO2486) but we think this is not the basis to actually validate the procedure to calculate the $k_{600}$ we used in the upscaling, due to a very restricted number of measurements mainly in high order streams. Text now reads "An error analysis on the GHG flux computation and upscaling was carried out by error propagation of the GHG concentration measurements, the $k$ value estimates, and the estimate of surface areas of river channels to scale the areal fluxes, using a Monte Carlo simulation with 1000 iterations. The uncertainty on the GHG concentrations led to an uncertainty of areal fluxes of ±1.2%, ±2.3% and ±7.1% for $CO_2$, $CH_4$ and $N_2O$, respectively. The uncertainty on $k$ based on tracer experiments is typically ±30.0% (Ulseth et al. 2019). This leads to a cumulated uncertainty of areal fluxes of ±17.6%, ±17.9% and ±19.0% for $CO_2$, $CH_4$ and $N_2O$, respectively. The uncertainty of the river/stream surface areas based on GIS analysis of Allen and and Pavelsky (2018) is estimated to ±10% leading to an overall uncertainty of integrated fluxes of ±18.3%, ±18.3% and ±19.6% for $CO_2$, $CH_4$ and $N_2O$, respectively." (P35L6)

**Reviewer:** Figure 2. Detailed maps with green borders to the right seem not correctly positioned on the central big map.

Reply: We have redrawn the figure to take correct this, and we thank the reviewer for spotting this.

**Reviewer:** Figure 7. The outliers indicated as red dots do not seem to deviate so much from the other nearby dots. How much difference did removal of them make and is such a removal needed for some good reasons?

Reply: The inclusion of the outliers in the regression decreases slightly the $r^2$ and changes imperceptibly the slope; but it does not change the sign of the slope (positive or negative) and does not change the statistical significance of regression. However, one or two points visually stood out, and this was confirmed with the statistical test. We found this information relevant enough to be included in the figures.

**Reviewer:** Tab S1-S3 in Spatial Analysis supplement: Is it realistic that the velocity was lowest in the highest slope low order streams? I realize this is the consequence of Equation S2 and that the river cross section dimensions change much more than the slope and drives this pattern: : : but is it realistic and was this validated? One could imagine a positive correlation between velocity and slope and that velocity-slope relationship eventually break down or change character for higher order rivers.

Reply: While it is seemingly unintuitive, flow velocity has long been shown, both empirically and theoretically, to be lowest in low-order streams. As the reviewer alluded to, the "increase of velocity downstream results from the fact that the increase in depth overcompensates for the decrease in slope" (Leopold and Maddock, 1953, https://pubs.usgs.gov/pp/0252/report.pdf). To answer any questions a reader might have, we added the following sentence to section S2.2 in the Spatial Analysis Supplement: "Although this produces an unintuitive positive relationship between flow velocity and stream order, there is long-standing empirical evidence that shows that mean flow velocity is lower in low-order streams where hydraulic roughness is greatest (Leopold and Maddock, 1953)."

**Anonymous Referee #2**

**Reviewer:** General Comments:
This study examines geochemical dynamics, with a focus on greenhouse gases, in the Congo River. The manuscript presents an impressive amount of data and has unprecedented spatiotemporal coverage in a globally important, yet understudied river network. However, the amount of data presented makes the manuscript very hard to read. Further, the study's main conclusion, and title of the paper, "Variations of dissolved greenhouse gases (CO2, CH4, N2O) in the Congo River network overwhelmingly driven by fluvial-wetland connectivity" is based on some major assumptions that are not adequately addressed in the manuscript or data analyses. The data and discussion driving this conclusion is presented in essentially one paragraph buried in a 34 page manuscript. The manuscript has 21 Figures, many of which are difficult to read because of numerous panels and large ranges in axis scales, so it is difficult to evaluate the data.
I suggest that the authors significantly refocus the manuscript to tell a more concise story and perhaps consider separating this large dataset into several manuscripts. With the current data that is presented, a more appropriate title would be something like: "Geochemical dynamics in the Congo River." A conclusive statement should not be made in the title, as there is little quantitative evidence for the conclusion that has been made other than some simple mass balances with considerable assumptions.

Reply: We thank the reviewer for her/his comments on the previous version of text that stimulated us to clarify the presentation and discussion of our data.
In green, additional information/replies to those already published in the BDG forum (in blue).

Regarding comments on length of the paper and suggestion to rewrite the data-set into 3 separate shorter papers: Young scientists are pressured nowadays into producing as many papers as possible for the advancement of their career, to comply with the "publish or perish" model. They are further pressured to publish in "high-rank" journals that have very strict limitations in terms of word-counts and number of illustrations. This of course encourages slicing data-sets to produce as many as possible short papers, and in recent years this has become common practice. However, this should not forbid long papers presenting large and broad data-sets, unless there is a size limit imposed by the journal, which is not the case for *Biogeosciences*. The Associate Editors of BG are required to carefully evaluate the manuscript before they are published in the *Biogeosciences Discussion* forum, which is an indication that for our present submission the Associate Editor decided that the length of text and number of figures were acceptable for an article in *Biogeosciences*. Furthermore, the conciseness of papers does not guarantee quality, and conversely long papers are not automatically of poor quality. In fact, the contrary seems to be objectively true, longer papers tend to be more cited, we refer to Fox et al. (2016) and other similar studies cited therein.
In our particular case, we have gathered a large data-set on the biogeochemistry of the Congo basin that we think deserves to be presented in sufficient detail, since this is the first time such a comprehensive data-set was gathered for this relatively unexplored river basin. Our personal perception is that we present a self-contained and consistent "story" based on several variables that were chosen to help understanding the patterns of spatial and temporal variations of GHGs, and that for a reader interested in understanding our dataset, the full dataset should be discussed in its totality rather than in separate manuscripts.

To conclude, we feel that slicing the data-set into separate shorter papers might make it easier to digest but this does not outweigh the scientific merit of keeping the data in a single manuscript.

Regarding the title of the paper we agree to remove the "conclusive" element of the title, but want to avoid a more general title along the lines of "Geochemical dynamics in the Congo River" that would in our opinion lead to a loss of visibility and attractiveness of the paper. Dynamics of GHGs are the central topic of the paper, while other data gravitate around this central topic. We think that the primary readership of the paper will be specifically interested in GHGs (rather than geochemistry at large).

Specific Comments:
**Reviewer:** P5, L31: Much of the information in this paragraph is perhaps more suitable as a "site description" at the beginning of the methods section

Reply: this paragraph provides information on the importance of C cycling in the Congo basin and at the same time the paragraph highlights the near lack of information/data. This message is introductory and helps to motivate and justify our research and goes beyond a simple "site description". As such we kept the paragraph in its original location.

**Reviewer:** P8, L13: Did the depth of the pump adequately prevent aeration while the ship was traveling at high speeds? Also, please describe how the underway data was postprocessed to exclude erroneous data related to factors such as aeration, etc. It is also very difficult to evaluate the quality of data in the figures because of the amount of information displayed in each.

Reply: We adjusted the depth of pump at the start of the cruise to avoid bubble entrapment (cavitation) and this was sufficient for the rest of the cruise. Our system is quite robust and rugged based on experience from more than 20 years of field deployments, and during the cruises the system was supervised quasi-constantly, so there was no need for a particular post-processing of the data. There were some occasional problems on rare occasions (clogging of pump or power shortage) that were identified rapidly and logged, and the data were processed (filtered out) accordingly. Should the reviewer want to evaluate the quality of the $pCO_2$ data, the comparison of discrete and continuous measurements of $pCO_2$ is given in Abril et al. (2015, doi: 10.5194/bg-12-67-2015).
Text was modified P9L22.

**Reviewer:** P11, L5: This methods section is poorly organized. Consider breaking into additional subsections. For example, flux calculations are mentioned here, then how k was calculated is not mentioned until the next section, which refers readers to the supplement for the actual details.

Reply: We have added some sub-section titles and we have moved the text giving the parameterization of the gas transfer velocity.

**Reviewer:** Methods: What statistical tests were used to evaluate the data? For example, throughout the results, the word "significantly" is used, and P20, L30 says "The pCO2 values were statistically higher: : :"

Reply: Statements on statistical significance in text refer to figures, where the statistical

significance is shown with symbols and the type of test is explained in the figure legend. We preferred to alleviate the main text from names of statistical tests and the coefficients of statistical significance. We nevertheless added the reference to corresponding figure to the main text each time the word "significant(ly)" was used. We also added a few words on the statistical methods in the Methods section.

**Reviewer:** Results and discussion: Perhaps consider a separate results and discussion section. Considering the large amount of data presented, the discussion points get buried in the weeds.

Reply: While elaborating the manuscript we spent a lot of time contemplating the best way to articulate the text, and we concluded that a joint results and discussion section was the best option since it allows the "story" to flow from the most descriptive aspects (spatial variations) to the most integrative aspects (discussion of fluxes at the scale of the basin) to end with the most conceptual aspects (implications of the comparison with terrestrial NEE). Also, separating results and discussion sections would further increase the length of paper (need to add text to recall results in the Discussion). Hence, rather than splitting Results and Discussion, we have addressed the underlying concern by adding several new sub-section headings, and a paragraph at the start of the results-discussion section that explains the "road-map" of the paper. This should help the readers to pick directly the main points they're interested in.
This appears at P17L21-30, P23L11, P24L19, P29L29, P36L24. We have added some sentences that provide the logical link between some sections (P20L23, P22L5, P24L7). These logical links were clear when we wrote the paper, but we admit they might less obvious for readers.

**Reviewer:** P15: While this information provides interesting information about the Congo River, the volume of information distracts from the overall story about GHG cycling and makes the manuscript difficult to read.

Reply: Virtually no information is easily available in the literature on the variations of basic limnological variables in the Congo River, so we think it is important to present these data, which we admit makes the section a bit descriptive. Whether or not this information is a distraction is fairly subjective, and some readers might be extremely interested by more basic limnological information such as pH or conductivity or more advanced variables such as $\delta^{18}O$-$H_2O$.

**Reviewer:** P22, L30: Does the presence of high CH4:CO2 ratios in the wetlands fit with the hypothesis that wetlands drive CO2 emissions in the basin?

Reply: We do not see a contradiction here. The high $CH_4$:$CO_2$ ratio means that $CH_4$ is relatively higher than $CO_2$, it does not necessarily mean that $CO_2$ is lower in absolute terms. Figure 12 shows that $CO_2$ is higher in rivers draining the Cuvette Centrale Congolaise (CCC), so it does fit with the hypothesis that wetlands drive $CO_2$ emissions in the basin. This means that both CO2 and CH4 increase in the CCC, but the increase of CH4 is relatively more important.

**Reviewer:** P23, L20: Was depth-integrated community respiration calculated? This is not mentioned in the methods.

Reply: We added to the methods the following sentence: "Depth integration was made by multiplying the CR in surface waters by the depth measured at the station with a portable depth meter (Plastimo Echotest-II)."
P11L15

**Reviewer:** It would also be useful to indicate how much lower CR was from FCO2 rather than simply saying it was lower.

Reply: This information was already given in the original submission P23L22-24.

**Reviewer:** Further, only an average value was reported for CR. Please indicate the range of observed values and where these values were observed. This data is perhaps the most important contributor to the main conclusion of the manuscript, but is only described in a few sentences.

Reply: We have added the range of CR values to the text. Since the reviewer acknowledges the CR data are (the most) important, we further analyzed these data. We found no correlation with most variables expected to have some explanatory power (TSM, POC, Chl-a) but we found a relation with DOC that is now included in the manuscript. We also discuss a decreasing relation between the $FCO_2$:CR ratio and stream order. We added the following text: "The $FCO_2$:CR ratio was higher in lower order streams than higher order streams, with median values ranging between 21 and 139 in stream orders 2-5 and between 3 and 17 in stream orders 6-10 (Fig. 16). This indicates a prevalence of lateral $CO_2$ inputs either from soil-water or riparian wetlands in sustaining $FCO_2$ in lower order streams than higher order streams where in-stream $CO_2$ production from net heterotrophy is more important. These patterns are in general agreement with the conceptual frame developed by Hotchkiss et al. (2015), although lateral $CO_2$ inputs were exclusively attributed by these authors to soil-water or ground-water inputs and riparian wetlands were not considered. These patterns are also in agreement with the results reported by Ward et al. (2018) who show that in large high-order rivers of the lower Amazon, in-stream production of $CO_2$ from respiration is sufficient to sustain $CO_2$ emissions to the atmosphere." P27L19

**Reviewer:** P23, L30: It is unclear why a shorter incubation time would alleviate the need to disturb the sample in any way. In biological sciences it is well-known that agitation significantly influences biological oxygen demand compared to stagnant conditions, and that microbial reaction kinetics occur on the time scale of seconds to minutes. for example, see the following studies:
Al-Homoud, A., M. Hondzo, and T. LaPara. 2007. Fluid dynamics impact on bacterial physiology: biochemical oxygen demand. J. Environ. Eng. 133: 226–236.
Coleman, M. E., M. L. Tamplin, J. G. Phillips, and B. S. Marmer. 2003. Influence of agitation, inoculum density, pH, and strain on the growth parameters of Escherichia coli O157: H7ăĂˇTrelevance to risk assessment. Int. J. Food Microbiol. 83: 147–160.
Perhaps consider describing this methodological constraint as one factor leading to uncertainty in your conclusions rather than making an excuse and writing off the results of the Richardson and Ward studies. The Ward et al., 2018 study showed that bottle effects are also a factor leading to underestimates, and that rotation velocity also influenced respiration in clearwater rivers with little suspended sediment load. This statement "Nevertheless, it seems unrealistic to envisage an under-estimation of CR by an order of magnitude that would allow reconciling the CR (and NCP) estimates with those of FCO2"

does nothing to contribute to the advancement of aquatic sciences by ignoring efforts to improve mechanistic understanding and methodological biases.

Reply: We have the impression the reviewer misinterpreted our discussion. We do not have the intention to dismiss the work of Ward and of Richardson, on the contrary, we mentioned upfront that our CR measurements might be under-estimated based on these publications. We have rephrased the sentence as: "We acknowledge that our CR measurements might be under-estimated due to bottle effects and lack of rotation, nevertheless, it seems unrealistic to envisage an under-estimation of CR by an order of magnitude that would allow reconciling the CR (and NCP) estimates with those of $F$CO$_2$." We stand by this statement. It is unlikely that our measurements are 10 times too low, when the experiments of Ward and Richardson show at most an underestimation of a factor of 2. We have removed the statement regarding the incubation time that was initially motivated by the Richardson study for which the experiments were 40 days long, and major differences are only apparent in data from 15 days onwards, from the inspection of their Figure 2. We cannot check the effect of TSM on respiration since Ward et al. (2018) did not explicitly report TSM data at their sites, but it is now well established that the Congo is much less turbid than the Amazon (e.g. Descy et al. 2018) and that effect of rotation on respiration should be linked to the presence of particles. Accordingly, the median of TSM in our data-set (14 mg/L) is more than 2 times lower than at the sites studied by Ward et al. (2018), as reported by Ward et al. (2015).
Sentence was changed from "an order of magnitude" as stated above to "a factor of 10"

**Reviewer:** P24, L5: The primary conclusion that the authors make, and the title of this paper are based on these several sentences rather than robust quantitative conclusions, and is buried in a 35 page manuscript. The title is not appropriate for this reason.

Reply: Our conclusions are built on several lines evidences gathered from a combination of metabolic measurements, stable carbon isotope ratios of DIC, patterns in the spatial variations of CO2, and the comparison between CO2 emissions and terrestrial NEE.. Note also that the primary conclusions of our work are highlighted in the "Abstract" and "Conclusion" so we disagree with the reviewer's comment that they are buried in the text. Nevertheless, we now added several new sub-section headings and added a paragraph at the start of the results-discussion section that explains the "road-map" of the paper. This should help the readers to pick directly the main points they might be interested in.

**Reviewer:** P34, L10: See previous comment. If this is the main conclusion that is made, the authors should focus much more detailed evaluation to this conclusion rather than diluting the manuscript with massive amounts of unrelated data.

Reply: See previous replies regarding subjective appreciations on the length of papers. We do not see how the other data would be unrelated.

**Reviewer:** P34, L25: What is meant by ": : :organic and inorganic CO2 inputs?" Do you mean to say organic matter and CO2 inputs from riparian wetlands." By invoking inputs of wetland OM into rivers as an important pathway for CO2 emissions, then you must also consider that river respiration of this organic matter drives a large fraction of river CO2 degassing. When evaluating how much total CO2 efflux from river channels is from inputs of CO2 from wetlands to the river, the residence time of CO2 must also be considered, i.e. how much respiration is needed to sustain high levels of CO2 downstream of floodplain

inputs?

Reply: Yes, we mean OM inputs from wetlands, we have clarified this in the revised version ("by organic matter inputs as well as direct CO2 inputs"). However, the fact that organic inputs contribute does not necessarily make these to drive a large fraction of river degassing – our comparison of CR and FCO2 addresses this point adequately, we feel. Residence time needs to be accounted when making detailed budgets at small scales, for instance, over a given short stretch of a river. We do not have the data to make such budgeting exercises. We looked at riverine and terrestrial $CO_2$ fluxes integrated at the scale of the entire basin, for which the residence time of water is irrelevant.

**References used (not cited in manuscript)**

Fox, C.W., Paine, C. E. T., and Sauterey B.: Citations increase with manuscript length, author number, and references cited in ecology journals, Ecology and Evolution, 6, 7717–7726, doi:10.1002/ece3.2505, 2016

[revised manuscript text omitted]